# Dynamic Mixture of Experts: An Auto-Tuning Approach for Efficient Transformer Models

**Yongxin Guo**[1,*]  **Zhenglin Cheng**[4,5,6,*]  **Xiaoying Tang**[1,2,3,†]  **Zhaopeng Tu**[8]  **Tao Lin**[5,7,†]

[1]School of Science and Engineering, The Chinese University of Hong Kong, Shenzhen 518172, China
[2]Shenzhen Institute of Artificial Intelligence and Robotics for Society (AIRS), Shenzhen, China
[3]Guangdong Provincial Key Laboratory of Future Networks of Intelligence, Shenzhen, China
[4]Zhejiang University   [5]School of Engineering, Westlake University   [6]SII
[7]Research Center for Industries of the Future, Westlake University   [8]Tencent AI Lab

## Abstract

The Sparse Mixture of Experts (SMoE) has been widely employed to enhance the efficiency of training and inference for Transformer-based foundational models, yielding promising results. However, the performance of SMoE heavily depends on the choice of hyper-parameters, such as the number of experts and the number of experts to be activated (referred to as top-$k$), resulting in significant computational overhead due to the extensive model training by searching over various hyper-parameter configurations. As a remedy, we introduce the Dynamic Mixture of Experts (DynMoE) technique. DynMoE incorporates (1) a novel gating method that enables each token to automatically determine the number of experts to activate. (2) An adaptive process automatically adjusts the number of experts during training. Extensive numerical results across Vision, Language, and Vision-Language tasks demonstrate the effectiveness of our approach to achieve competitive performance compared to GMoE for vision and language tasks, and MoE-LLaVA for vision-language tasks, while maintaining efficiency by activating fewer parameters. Our code is available at https://github.com/LINs-lab/DynMoE.

## 1 Introduction

The scalable nature of Transformer models (Kaplan et al., 2020) has gained remarkable successes across a spectrum of applications, ranging from language (Achiam et al., 2023; Touvron et al., 2023a;b) and vision Kirillov et al. (2023); Peebles & Xie (2023) to cross-modality domains (Liu et al., 2024; Li et al., 2022b; 2023b). To further enhance performance while maintaining high efficiency, Sparse Mixture of Experts (SMoE) has emerged as a promising technique that significantly reduces computation costs during both training and inference stages (Fedus et al., 2022; Lepikhin et al., 2020; Zhang et al., 2022), and has been shown to achieve comparable or superior performance compared to traditional dense models (Li et al., 2022a; Jiang et al., 2024; Dai et al., 2024).

Despite its success, SMoE has an unavoidable drawback: *the performance of SMoE heavily relies on the choice of hyper-parameters*, such as the number of activated experts per token, referred as top-$k$, and the number of experts (Clark et al., 2022; Fan et al., 2024; Yang et al., 2021), denoted as $K$. As illustrated in Figure 1(a), the performance discrepancy of MoE models under various configurations can be approximately 1%-3%. Notably, *identifying the optimal hyper-parameter without a sufficient number of ablation studies is challenging.* As the size of the models continues to grow, this limitation could result in a significant waste of computational resources, and in turn, could hinder the efficiency of training MoE-based models in practice.

To tackle the above problems, the objective of this paper is to explore a novel training technique for MoE models, with the aim of addressing the following core question:

*Is it possible to develop a MoE training strategy that can **automatically** determine the number of experts and the number of activated experts per token during the training process?*

---

*Equal contributions.
†Tao Lin and Xiaoying Tang are corresponding authors.

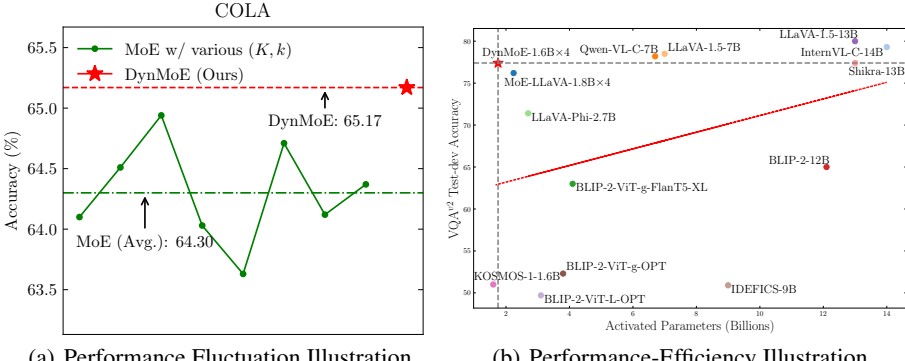

(a) Performance Fluctuation Illustration

(b) Performance-Efficiency Illustration

Figure 1: **Illustration of performance and efficiency of DYNMOE.** In Figure 1(a), we carried out experiments on GLUE benchmark (Wang et al., 2018), employing BERT-large (Devlin et al., 2019) as backbone. In Figure 1(b), we follow the MoE-LLaVA (Lin et al., 2024) settings, the $x$-axis represents the number of activated parameters, while the $y$-axis shows the performance on the Visual Question Answering (VQA) task.

Hence, we introduce the Dynamic Mixture of Experts (DYNMOE) method, which addresses the aforementioned question through the introduction of two innovative components: (1) a top-any gating method that enables each token to autonomously determine the number of experts to activate, thereby allowing different tokens to activate varying numbers of experts; (2) an adaptive training process that dynamically adjusts the number of experts, increasing it when the current quantity is inadequate and removing redundant experts as necessary. Additionally, we introduce a new auxiliary loss function specifically designed to encourage sparsity when employing the top-any gating approach. This loss encourages different experts to be diverse, rather than mandating that all experts be activated with the same frequency. We summarize the contributions of this paper as follows:

- Introducing DYNMOE, a novel method frees the burden of pivotal hyper-parameter selection for MoE training, which is capable of autonomously determining the number of experts and the number of experts to be activated per token.
- Conducting extensive empirical experiments across Vision, Language, and Vision-Language tasks. The results illustrate that DYNMOE achieves comparable or superior performance and efficiency compared to the well-tuned MoE settings (Figure 1(b)).

## 2 RELATED WORKS

The Sparse Mixture of Experts (SMoE) approach (Eigen et al., 2013; Shazeer et al., 2017; Lepikhin et al., 2020) has been proven to effectively enhance the training and inference efficiency of foundational models. Contemporary studies primarily modify the MLP layer of transformer models into multiple expert models and employ a gating network to determine which expert to select. They only choose a subset of experts for each token during both training and inference (Lepikhin et al., 2020; Fedus et al., 2022). Recently, the SMoE structure has shown success in various research areas. For instance, GMoE (Li et al., 2023a) has demonstrated that SMoE can enhance generalization performance in vision tasks. Large Language Models (LLMs) have also employed MoE to simultaneously reduce training and inference costs while improving model performance (Fedus et al., 2022; Jiang et al., 2024; Dai et al., 2024; Ren et al., 2023; Lin et al., 2024). However, most of these models employ standard SMoE structures and apply the SMoE to various tasks. Our paper focuses on improving the MoE training process, which can be easily integrated with these methods.

Recently, some attempts have been made to improve the architecture of MoE models. For example, researchers have investigated the benefits of sample-wise (Ramachandran & Le, 2018; Gross et al., 2017) and token-wise (Shazeer et al., 2017; Riquelme et al., 2021; Fedus et al., 2022) routing. Some studies introduce load balancing loss to ensure that the experts are activated an equal number of times (Lepikhin et al., 2020; Fedus et al., 2022). Expert choice routing (Zhou et al., 2022) addresses load balance by allowing experts to choose tokens; however, this approach also suffers from dropped tokens. SoftMoE (Puigcerver et al., 2023) uses a slot mechanism to simultaneously resolve the issues of load balance and dropped tokens. Nevertheless, these approaches also require pre-defined

hyperparameters, such as the number of experts or the number of experts to be activated. Some studies enable tokens to activate a varying number of experts (Huang et al., 2024; Yang et al., 2024; Huang et al., 2024; Yang et al., 2024). However, these approaches either rely on modifying the routing mechanism from top-$k$ to top-$p$ (which introduces the additional hyperparameter $p$), or use dense training during the initial stages, neither of which provide an optimal implementation. In this paper, we tackle this problem by presenting DYNMOE, an algorithm that automatically determines the number of activated experts for each token and dynamically adds or removes experts during the training process. Furthermore, we introduce a new auxiliary loss function that ensures sparsity when utilizing the DYNMOE algorithm.

## 3 METHOD

In this section, we introduce the Dynamic Mixture of Experts (DYNMOE), an algorithm capable of automatically determining the number of experts and the number of experts to be activated for both training and inference stages. This is achieved through the incorporation of two crucial components:

(1) *The top-any gating method* (Figure 2), which models the gating mechanism as a multi-label classification problem, allowing tokens to decide the number of experts to be activated on their own. This enables different tokens to activate varying numbers of experts, including the option to activate no experts.

(2) *A carefully designed adaptive process* that adds new experts when tokens choose to not activate any existing experts, and removes any surplus experts that have not been activated by any tokens.

The overall process is summarized in Algorithm 1.

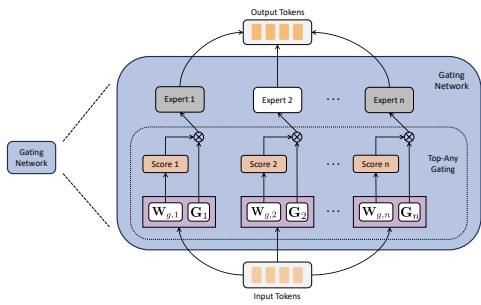

Figure 2: **Illustration of the top-any gating method.** The input tokens pass through the gating weights $\mathbf{W}_{g,e}$ corresponding to each expert $e$, obtaining the gating scores. The scores surpass gates $\mathbf{G}_e$ will activate the subsequent expert. Finally, the expert outputs are combined to produce the output tokens.

### 3.1 TOP-ANY GATING

In this section, we present the superior gating method to eliminate the need for tuning the top-$k$ value. We further improve the test-time inference procedure and introduce an additional auxiliary loss to prevent token dropping and boost efficiency.

**Traditional top-$k$ gating and the limitations.** The traditional top-$k$ gating method takes the token embedding $\mathbf{x}$ as input and employs an additional gating network $g$ to predict the gating scores. These gating scores are then used to determine which experts will be activated for the input tokens. Typically, given token $\mathbf{x} \in \mathbb{R}^d$ as input, the gating process is defined as the follows (Rajbhandari et al., 2022; Hwang et al., 2023):

$$g(\mathbf{x}) \in \mathbb{R}^K := \text{softmax}(\mathbf{W}_g^T \mathbf{x}), \tag{1}$$

where $\mathbf{W}_g \in \mathbb{R}^{d \times K}$ is the parameter of the gating network, and $K$ is the number of experts. Then the output of the MoE layer is defined by

$$\mathbf{y} = \frac{1}{\sum_{e \in \text{Top-}k(g(\mathbf{x}))} g(\mathbf{x})_e} \sum_{e \in \text{Top-}k(g(\mathbf{x}))} g(\mathbf{x})_e E_e(\mathbf{x}), \tag{2}$$

where $E_e(\mathbf{x}) \in \mathbb{R}^d$ is the output of $e$-th expert given input $\mathbf{x}$, and $g(\mathbf{x})_e$ is the $e$-th entry of $g(\mathbf{x})$.

Despite the considerable success of the top-$k$ gating method in enhancing training and inference efficiency, two limitations persist:

1. *The value of $k$ must be fine-tuned to optimize model performance.* As demonstrated in Figure 1(a), the performance of MoE models can vary significantly with different top-$k$ values. This observation has also been noted in recent studies (Clark et al., 2022; Fan et al., 2024; Yang et al., 2021). Consequently, substantial computational resources are needed to identify the optimal value of $k$.

2. *The top-$k$ gating approach assumes that each token must activate the same number of experts, which may not always hold in practice.* For instance, when considering different tasks, there could exist tokens shared by all tasks and those specific to certain tasks, i.e. different tokens could activate different numbers of experts.

**Addressing the limitations of top-$k$ gating by tuning-free top-any gating.**   To address the aforementioned limitations, we propose the ***top-any gating method***, which does not require a predefined value of $k$ and allows different tokens to activate varying numbers of experts during both training and inference stages.

The design of the top-any gating method draws inspiration from the multi-label classification problem. We consider each expert as an individual class and calculate the classification (gating) score for each class (expert) independently. Subsequently, all classes (experts) with scores exceeding the threshold are deemed positive (activated). In detail, given the expert representation matrix $\mathbf{W}_g \in \mathbb{R}^{d \times K}$, where the $k$-th row of $\mathbf{W}_g$ acts as the representation of expert $k$, and an input token $\mathbf{x} \in \mathbb{R}^d$, the key steps of top-any gating can be formulated by the following equation:

$$s(\mathbf{x}) = \frac{\langle \mathbf{x}, \mathbf{W}_g \rangle}{\|\mathbf{x}\| \|\mathbf{W}_g\|}, \tag{3}$$

$$g(\mathbf{x}) = \text{sign}\left(\sigma\left(s(\mathbf{x})\right) - \sigma(\mathbf{G})\right), \tag{4}$$

where $\mathbf{W}_g \in \mathbb{R}^{d \times K}$ and $\mathbf{G} \in \mathbb{R}^K$. To illustrate, we first compute the cosine similarities between the token and the expert representation matrix $\mathbf{W}_g$ and obtain the similarity score $s(\mathbf{x}) \in \mathbb{R}^K$. Then the sigmoid function $\sigma$ is applied to the similarity score $s(\mathbf{x})$ to obtain the scores between $0$ and $1$. Finally, experts with similarity scores greater than the trainable per-expert threshold $\mathbf{G}$ are considered to activate experts for the token $\mathbf{x}$. It is important to note that the sign function does not support back-propagation, and thus we customize the back-propagation process of this part by directly copying the gradient of $g(\mathbf{x})$ to $\sigma\left(s(\mathbf{x})\right) - \sigma(\mathbf{G})$ to effectively bypass the sign function.

Given the gating score $g(\mathbf{x}) \in \mathbb{R}^K$, the number of activated experts is then defined by

$$k := \text{sum}\left(g(\mathbf{x})\right), \tag{5}$$

where $k$ represents the number of experts to be activated for token $\mathbf{x}$. The model output of the MoE layer with the top-any gating method can be derived as follows

$$\mathbf{y} = \frac{1}{k} \sum_{g(\mathbf{x})_e > 0} E_e(\mathbf{x}). \tag{6}$$

**Remark 3.1** (Discussion on not to consider the magnitude of scores when averaging the expert outputs.)**.** *In our top-any gating approach, the scores of different experts are calculated independently. As a result, the scores of different experts may have different scales and ranges. For instance, there may be cases where the scores of Expert 1 are within the range of (0.1, 0.2), but the scores of Expert 2 are within the range of (0.8, 0.9). To avoid this mismatch, we have decided not to consider the magnitude of scores in Equation (6). Ablation studies can be found in Table 23.*

**Improving the top-any gating during test-time to prevent token dropping.**   To facilitate the design of the adaptive expert number process, we did not impose a minimum value on $k$. Consequently, some tokens may not activate any experts. To address this issue, during model performance evaluation, we modify the top-any gating to enable top-1 gating for tokens that do not choose to activate any experts. In detail, for the input token $\mathbf{x}$ with $\text{sum}(g(\mathbf{x})) = 0$, the modified gating score $\tilde{g}(\mathbf{x})$ is obtained by

$$\tilde{g}(\mathbf{x})_k = \begin{cases} 0 & k \neq \arg\max_k \sigma(s(\mathbf{x})), \\ \sigma(s(\mathbf{x})) & k = \arg\max_k \sigma(s(\mathbf{x})). \end{cases} \tag{7}$$

**Guarding efficiency for top-any gating by auxiliary loss.**   The primary goal of using MoE models is to improve the training and inference efficiency. However, in the absence of a cap on the maximum number of activated experts, tokens might activate all experts, which is counterproductive to our primary goal.

Using an auxiliary loss as a regularization over experts may alleviate our issue. However, existing auxiliary loss methods (Lepikhin et al., 2020; Fedus et al., 2022; Wu et al., 2024) are primarily designed to ensure load balancing across experts and thus cannot align with our objectives. While activating all experts can indeed achieve load balancing, it contradicts our aim of improving efficiency by limiting the number of activated experts. Therefore, we need a solution that not only ensures load balancing but also restricts the number of activated experts [1].

As a remedy, we propose a new auxiliary loss, namely *sparse and simple gating loss*, as shown in (8). The *diversity loss* and *simplicity loss* in (8) work together to improve the efficiency of the model by addressing different aspects of the expert representations. On one hand, the *diversity loss* encourages independence among the $\mathbf{W}_g$ representations of various experts. It serves two purposes: First, it prevents a high degree of similarity between experts, thereby enhancing the model's representational capacity; Second, it guides tokens to avoid simultaneous activation of all experts, thereby promoting sparse gating for improved efficiency. On the other hand, the *simplicity loss* normalizes $\mathbf{W}_g$ to avoid excessively large values within the matrix, which helps maintain numerical stability and prevents overfitting due to extreme parameter values. The detailed loss function is defined as follows:

$$\mathcal{L} = \underbrace{\left\|\mathbf{W}_g^T \mathbf{W}_g - \mathbf{I}_K\right\|_2}_{\text{diversity loss}} + \underbrace{\frac{1}{K}\sum_{e=1}^{K}\|\mathbf{w}_{g,e}\|_2}_{\text{simplicity loss}}, \tag{8}$$

where $\mathbf{I}_K$ is the identity matrix with dimension $K$, and $\mathbf{w}_{g,e} \in \mathbb{R}^d$ is the $e$-th element of $\mathbf{W}_g$, indicating the representation of the $e$-th expert.

## 3.2 ADAPTIVE TRAINING PROCESS

In this section, we elaborate on the adaptive training process, which is designed to automatically determine the number of experts. As illustrated in Figure 3, the adaptive process consists of three parts, namely (1) *Routing Recording*: recording the routing results during training; (2) *Adding Experts*: adding new experts when tokens choose not to activate any existing experts; and (3) *Removing Experts*: removing experts that have not been chosen by any tokens. To promising efficiency and avoiding burden communication, we only check if experts required to be added or removed every 100-300 iterations.

***Routing Recording.*** To facilitate the removal and addition of experts, it is essential to track the routing status. Specifically, we record two key pieces of information for each MoE layer: (1) For each expert $e$, we record the time at which expert $e$ is activated, denoted as $\mathbf{R}_E \in \mathbb{R}^K$ (as shown in Line 9 of Algorithm 1). (2) For input data that does not activate any expert, we compute the sum of their embeddings $\mathbf{x}$ as $\mathbf{R}_S \in \mathbb{R}^d$ (as outlined in Line 10 of Algorithm 1). Note that this approach simplifies the expert addition process: by using the token embeddings to initialize the expert representation $\mathbf{W}_g$, we can achieve a high similarity score between these tokens and the new experts, ensuring that the new expert will be activated by these tokens when added.

As demonstrated in Algorithm 1, we utilize $flag_s$ and $flag_f$ to determine when to start and stop routing recording. Users can control these two flags as needed.

***Adding Experts*** **when there exist tokens that choose not to activate any experts.** We add new experts when the recorded $\mathbf{R}_S \neq \mathbf{0}$, as some tokens do not activate any experts and $\mathbf{R}_S$ is the sum of these tokens. Therefore, given $K$ activated experts and new expert $K + 1$, we initialize $\mathbf{W}_{g,K+1} = \frac{\mathbf{R}_S}{\|\mathbf{R}_S\|}$ and $\mathbf{G}_{K+1} = \mathbf{0}$. Moreover, due to the device constrain, the maximum number of experts should be constrained. We set the maximum number of experts to 16 for vision and language tasks, and 4 for vision-language tasks in practice. Discussions on additional strategies for initializing new experts can be found in Appendix 8.4.

***Removing Experts*** **when there exist experts not activated by any token.** We remove experts when there is an expert $e$ such that $\mathbf{R}_E^e = \mathbf{0}$ (as shown in Line 13 in Algorithm 1), which indicates that there is no token choose to activate the expert $e$.

---

[1]We also conducted experiments incorporating other auxiliary losses with DYNMOE, as shown in Table 15.

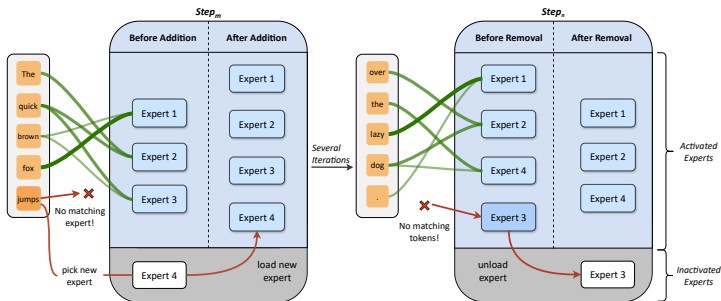

Figure 3: **Elaboration on the adaptive training process.** We visualize the adaptive training process of DYNMoE, including record routing, experts adding, and experts removing. The green strip connecting the token and the expert indicates records of a token routing to an expert. The red arrow at the bottom part of the figure shows where and when expert addition and removal happens.

## 4 EXPERIMENTS

In this section, we carry out experiments to address the following questions:

- **Q1**: Can DYNMoE achieve competitive performance among different MoE settings? See 4.2.
- **Q2**: Can DYNMoE handle tasks with varying modalities and scales? See 4.3.
- **Q3**: Will the model trained by DYNMoE maintain sparsity to ensure efficiency? See 4.4.
- **Q4**: Can DYNMoE offer insights that could guide the design of MoE models? See 4.5.

Additional numerical results, including (1) detailed results on vision and language tasks, (2) ablation studies on auxiliary losses, (3) comparison to top-p gating baselines, (4) pretraining and fine-tuning results on more vision tasks, (5) training efficiency evaluation, (6) overhead of introducing top-any gating, and (7) ablation studies on the weights for averaging expert outputs, can be found in the Appendix 8.

### 4.1 EXPERIMENT SETUP

To answer the above four questions, we conduct experiments on Vision, Language, and Vision-Language tasks. The details are shown in the following.

- **Vision Task.** For the vision tasks, we follow the same settings as in GMoE (Li et al., 2023a). We employ the pre-trained ViT-S/16 Dosovitskiy et al. (2020) model and evaluate it on the DomainBed (Gulrajani & Lopez-Paz, 2020) benchmark. Our experiments encompass four Domain Generalization datasets: PACS (Li et al., 2017), VLCS (Albuquerque et al., 2019), Office-Home (Venkateswara et al., 2017), and DomainNet (Peng et al., 2019). All results are reported using the train-validation selection criterion.
- **Language Task.** The language tasks adhere to the same settings as those in MoEfication (Zhang et al., 2022) and EMoE (Qiu et al., 2023). The MoE models are built upon the BERT-large (Devlin et al., 2019) architecture using the MoEfication method and are fine-tuned on GLUE (Wang et al., 2018) tasks, which include COLA (Warstadt et al., 2019), QNLI (Wang et al., 2018), RTE (Bentivogli et al., 2009), MNLI (Xu et al., 2020), and MRPC (Dolan & Brockett, 2005). For each MoE setting, we tune the learning rates in {2e-5, 3e-5, 5e-5} and report the best results.
- **Vision-Language Task.** The vision-language tasks follows the setting in MoE-LLaVA (Lin et al., 2024), where we use StableLM-2-1.6B (Bellagente et al., 2024), Qwen-1.8B (Bai et al., 2023) and Phi-2-2.7B (Hughes) as backbone language models, and use clip-vit-large-patch14-336 (Radford et al., 2021) as the vision encoder. The models are evaluated on image understanding benchmarks including VQA-v2 (Goyal et al., 2017), GQA (Hudson & Manning, 2019), VisWiz (Gurari et al., 2018), ScienceQA-IMG (Lu et al., 2022), TextVQA (Singh et al., 2019), POPE (Li et al., 2023c), MME (Yin et al., 2023), MMBench (Liu et al., 2023), LLaVA-Bench (in-the-Wild) (Liu et al., 2024), and MM-Vet (Yu et al., 2023). Furthermore, we keep routing records in our model during testing time. For each benchmark, we collect the number of experts' activations per MoE layer and total processed tokens during testing. The hyper-parameter settings are the same to MoE-LLaVA for fair comparision.

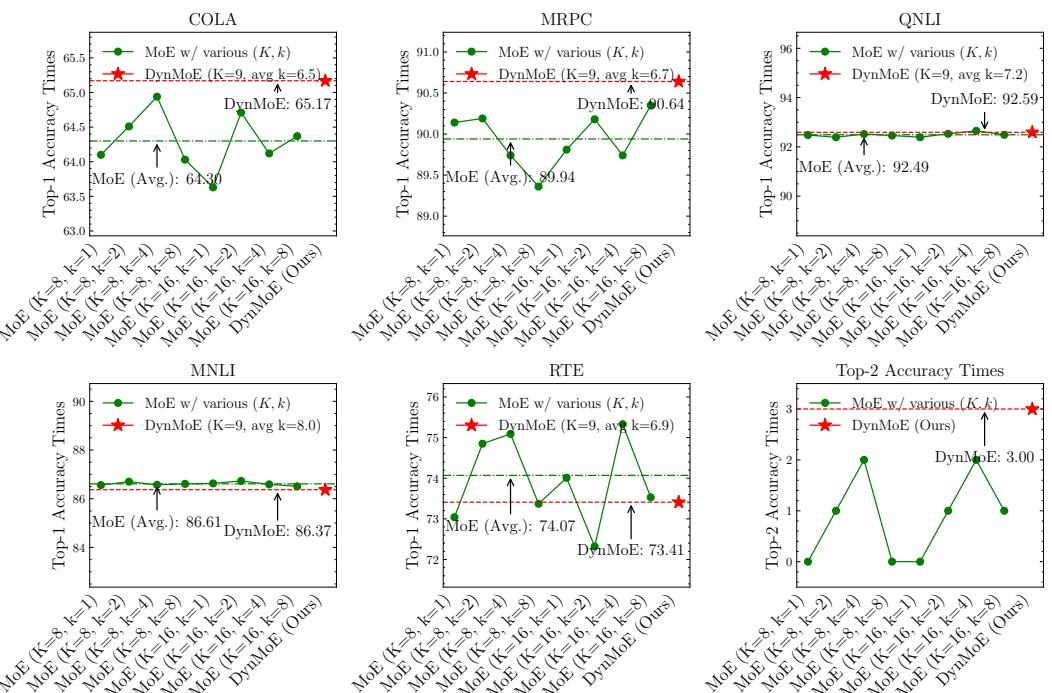

Figure 4: **Performance of DYNMOE on language tasks.** We conduct experiments on the GLUE benchmark. The $x$-axis represents MoE settings with varying $K$ and top-$k$ values. The $y$-axis denotes the model's performance. Dashed lines indicate the average performance across different settings, as well as the performance of DYNMOE. For all the MoE settings, we tune the learning rates in {2e-5, 3e-5, 5e-5} and report the best results. We also report the times when each MoE setting attains the top-2 best results across all configurations.

## 4.2 A1: DYNMOE ACHIEVES COMPETITIVE PERFORMANCE AMONG VARIOUS MOE SETTINGS

In this section, we carry out experiments on the GLUE benchmark (Wang et al., 2018), varying the number of experts ($K$) and the value of top-$k$. The results of these experiments can be observed in Figure 4. More detailed results of each MoE setting can be found in Tables 6-10 of Appendix.

**The performance of DYNMOE surpasses the average performance among various MoE settings.**
As seen in Figure 4, we can observe that

1. The DYNMOE outperforms the average performance for various $K$ and top-$k$ values in most tasks. DYNMOE also achieves the highest number of top-1/2 best performances among all MoE settings, demonstrating its competitive performance.
2. The performance fluctuates considerably with different $K$ and top-$k$ values, such as up to 3.0% on the RTE task and 1.3% on the COLA task. DYNMOE overcomes this issue by not requiring pre-defined $K$ and top-$k$ values.
3. The performance gain of specific $K$ and top-$k$ choice is not consistent among tasks. For instance, the $K = 16, k = 4$ setting performs well on QNLI but poorly on MRPC. In contrast, the DYNMOE always achieve competitive performance among tasks.

## 4.3 A2: DYNMOE CAN HANDLE VISION, LANGUAGE, AND VISION-LANGUAGE TASKS

In addition to Language tasks, we also conduct experiments on Vision and Vision-Language tasks to verify the performance of DYNMOE on different modalities and task scales. The results can be found in Tables 1, and 2.

**The effectiveness of DYNMOE remains consistent in both Vision and Vision-Language tasks.**
Compared to the standard MoE, we can observe the following: **A.** DYNMOE outperforms standard MoE with well-tuned learning rate, number of experts, and top-$k$ (Qiu et al., 2023) in Vision tasks. The performance difference between DYNMOE and another well-tuned MoE setting in (Li et al., 2023a), falls within the range of random fluctuation. **B.** When using StableLM-1.6B and

Table 1: **Performance of DYNMoE on vision tasks**: Our study investigates the performance of DYNMoE on vision tasks using the DomainBed benchmark, with ViT-small serving as the backbone model. The effectiveness of GMoE is elucidated based on meticulously tuned results as presented in the previous works Li et al. (2023a) and Qiu et al. (2023). In our implementation of DYNMoE, we configure the maximum number of experts to 8, with an initial setting of 6 experts. The number of experts is dynamically adjusted in each iteration for DYNMoE. We also report the performance of DYNMoE using Gshard loss (Lepikhin et al., 2020) as the auxiliary loss.

| Algorithms | PACS | VLCS | OfficeHome | DomainNet | Average |
|---|---|---|---|---|---|
| GMoE (in Li et al. (2023a)) | 88.1 | 80.2 | 74.2 | 48.7 | 72.8 |
| GMoE (carefully tuned (Qiu et al., 2023)) | 87.7 | 79.6 | 73.1 | - | - |
| GMoE (with DYNMoE, Gshard Loss) | 88.4 | 79.4 | 73.6 | 47.4 | 72.2 |
| GMoE (with DYNMoE, Diverse and Simple Gating Loss) | 87.6 | 80.3 | 73.5 | 48.2 | 72.4 |

Table 2: **Performance of DYNMoE on vision-language tasks**: Our study investigates the performance of DYNMoE-LLaVA on image understanding benchmarks. Evaluation Benchmarks include VQA-v2; GQA; VisWiz; $SQA^I$ (ScienceQA-IMG); $VQA^T$ (TextVQA); POPE; MME; MMB (MMBench); $LLaVA^W$ (LLaVA-Bench (in-the-Wild)); MM-Vet. For a fair comparison, we set the maximum number of experts to 4 for DYNMoE-LLaVA (the same as the number of experts in MoE-LLaVA) and set the initial number of experts to 2. $N_A$ indicates the number of activated parameters.

| Algorithms | $N_A$ | $VQA^{v2}$ | GQA | VisWiz | $SQA^I$ | $VQA^T$ | POPE | MME | MMB | $LLaVA^W$ | MM-Vet |
|---|---|---|---|---|---|---|---|---|---|---|---|
| *Dense* | | | | | | | | | | | |
| LLaVA-1.5 (Vicuna-13B) | 13B | 80.0 | 63.3 | 53.6 | 71.6 | 61.3 | 85.9 | 1531.3 | 67.7 | 70.7 | 35.4 |
| LLaVA-1.5 (Vicuna-7B) | 7B | 78.5 | 62.0 | 50.0 | 66.8 | 58.2 | 85.9 | 1510.7 | 64.3 | 63.4 | 30.5 |
| LLaVA-Phi (Phi-2-2.7B) | 2.7B | 71.4 | - | 35.9 | 68.4 | 48.6 | 85.0 | 1335.1 | 59.8 | - | 28.9 |
| *Sparse (StableLM-1.6B)* | | | | | | | | | | | |
| MoE-LLaVA ($K=4, k=2$) | 2.06B | 76.7 | 60.3 | 36.2 | 62.6 | 50.1 | 85.7 | 1318.2 | 60.2 | 86.8 | 26.9 |
| DYNMoE-LLaVA (avg $k=1.25$) | 1.75B | 77.4 | 61.4 | 40.6 | 63.4 | 48.9 | 85.7 | 1300.9 | 63.2 | 86.4 | 28.1 |
| *Sparse (Qwen-1.8B)* | | | | | | | | | | | |
| MoE-LLaVA ($K=4, k=2$) | 2.24B | 76.2 | 61.5 | 32.6 | 63.1 | 48.0 | 87.0 | 1291.6 | 59.7 | 88.7 | 25.3 |
| DYNMoE-LLaVA (avg $k=1.86$) | 2.19B | 76.4 | 60.9 | 32.4 | 63.2 | 47.5 | 85.8 | 1302.4 | 61.3 | 89.2 | 24.2 |
| *Sparse (Phi-2-2.7B)* | | | | | | | | | | | |
| MoE-LLaVA ($K=4, k=2$) | 3.62B | 77.6 | 61.4 | 43.9 | 68.5 | 51.4 | 86.3 | 1423.0 | 65.2 | 94.1 | 34.3 |
| DYNMoE-LLaVA (avg $k=1.68$) | 3.35B | 77.9 | 61.6 | 45.1 | 68.0 | 51.8 | 86.0 | 1429.6 | 66.6 | 95.6 | 33.6 |

Phi-2-2.7B as the backbone, the performance of DYNMoE-LLaVA surpasses that of MoE-LLaVA. **C.** With Qwen-1.8B as the backbone, the performance of DYNMoE-LLaVA remains comparable to MoE-LLaVA. In this setting, the average top-$k$ of DYNMoE-LLaVA (avg $k = 1.86$) is also close to the MoE-LLaVA setting ($k = 2$). **D.** In the BERT experiments (Figure 4), DYNMoE generally activate more experts for each token compared to larger scale MoE-LLaVA experiments (Table 2). This observation aligns with the BERT experiments results obtained when using a fixed k value, i.e., k=4 generally performs better among the set {1,2,4,8}.

### 4.4 A3: DYNMoE MAINTAINS EFFICIENCY BY ACTIVATING LESS PARAMETERS

In this section, we aim to demonstrate that although we did not enforce sparsity on the DYNMoE models, the trained DYNMoE models are still sparse, promising improved inference efficiency.

**DYNMoE-LLaVA activates fewer parameters compared to MoE-LLaVA.** In Table 2, we display the number of activated parameters in the "$N_A$" column. When using StabeLM-1.6B as the backbone, DYNMoE-LLaVA activates approximately 15.0% fewer parameters than MoE-LLaVA. For Qwen-1.8B, DYNMoE-LLaVA activates about 2.2% fewer parameters than MoE-LLaVA. For Phi-2-2.7B, DYNMoE-LLaVA activates about 7.5% fewer parameters than MoE-LLaVA. In these three cases, the reduction in activated parameters does not compromise the model's performance.

**Ablation studies on the value of top-$k$ during test.** In Table 3, we examine the performance of DYNMoE-LLaVA when using different top-$k$ values during the testing phase. The results indicate that (1) The original DYNMoE-LLaVA outperforms other settings in most cases while activating the

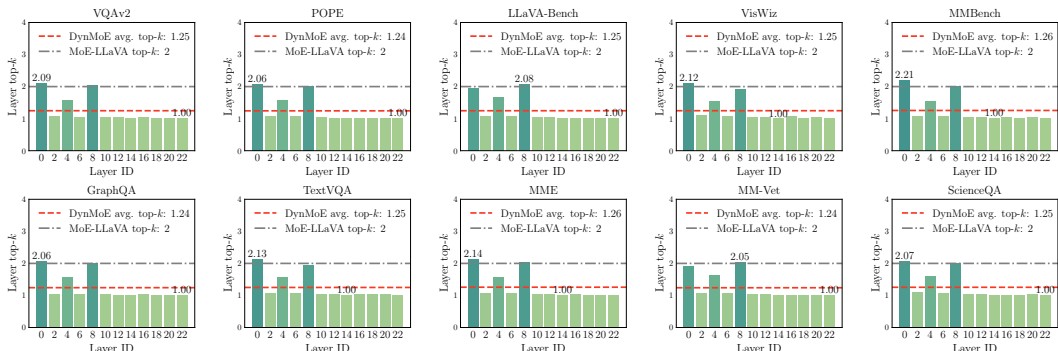

Figure 5: **Average top-$k$ activated experts of DYNMOE on vision-language benchmarks.** We record average top-$k$ activated experts for each MoE layer when using StableLM-1.6B as the language model backbone.

Table 3: **Ablation studies on the value of top-$k$ during test.** We train the models using DYNMOE and set different values of top-$k$ during the test. Training and evaluation settings are identical to that of Table 2.

| Algorithms | $N_A$ | VQA$^{v2}$ | GQA | VisWiz | SQA$^I$ | VQA$^T$ | POPE | MME | MMB | LLaVA$^W$ | MM-Vet |
|---|---|---|---|---|---|---|---|---|---|---|---|
| *StableLM-1.6B* | | | | | | | | | | | |
| DYNMOE-LLaVA | 1.75B | 77.4 | 61.4 | 40.6 | 63.4 | 48.9 | 85.7 | 1300.9 | 63.2 | 86.4 | 28.1 |
| DYNMOE-LLaVA ($k=2$) | 2.06B | 76.9 | 61.0 | 39.1 | 62.1 | 49.2 | 85.7 | 1320.4 | 62.4 | 73.6 | 28.2 |
| DYNMOE-LLaVA ($k=3$) | 2.47B | 76.8 | 60.7 | 37.0 | 62.6 | 48.9 | 85.5 | 1306.9 | 62.5 | 74.0 | 26.8 |
| DYNMOE-LLaVA ($k=4$) | 2.89B | 76.8 | 60.5 | 34.8 | 61.9 | 49.0 | 85.8 | 1321.9 | 61.9 | 75.8 | 27.8 |
| *Qwen-1.8B* | | | | | | | | | | | |
| DYNMOE-LLaVA | 2.19B | 76.2 | 61.5 | 32.6 | 63.1 | 48.0 | 87.0 | 1291.6 | 59.7 | 88.7 | 25.3 |
| DYNMOE-LLaVA ($k=2$) | 2.24B | 76.2 | 60.8 | 33.8 | 62.2 | 47.7 | 87.5 | 1281.3 | 60.4 | 91.3 | 23.0 |
| DYNMOE-LLaVA ($k=3$) | 2.65B | 76.2 | 60.5 | 32.2 | 62.9 | 48.1 | 88.4 | 1263.7 | 60.7 | 87.8 | 23.4 |
| DYNMOE-LLaVA ($k=4$) | 3.05B | 75.7 | 60.0 | 31.6 | 62.8 | 48.3 | 88.1 | 1263.4 | 61.0 | 86.7 | 23.7 |
| *Phi-2-2.7B* | | | | | | | | | | | |
| DYNMOE-LLaVA | 3.35B | 77.9 | 61.6 | 45.1 | 68.0 | 51.8 | 86.0 | 1429.6 | 66.6 | 95.6 | 33.6 |
| DYNMOE-LLaVA ($k=2$) | 3.62B | 77.8 | 61.5 | 41.6 | 67.6 | 51.8 | 85.5 | 1433.5 | 66.8 | 95.1 | 32.7 |
| DYNMOE-LLaVA ($k=3$) | 4.46B | 77.7 | 61.8 | 42.0 | 68.0 | 52.3 | 86.3 | 1438.1 | 66.8 | 94.3 | 30.8 |
| DYNMOE-LLaVA ($k=4$) | 5.30B | 77.5 | 61.4 | 41.7 | 68.0 | 52.4 | 87.0 | 1431.5 | 66.5 | 95.8 | 32.8 |

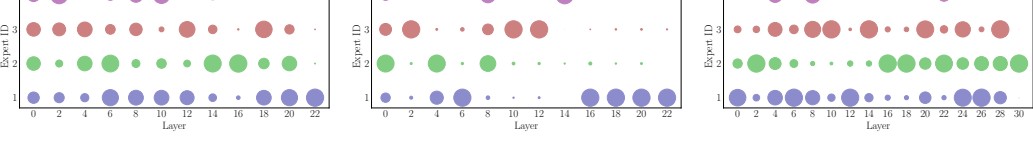

(a) Activation frequency (Qwen)  (b) Activation frequency (StableLM)  (c) Activation frequency (Phi-2)

Figure 6: **Statistics of expert activation frequency in different layers.** We report the frequency of expert activations in various layers for the VQA task. Larger circles indicate experts that are activated more frequently.

Table 4: **Efficiency evaluation of DYNMOE comparing to MoE-LLaVA.** We conduct experiments on single A100 GPU (80 GB) paired with 16 CPUs using identical environment and identical inference configurations. We report the performance of MoE-LLaVA using DeepSpeed's top-2 gating implementation. The symbols ↓ and ↑ indicate that lower and higher values, respectively, denote better performance. The results in this table are averaged over 5 trials, with the first sample excluded to avoid measuring unnecessary memory allocations. Other metrics are reported in Table 24.

| Model | Memory ↓ (GB) | Throughput ↑ (token / second) | First Token Latency ↓ (ms) | Wall-clock Time ↓ (second / sample) | Routing Time ↓ (ms / sample × layer) | Gating Time ↓ (ms / sample × layer) | Expert Passing Time ↓ (ms / sample × layer) |
|---|---|---|---|---|---|---|---|
| Dense-LLaVA (StableLM-1.6B) | 3.68 | 32 | 72 | 4.7 | - | - | - |
| MoE-LLaVA (StableLM-1.6B×4) | 5.98 | 27 | 137 | 6.2 | 0.04 | 1.23 | 1.30 |
| DYNMOE (StableLM-1.6B×4) | 5.98 | 30 | 124 | 5.7 | 0.52 | 0.81 | 1.17 |

fewest number of parameters. (2) Compared to the StableLM-1.6B backbone, DYNMOE-LLaVA trained with the Qwen-1.8B backbone sometimes favors activating two experts. This observation aligns with the fact that DYNMOE-LLaVA also chooses to activate about 2 experts (see Table 2).

**Inference efficiency of DYNMOE.** To further evaluate the inference efficiency of DynMoE, we have compared its FLOPs, MACs, speed, and memory usage to those of MoE-LLaVA. The results in Table 4 show that: (1) DYNMOE chieves higher throughput, lower latency, and reduced wall-clock

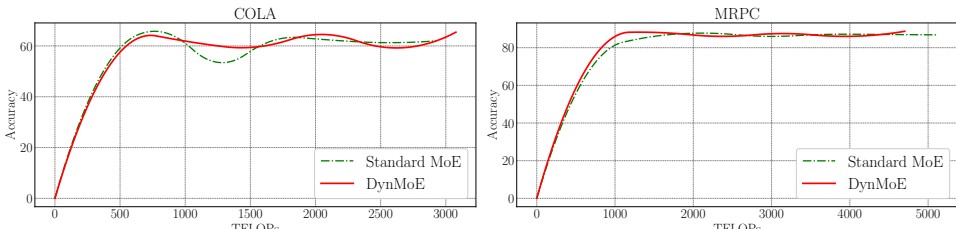

Figure 7: **Convergence curve w.r.t. training FLOPs.** We present the convergence curve with respect to training FLOPs for DYNMOE and the best-performance MoE setting on the GLUE benchmark.

time compared to MoE-LLaVA, indicating improved efficiency. (2) The top-any gating introduces additional cost in the router, but the gating and expert passing steps are more efficient than in MoE-LLaVA. (3) In the current implementation, all experts, loaded or unloaded, occupy GPU memory, resulting in the same memory usage as MoE-LLaVA. Offloading unloaded experts from GPU memory could improve efficiency.

**Training efficiency of DYNMOE.**  We present training FLOPs for both Language (Figure 7) and Vision-Language (Table 24) experiments. The results show that DYNMOE achieves comparable or lower FLOPs than standard MoE, ensuring both efficiency and performance without extensive parameter tuning.

### 4.5  A4: DYNMOE PROVIDE INSIGHTS ON MOE ARCHITECTURE DESIGN

**MoE structure is required for bottom layer rather than top layer.**  In Figures 5 and 6, we present the average top-$k$ of DYNMOE-LLaVA and the frequency of expert activation across various layers. Our observations indicate that: (1) In the top layer (the layer closest to the LM prediction head), tokens tend to select the same expert, while in the bottom layer, tokens activate all experts uniformly. This suggests that there is no need to convert the top layer to MoE layer, whereas the bottom layer should be transformed into MoE layer. (2) Different LLM backbones may exhibit distinct expert activation frequency patterns. For the StableLM backbone, most MoE layers activate only one expert, whereas for the Phi-2 backbone, experts are more likely to be activated uniformly.

**Shared experts exist in each MoE layer.**  Figures 18- 22 display the threshold **G** values for each MoE layer. We notice that typically, one expert per layer has a significantly lower threshold, making it more easier to be activated. This observation is consistent with Deepseek-MoE's (Dai et al., 2024) design of incorporating shared experts for all tokens in each MoE layer.

## 5  CONCLUSION AND FUTURE WORKS

In this paper, we introduce DYNMOE, a method that automatically determines both the number of experts and the number of experts to activate. Our results show that DYNMOE delivers comparable or even superior performance across various MoE model configurations, while maintaining efficiency. This demonstrates DYNMOE's potential to save researchers time and computational resources in hyperparameter tuning. Additionally, our visualizations reveal interesting insights, such as the reduced number of experts needed for the top layers, which could inspire future advancements in MoE model design. For future work, as discussed in Han et al. (2021), MoE can be considered a dynamic model because different tokens may activate different experts, thereby enabling adaptive computation and enhancing the model's ability to adapt to input data. While DYNMOE addresses dynamic challenges through adaptive top-$k$ selection and an adaptive number of experts, exploring integration with other dynamic techniques, such as layer skipping (Zhao et al., 2024), would also be valuable. Moreover, the current adaptive process and top-any gating method are not sufficiently efficient. Developing more optimized implementations, such as designing CUDA kernels, would be valuable in the future.

## ACKNOWLEDGMENTS

This work is supported in part by the National Science and Technology Major Project (No. 2022ZD0115101), Research Center for Industries of the Future (RCIF) at Westlake University, Westlake Education Foundation, and Westlake University Center for High-performance Computing. This work is also supported in part by the funding from Shenzhen Institute of Artificial Intelligence and Robotics for Society, in part by the Shenzhen Key Lab of Crowd Intelligence Empowered Low-Carbon Energy Network (Grant No. ZDSYS20220606100601002), in part by Shenzhen Stability Science Program 2023, and in part by the Guangdong Provincial Key Laboratory of Future Networks of Intelligence (Grant No. 2022B1212010001).

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

## CONTENTS OF APPENDIX

## 6 EXPERIMENT SETTINGS

We conduct experiments on Vision, Language, and Vision-Language tasks. The detailed experiment settings are shown in the following.

- **Vision Task.** For the vision tasks, we follow the same settings as in GMoE (Li et al., 2023a). We employ the pre-trained ViT-S/16 Dosovitskiy et al. (2020) model and evaluate it on the DomainBed (Gulrajani & Lopez-Paz, 2020) benchmark. Our experiments encompass four Domain Generalization datasets: PACS (Li et al., 2017), VLCS (Albuquerque et al., 2019), Office-Home (Venkateswara et al., 2017), and DomainNet (Peng et al., 2019). All results are reported using the train-validation selection criterion. We conduct all experiments on a single RTX 3090 GPU, and the reported results are averaged over three random seeds. For DYNMOE, we set the maximum number of experts to 8 and the initial number of experts to 6. The adaptive process is executed for each iteration.

- **Language Task.** The language tasks adhere to the same settings as those in MoEfication (Zhang et al., 2022) and EMoE (Qiu et al., 2023). The MoE models are built upon the BERT-large (Devlin et al., 2019) architecture using the MoEfication method and are fine-tuned on GLUE (Wang et al., 2018) tasks, which include COLA (Warstadt et al., 2019), QNLI (Wang et al., 2018), RTE (Bentivogli et al., 2009), MNLI (Xu et al., 2020), and MRPC (Dolan & Brockett, 2005). We conduct all experiments on a single RTX 3090 GPU, and the reported results are averaged over three random seeds. For DYNMOE, we set the maximum number of experts to 8 and the initial number of experts to 6. For each epoch, we begin recording routing at 1/3 of the epoch and complete recording routing and execute the adaptive process at 2/3 of the epoch.

- **Vision-Language Task.** The vision-language tasks follows the setting in MoE-LLaVA (Lin et al., 2024), where we use StableLM-2-1.6B (Bellagente et al., 2024), Qwen-1.8B (Bai et al., 2023) and Phi-2 (Hughes) as backbone language models, and use clip-vit-large-patch14-336 (Radford et al., 2021) as the vision encoder. We conduct model training on 8 A100 (80G) GPUs, completing within 2 days, detailed hyper-parameters setting are shown in Table 5. The models are evaluated on image understanding benchmarks including VQA-v2 (Goyal et al., 2017), GQA (Hudson & Manning, 2019), VisWiz (Gurari et al., 2018), ScienceQA-IMG (Lu et al., 2022), TextVQA (Singh et al., 2019), POPE (Li et al., 2023c), MME (Yin et al., 2023), MMBench (Liu et al., 2023), LLaVA-Bench (in-the-Wild) (Liu et al., 2024), and MM-Vet (Yu et al., 2023). Furthermore, we keep routing records in our model during testing time. For each benchmark, we collect the number of experts' activations per MoE layer and total processed tokens during testing.

Table 5: **Detailed training hyper-parameters and configuration.**

| Config | Models | | |
|---|---|---|---|
| | StableLM | Qwen | Phi-2 |
| Maximum experts | | 4 | |
| Deepspeed | Zero2 | Zero2 | Zero2_offload |
| Data | | LLaVA-Finetuning | |
| Image resolution | | $336 \times 336$ | |
| Image encoder | | CLIP-Large/336 | |
| Feature select layer | | -2 | |
| Image projector | | Linear layers with GeLU | |
| Epoch | | 1 | |
| Learning rate | | 2e-5 | |
| Learning rate schedule | | Cosine | |
| Weight decay | | 0.0 | |
| Batch size per GPU | 8 | 8 | 4 |
| GPU | $4 \times$ A100 (80G) | $8 \times$ A100 (80G) | $8 \times$ A100 (80G) |
| Precision | | Bf16 | |

---

**Algorithm 1** Pseudo code of DYNMOE on each iteration and MoE layer.

---

**Require:** Input data $\mathbf{x}$, initial gating network parameters $\mathbf{W}_g$, $\mathbf{G}$, and $\tau$, experts $E_1, \cdots, E_K$, start record routing flag $flag_s$, finish record routing flag $flag_f$.
**Ensure:** MoE layer output $\mathbf{y}$, auxiliary loss value.
1: **if** $flag_s$ **then**
2:     Set routing flag $flag_{rout} = 1$.
3:     Initialize routing records by $\mathbf{R}_{\text{rout}} = \mathbf{0}_K$.
4:     Initialize non-activate sample records $\mathbf{R}_{\text{sam}} = \mathbf{0}_d$.
5: Get the gating outputs $g(\mathbf{x})$ and $\mathbf{k}$ by Eq (4) and (5).
6: Get MoE layer output $\mathbf{y}$ by Eq (6).
7: Calculate auxiliary loss by Eq (8).
8: **if** $flag_{rout} = 1$ **then**
9:     $\mathbf{R}_E = \mathbf{R}_E + \text{sum}(g(\mathbf{x}), \text{dim} = 0)$.
10:     $\mathbf{R}_S = \mathbf{R}_S + \sum_{i=1}^{N} \mathbf{1}_{\mathbf{k}_i=0} \mathbf{x}_i$
11: **if** $flag_f$ **then**
12:     $flag_{rout} = 0$.
13:     **if** Exists $e$ that $\mathbf{R}_E^e = \mathbf{0}$ **then**
14:        Remove experts $e$.
15:     **if** $\mathbf{R}_{S,e} \neq \mathbf{0}$ **then**
16:        Add new expert $K + 1$ with expert representation $\mathbf{W}_{g,K+1} = \mathbf{R}_S / \|\mathbf{R}_S\|$.

---

# 7 DETAILED ALGORITHM FRAMEWORK

# 8 ADDITIONAL EXPERIMENTS

## 8.1 DETAILED RESULTS ON LANGUAGE AND VISION TASKS

In this section, we present the detailed results of our experiments on the GLUE benchmark (Wang et al., 2018) in Table 11 and on the DomainNet dataset in Table 12. These results demonstrate that incorporating the specially designed diversity and simplicity loss significantly enhances the model's performance.

Moreover, we present the detailed results using different learning rates on the GLUE benchmark in Tables 6- 10.

## 8.2 COMBINE DYNMOE WITH LOAD BALANCE AND EFFICIENCY LOSSES

In Tables 13, 14, 15, and 16, we report the following metrics:
- Performance (Table 13): The performance of different settings.

Table 6: **Detailed performance of DYNMOE and various MoE settings on COLA dataset**

| COLA | $K=8,k=1$ | $K=8,k=2$ | $K=8,k=4$ | $K=8,k=8$ | $K=16,k=1$ | $K=16,k=2$ | $K=16,k=4$ | $K=16,k=8$ | DynMoE |
|---|---|---|---|---|---|---|---|---|---|
| lr = 2e-5 | 64.10 | 64.51 | 64.94 | 43.00 | 63.63 | 64.71 | 64.12 | 64.37 | 65.17 |
| lr = 3e-5 | 63.86 | 62.10 | 64.73 | 64.03 | 61.76 | 22.04 | 63.42 | 63.13 | 62.80 |
| lr = 5e-5 | 41.83 | 39.68 | 62.63 | 0.00 (fail) | 37.26 | 38.30 | 20.24 | 25.79 | 40.68 |

Table 7: **Detailed performance of DYNMOE and various MoE settings on MRPC dataset**

| MPRC | $K=8,k=1$ | $K=8,k=2$ | $K=8,k=4$ | $K=8,k=8$ | $K=16,k=1$ | $K=16,k=2$ | $K=16,k=4$ | $K=16,k=8$ | DynMoE |
|---|---|---|---|---|---|---|---|---|---|
| lr = 2e-5 | 89.74 | 89.63 | 89.74 | 89.36 | 88.07 | 89.02 | 89.74 | 89.56 | 89.57 |
| lr = 3e-5 | 90.14 | 90.19 | 89.50 | 88.67 | 89.81 | 90.18 | 89.38 | 90.35 | 90.64 |
| lr = 5e-5 | 88.70 | 84.62 | 88.72 | 84.48 | 88.30 | 89.08 | 87.40 | 79.95 | 90.09 |

Table 8: **Detailed performance of DYNMOE and various MoE settings on QNLI dataset**

| QNLI | $K=8,k=1$ | $K=8,k=2$ | $K=8,k=4$ | $K=8,k=8$ | $K=16,k=1$ | $K=16,k=2$ | $K=16,k=4$ | $K=16,k=8$ | DynMoE |
|---|---|---|---|---|---|---|---|---|---|
| lr = 2e-5 | 92.48 | 84.94 | 92.52 | 92.46 | 92.39 | 92.51 | 92.65 | 92.49 | 92.39 |
| lr = 3e-5 | 92.45 | 92.39 | 92.01 | 78.39 | 78.22 | 92.53 | 92.50 | 92.31 | 92.59 |
| lr = 5e-5 | 50.54 | 64.46 | 78.13 | 64.43 | 50.54 | 50.54 | 64.27 | 64.43 | 75.50 |

Table 9: **Detailed performance of DYNMOE and various MoE settings on MNLI dataset**

| MNLI | $K=8,k=1$ | $K=8,k=2$ | $K=8,k=4$ | $K=8,k=8$ | $K=16,k=1$ | $K=16,k=2$ | $K=16,k=4$ | $K=16,k=8$ | DynMoE |
|---|---|---|---|---|---|---|---|---|---|
| lr = 2e-5 | 86.56 | 86.70 | 86.57 | 86.61 | 86.63 | 86.73 | 86.55 | 86.51 | 86.37 |
| lr = 3e-5 | 86.46 | 52.40 | 69.40 | 69.35 | 69.57 | 68.47 | 86.59 | 69.47 | 52.34 |
| lr = 5e-5 | 51.44 | 35.45 | 35.45 | 35.45 | 35.45 | 34.54 | 35.45 | 34.24 | 51.68 |

Table 10: **Detailed performance of DYNMOE and various MoE settings on RTE dataset**

| RTE | $K=8,k=1$ | $K=8,k=2$ | $K=8,k=4$ | $K=8,k=8$ | $K=16,k=1$ | $K=16,k=2$ | $K=16,k=4$ | $K=16,k=8$ | DynMoE |
|---|---|---|---|---|---|---|---|---|---|
| lr = 2e-5 | 73.04 | 70.52 | 74.13 | 74.37 | 74.01 | 66.19 | 75.33 | 72.56 | 72.80 |
| lr = 3e-5 | 72.44 | 74.85 | 75.09 | 73.53 | 73.16 | 72.32 | 75.21 | 73.53 | 73.41 |
| lr = 5e-5 | 58.48 | 54.39 | 62.45 | 65.10 | 63.78 | 63.06 | 58.84 | 63.66 | 65.22 |

Table 11: **Performance of DYNMOE on language tasks**: Our study investigates the performance of DYNMOE on language tasks using the GLUE (Wang et al., 2018) benchmark, with BERT-large serving as the backbone model. The baselines including traditional MoE methods with different number of experts $K$ and top-$k$. In our implementation of DYNMOE, we configure the maximum number of experts to 16, with an initial setting of 8 experts. The number of experts is dynamically adjusted in each epoch for DYNMOE. The − represents experiment failure, final results could not be obtained using Gshard loss.

| Algorithms | COLA | MRPC | QNLI | MNLI | RTE | Average |
|---|---|---|---|---|---|---|
| MoE ($K=8,k=1$) | 64.10±0.94 | 90.14±0.60 | 92.48±0.21 | 86.56±0.06 | 73.04±2.13 | 81.26 |
| MoE ($K=8,k=2$) | 64.51±0.81 | 90.19±0.17 | 92.39±0.08 | 86.70±0.23 | 74.85±1.96 | 81.73 |
| MoE ($K=8,k=4$) | 64.94±0.62 | 89.74±0.99 | 92.52±0.12 | 86.57±0.28 | 75.09±1.84 | 81.77 |
| MoE ($K=8,k=8$) | 64.03±0.54 | 89.36±0.09 | 92.46±0.09 | 86.61±0.26 | 74.37±0.78 | 81.37 |
| MoE ($K=16,k=1$) | 63.63±0.20 | 89.81±0.30 | 92.39±0.21 | 86.63±0.17 | 74.01±0.29 | 81.29 |
| MoE ($K=16,k=2$) | 64.71±1.21 | 90.18±1.33 | 92.53±0.07 | 86.73±0.43 | 72.32±3.54 | 81.29 |
| MoE ($K=16,k=4$) | 64.12±1.42 | 89.74±0.40 | 92.65±0.09 | 86.59±0.16 | 75.33±0.95 | 81.69 |
| MoE ($K=16,k=8$) | 64.37±1.14 | 90.35±0.68 | 92.49±0.11 | 86.51±0.20 | 73.53±2.21 | 81.45 |
| DYNMOE, Gshard Loss | 64.88±0.86 | 89.85±0.22 | 92.42±0.07 | - | 73.41±0.68 | - |
| DYNMOE | 65.17±0.26 | 90.64±0.26 | 92.59±0.08 | 86.37±0.13 | 73.41±1.96 | 81.64 |

Table 12: **Detailed results on DomainNet dataset**: We report the detailed test results on each domain of the DomainNet dataset.

| Algorithms | clip | info | paint | quick | real | sketch | Average |
|---|---|---|---|---|---|---|---|
| GMoE (with DYNMOE, Gshard Loss) | 66.8 | 23.8 | 54.1 | 15.9 | 68.7 | 54.9 | 47.4 |
| GMoE (with DYNMOE, Diverse and Simple Gating Loss) | 68.0 | 24.4 | 55.4 | 16.6 | 69.5 | 55.1 | 48.2 |

- Load Balance (Table 14): The frequency with which each expert is activated, calculated as (expert activation time / total token count).
- Efficiency (Table 15 and 16): The top-k values per layer and the top-k activation frequency, calculated as (number of tokens that activate k experts / total tokens).

We can find that

Table 13: **Performance of DYNMOE with load balance and efficiency losses.** We conduct experiments using the MoE-LLaVA setup, incorporating (1) a load-balancing loss and (2) an efficiency loss to enforce sparsity, as proposed by Zheng et al. (2025).

| Performance (StableLM) | VQAv2 | GQA | VizWiz | SQA | TextVQA | POPE | MME | MMBench |
|---|---|---|---|---|---|---|---|---|
| DYNMOE | 77.4 | 61.4 | 40.6 | 63.4 | 48.9 | 85.7 | 1300.9 | 63.2 |
| DYNMOE + load balance | 77.1 | 61.6 | 37.0 | 61.4 | 50.3 | 85.3 | 1313.5 | 61.7 |
| DYNMOE + load balance + efficiency | 77.1 | 61.8 | 39.4 | 62.9 | 49.7 | 85.4 | 1321.2 | 61.9 |

Table 14: **Activation frequency per expert of DYNMOE with load balance and efficiency losses.** We conduct experiments using the MoE-LLaVA setup, incorporating (1) a load-balancing loss and (2) an efficiency loss to enforce sparsity, as proposed by Zheng et al. (2025). We report the activation frequency of each expert at layer 0 on the VQAv2 dataset.

| Activation Frequency per Expert (VQAv2, layer 0) | Expert 1 | Expert 2 | Expert 3 | Expert 4 |
|---|---|---|---|---|
| MoE (top-2) | 0.36 | 1.29 | 0.16 | 0.19 |
| DYNMOE | 0.29 | 0.97 | 0.48 | 0.35 |
| DYNMOE + load balance | 0.81 | 0.50 | 0.90 | 0.68 |
| DYNMOE + load balance + efficiency | 0.45 | 0.52 | 0.42 | 0.63 |

Table 15: **Sparsity of DYNMOE with load balance and efficiency losses.** We conduct experiments using the MoE-LLaVA setup, incorporating (1) a load-balancing loss and (2) an efficiency loss to enforce sparsity, as proposed by Zheng et al. (2025). We report the average top-k value of each expert at each layer on the VQAv2 dataset.

| Top-k per Layer (VQAv2) | Layer 0 | Layer 2 | Layer 4 | Layer 6 | Layer 8 | Layer 10 | Layer 12 | Layer 14 | Layer 16 | Layer 18 | Layer 20 | Layer 22 |
|---|---|---|---|---|---|---|---|---|---|---|---|---|
| DYNMOE | 2.09 | 1.07 | 1.57 | 1.06 | 2.04 | 1.03 | 1.03 | 1.00 | 1.03 | 1.02 | 1.02 | 1.00 |
| DYNMOE + load balance | 2.88 | 1.25 | 1.59 | 1.27 | 1.26 | 1.13 | 1.77 | 1.70 | 1.12 | 1.33 | 1.30 | 1.00 |
| DYNMOE + load balance + efficiency | 2.02 | 1.25 | 1.81 | 1.57 | 1.65 | 1.20 | 1.47 | 2.30 | 1.07 | 1.37 | 1.82 | 1.00 |

Table 16: **Top-k frequency of DYNMOE with load balance and efficiency losses.** We conduct experiments using the MoE-LLaVA setup, incorporating (1) a load-balancing loss and (2) an efficiency loss to enforce sparsity, as proposed by Zheng et al. (2025). We report the frequency of activating top-k experts for each configuration.

| Top-k Frequency (VQAv2) | Top-1 | Top-2 | Top-3 | Top-4 |
|---|---|---|---|---|
| DYNMOE | 0.79 | 0.16 | 0.04 | 0.01 |
| DYNMOE + load balance | 0.65 | 0.26 | 0.06 | 0.03 |
| DYNMOE + load balance + efficiency | 0.58 | 0.32 | 0.09 | 0.01 |

- Although it does not explicitly enforce load balancing, the original DynMoE achieves load balancing comparable to that of the standard top-2 MoE (Table 14).
- Adding the load balance loss slightly decreases the performance of DynMoE (Table 13) while increasing the number of activated experts (Table 15). However, it improves load balancing (Table 14).
- Adding an additional efficiency loss on top of the load balance loss improves performance (Table 13) and helps overcome some extreme cases, such as the reduction of the top-k values in the bottom layer from 2.88 to 2.02 (Table 15), and reduce the number of tokens that activate all 4 experts (Table 16). Moreover, the efficiency loss further enhances load balancing (Table 14).

## 8.3 COMPARISION TO TOP-P GATING BASELINE

In Table 17, we compare DYNMOE with the method proposed by Huang et al. (2024), which replaces the traditional top-k gating with top-p gating. We set $p = 0.4$ as suggested in the original paper. Our results show that DynMoE achieves better performance without requiring the additional parameter.

## 8.4 NUMBERICAL RESULTS ON MORE VISION TASKS

In Tables 18, 19, and 20, we show the performance of DYNMOE on more vision tasks, and also investigate the impact of strategies on initialize the new experts, including

- **Average:** Averaging the parameters of existing experts to initialize the new expert.

Table 17: **Comparision to top-p gating method.** We conduct experiments using the MoE-LLaVA setup, and compare DYNMOE to top-p gating method (Huang et al., 2024).

| StableLM | VQAv2 | GQA | VizWiz | SQA | TextVQA | POPE | MME | MMBench |
|---|---|---|---|---|---|---|---|---|
| DYNMOE | 77.4 | 61.4 | 40.6 | 63.4 | 48.9 | 85.7 | 1300.9 | 63.2 |
| top p (p=0.4) | 77.1 | 61.7 | 36.0 | 62.8 | 48.6 | 85.2 | 1332.9 | 62.3 |

Table 18: **Fintuneing results on Tiny-ImageNet dataset.** We fine-tune the pretrained ViT-S (Dosovitskiy et al., 2020) model on the TinyImageNet (Le & Yang, 2015) dataset and report the accuracy every two epochs.

| TinyImageNet (Finetune, ViT-S, 2 MoE layers) | E1 | E3 | E5 | E7 | E9 | E11 | E13 | E15 | E17 | E19 | E20 |
|---|---|---|---|---|---|---|---|---|---|---|---|
| MoE (K = 8, k = 1) | 78.32 | 82.79 | 84.03 | 84.83 | 85.20 | 85.61 | 85.82 | 86.27 | 86.44 | 86.61 | 86.65 |
| MoE (K = 8, k = 2) | 78.53 | 82.95 | 84.05 | 84.74 | 84.99 | 85.00 | 85.95 | 86.45 | 86.63 | 86.58 | 86.72 |
| MoE (K = 8, k = 4) | 79.25 | 83.38 | 83.73 | 84.72 | 85.00 | 85.50 | 85.93 | 86.27 | 86.00 | 86.64 | 86.56 |
| MoE (K = 8, k = 8) | 79.20 | 83.30 | 84.02 | 84.10 | 84.86 | 85.62 | 86.08 | 86.12 | 86.44 | 86.73 | 86.58 |
| DynMoE (Original, avg topk=6.5) | 79.10 | 83.09 | 84.20 | 84.84 | 85.18 | 85.56 | 85.91 | 86.09 | 86.37 | 86.40 | 86.70 |
| DynMoE (Average, avg topk=6.0) | 79.19 | 83.48 | 84.21 | 84.84 | 85.32 | 85.76 | 86.25 | 86.41 | 86.49 | 86.70 | 86.75 |
| DynMoE (W-Average, avg topk=6.0) | 78.96 | 83.18 | 84.15 | 84.92 | 85.34 | 85.93 | 86.10 | 86.30 | 86.60 | 86.70 | 86.80 |
| DynMoE (Most activated, avg topk=6.5) | 79.09 | 83.57 | 84.21 | 84.62 | 85.40 | 85.87 | 86.45 | 86.40 | 86.63 | 86.66 | 86.70 |

- **W-Average:** Using weighted averaging of the parameters of existing experts, where the weights correspond to the number of experts to be activated.

- **Most activated:** Initializing the new expert using the parameters of the most frequently activated expert.

Results show that (1) DynMoE converges faster than standard MoE settings; (2) W-Average achieves the best performance in most cases; and (3) incorporating load balance loss and efficiency loss accelerates training while improving performance.

## 8.5 EFFICIENCY EVALUATION

In Table 21, we compare the training efficiency of DYNMOE with standard top-1 and top-2 gating MoE models. The results show that DYNMOE trains faster than top-2 gating but slower than top-1 gating.

In Table 22, we show the overhead introduced by the top-any gating method. The results indicate that the primary source of overhead comes from the routing process, which likely explains why DynMoE (with forced top-1) is noticeably slower than MoE (also with forced top-1).

## 8.6 ABLATION STUDIES ON AGGREGATION WEIGHTS

In Remark 3.1, we discussed why we did not consider the magnitude of scores when averaging the expert outputs. In Table 23, we show that using different expert scores significantly reduce the model performance.

## 9 ADDITIONAL VISUALIZATION RESULTS

### 9.1 ACTIVATION FREQUENCY

We present the activation frequency of experts across various MoE layers and evaluation tasks using different backbones: StableLM-1.6B (Figures 9 and 10), Qwen-1.8B (Figures 11 and 12), and Phi-2-2.7B (Figures 13 and 14). The results suggest that compared to the StableLM-1.6B backbone, experts are more uniformly activated for models utilizing Qwen-1.8B and Phi-2-2.7B as backbone LLMs.

Table 19: **Pretrain results in CIFAR-10 dataset.** We pretrain the ViT-S (Dosovitskiy et al., 2020) model on the CIFAR-10 dataset and report the accuracy every 100 epochs.

| CIFAR10 (ViT-S, 2 MoE Layer, Acc per 100 Epoch) | 200 | 300 | 400 | 500 |
|---|---|---|---|---|
| MoE (K = 8, k = 1) | 72.66 | 77.51 | 80.10 | 81.08 |
| MoE (K = 8, k = 2) | 73.79 | 78.50 | 80.85 | 81.91 |
| MoE (K = 8, k = 4) | 72.77 | 77.84 | 80.30 | 81.14 |
| MoE (K = 8, k = 8) | 70.68 | 75.32 | 78.28 | 79.11 |
| DynMoE (Original, avg topk=7) | 74.84 | 79.24 | 81.77 | 82.50 |
| DynMoE (Average, avg topk=7) | 74.70 | 80.32 | 82.51 | 83.57 |
| DynMoE (W-Average, avg topk=6.5) | 74.16 | 78.77 | 81.30 | 82.01 |
| DynMoE (Most activated, avg topk=6.5) | 71.71 | 77.56 | 80.08 | 80.58 |

Table 20: **Pretrain results on ImageNet dataset.** We train ViT-S from scratch for 200 epochs on ImageNet dataset (Deng et al., 2009) with a batch size of 512 across 8 A100 GPUs. In the ViT-S architecture, the layers [0, 3, 6, 9] are replaced with MoE layers, and the maximum number of experts per layer is 4. The learning rate is set to 1e-4, and we use the Adam optimizer with parameters [0.9, 0.99] and cosine learning schedule, while the weight decay is set to 5e-5. During evaluation, we set the batch size to 128 and use 1 A100 GPU. DynMoE-A indicates DynMoE with a simple and diverse gating loss, and DynMoE-B indicates DynMoE with simple and diverse gating loss + load balance loss + efficiency loss.

| | Train Time (s/batch) | Train Wall-Clock Time (days) | Evaluation Time (s/batch) | Inference Time on Routing (ms / batch and MoE layer) | Inference Time on Gating (ms / batch and MoE layer) | Inference Time on Expert Passing (ms / batch and MoE layer) | Acc@1 (%) |
|---|---|---|---|---|---|---|---|
| DeepSpeed-MoE (top-1) | 0.39 | 2.3 | 0.53 | 0.06 | 14.54 | 94.32 | 64.7 |
| DeepSpeed-MoE (top-2) | 0.63 | 3.7 | 1.05 | 0.06 | 50.78 | 186.47 | 67.3 |
| DynMoE-A (k=1.63) | 0.51 | 2.8 | 0.99 | 0.53 | 36.45 | 186.18 | 66.5 |
| DynMoE-B (k=1.25) | 0.41 | 2.5 | 0.82 | 0.46 | 31.04 | 148.11 | 68.5 |

Table 21: **Training efficiency comparision.**

| Time (s/batch) | MoE-LLaVA (Train, 4 A100, batch size=32) | ViT-S, ImageNet (Train, 8 A100, batch size=512) | ViT-S, ImageNet (Test, 4 A100, batch size=256) |
|---|---|---|---|
| Top-1 MoE | 1.31 | 0.39 | 0.18 |
| Top-2 MoE | 1.60 | 0.63 | 0.32 |
| DynMoE | 1.48 | 0.51 | 0.27 |

Table 22: **Overhead of top-any gating.** We report the time taken for key steps in top-any routing, including routing, gating, and expert passing.

| Time (per sample and MoE layer) | Routing (ms) | Gating (ms) | Expert Passing (ms) |
|---|---|---|---|
| MoE-LLaVA (enforced top-1) | 0.04 | 0.60 | 1.08 |
| DynMoE (enforced top-1) | 0.45 | 0.75 | 1.10 |

Table 23: **Ablation studies on aggregation weights.** We conduct experiments using the MoE-LLaVA setup, and the "weighted scores" indicates averaging the expert outputs based on the gating outputs.

| StableLM | VQAv2 | GQA | VizWiz | SQA | TextVQA | POPE | MME | MMBench |
|---|---|---|---|---|---|---|---|---|
| ours | 77.4 | 61.4 | 40.6 | 63.4 | 48.9 | 85.7 | 1300.9 | 63.2 |
| weighted scores | 73.9 | 57.4 | 32.1 | 61.3 | 46.9 | 84.2 | 1176.8 | 52.1 |

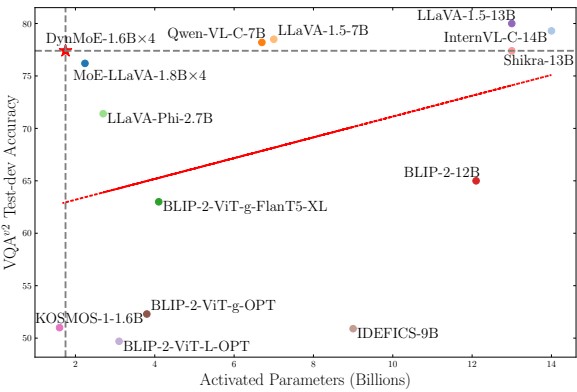

Figure 8: **Comparing the performance efficiency of models.** The $x$-axis represents the number of activated parameters, while the $y$-axis shows the performance on the Visual Question Answering (VQA) task.

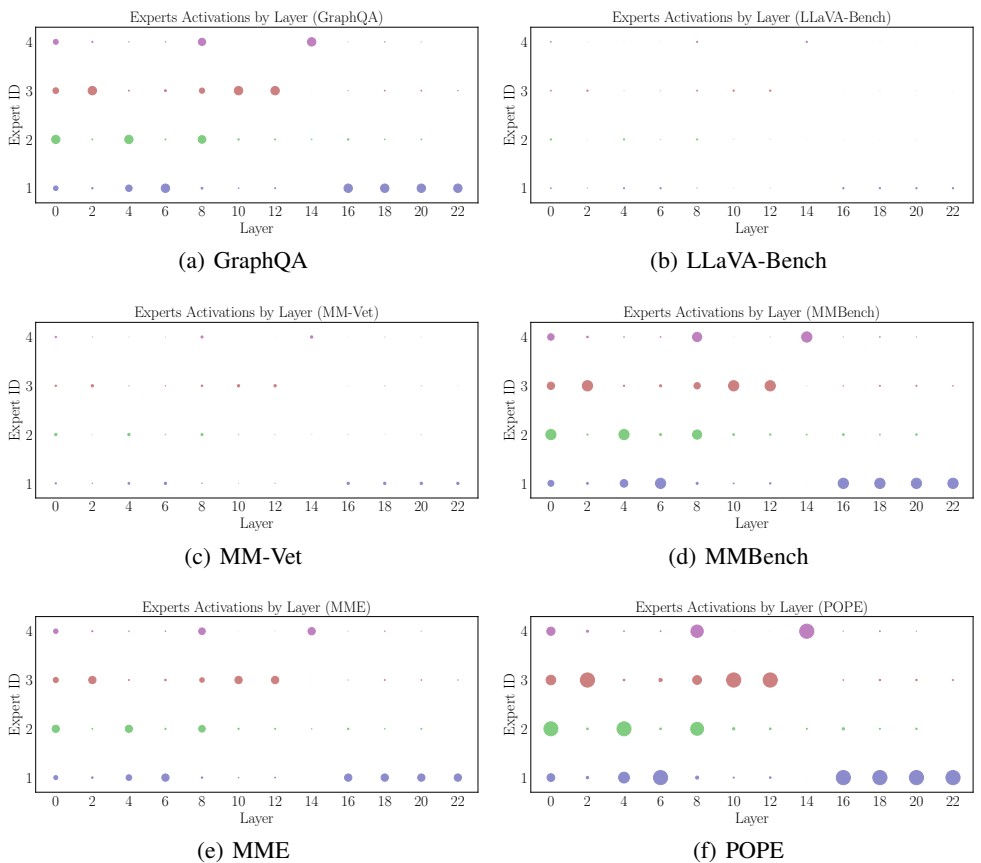

Figure 9: **Activation frequency of experts on various MoE layers and evaluation tasks using StableLM as backbone.**

## 9.2 AVERAGE TOP-$k$

In Figures 15 and 16 , we illustrate the average top-$k$ of DYNMOE models using Qwen and Phi-2 as backbone LLMs.

## 9.3 LAYER-WISE EXPERT SIMILARITY MATRIX

In Figures 17, 19, and 21, we illustrate the similarities between various expert representations, specifically, different rows of $\mathbf{W}_g$ across multiple MoE layers. These comparisons utilize StableLM-

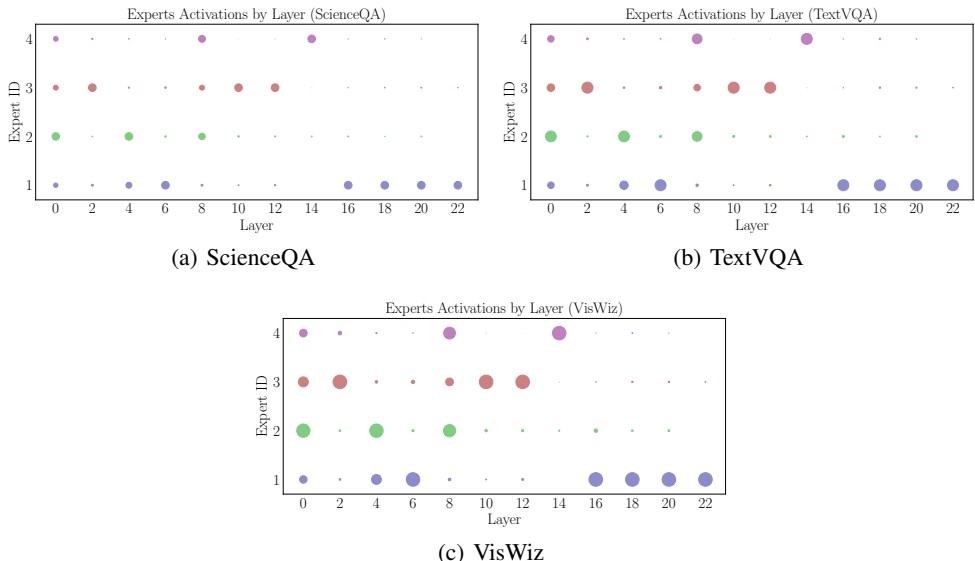

Figure 10: **Activation frequency of experts on various MoE layers and evaluation tasks using StableLM as backbone.**

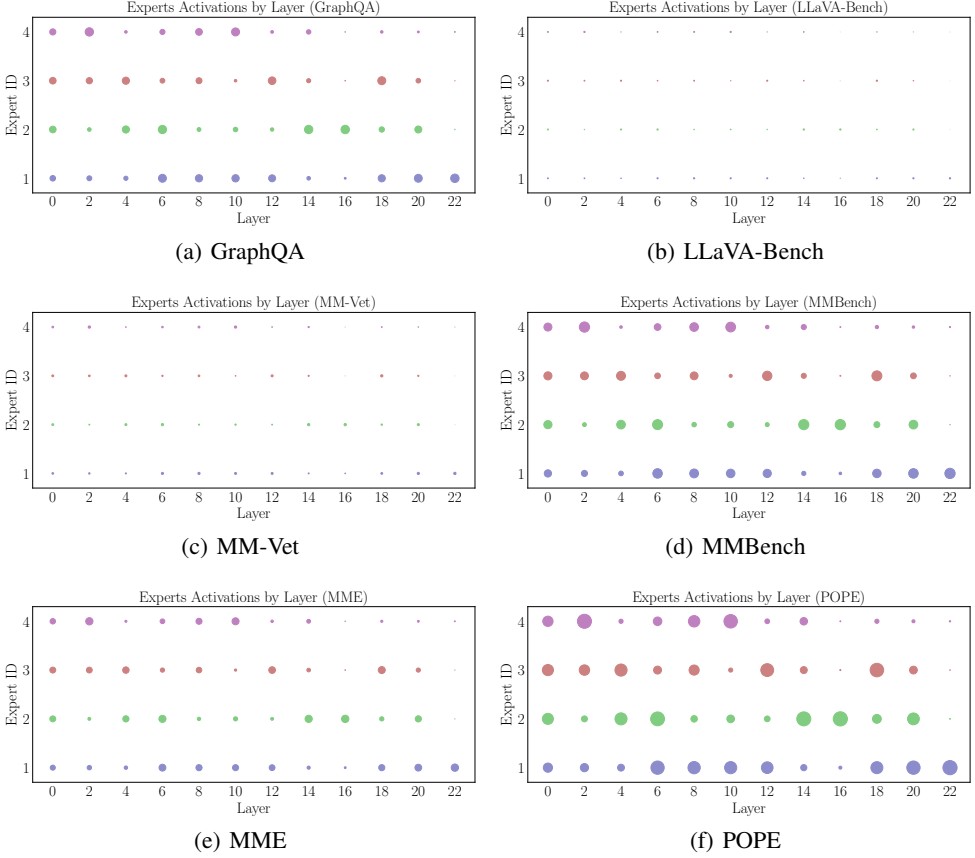

Figure 11: **Activation frequency of experts on various MoE layers and evaluation tasks using Qwen as backbone.**

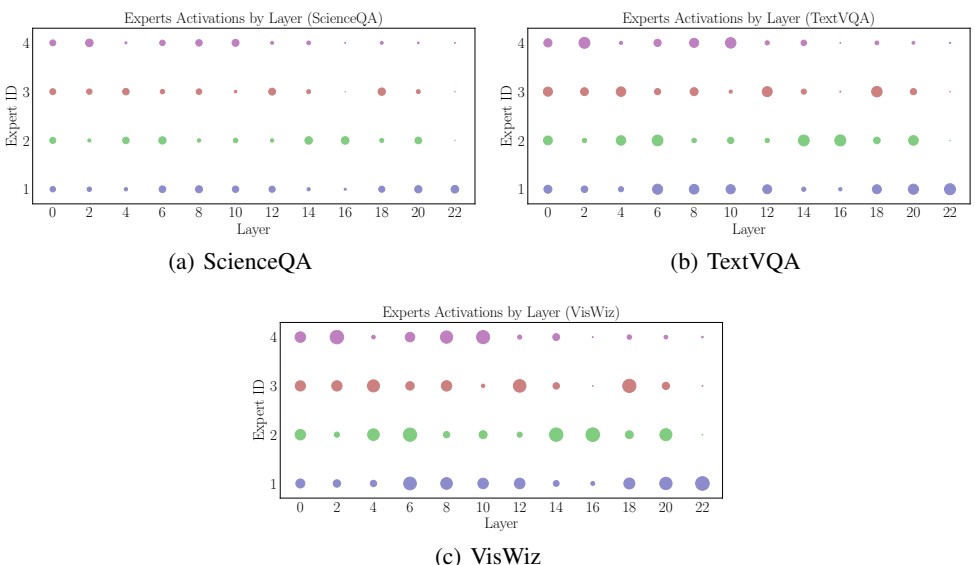

Figure 12: **Activation frequency of experts on various MoE layers and evaluation tasks using Qwen as backbone.**

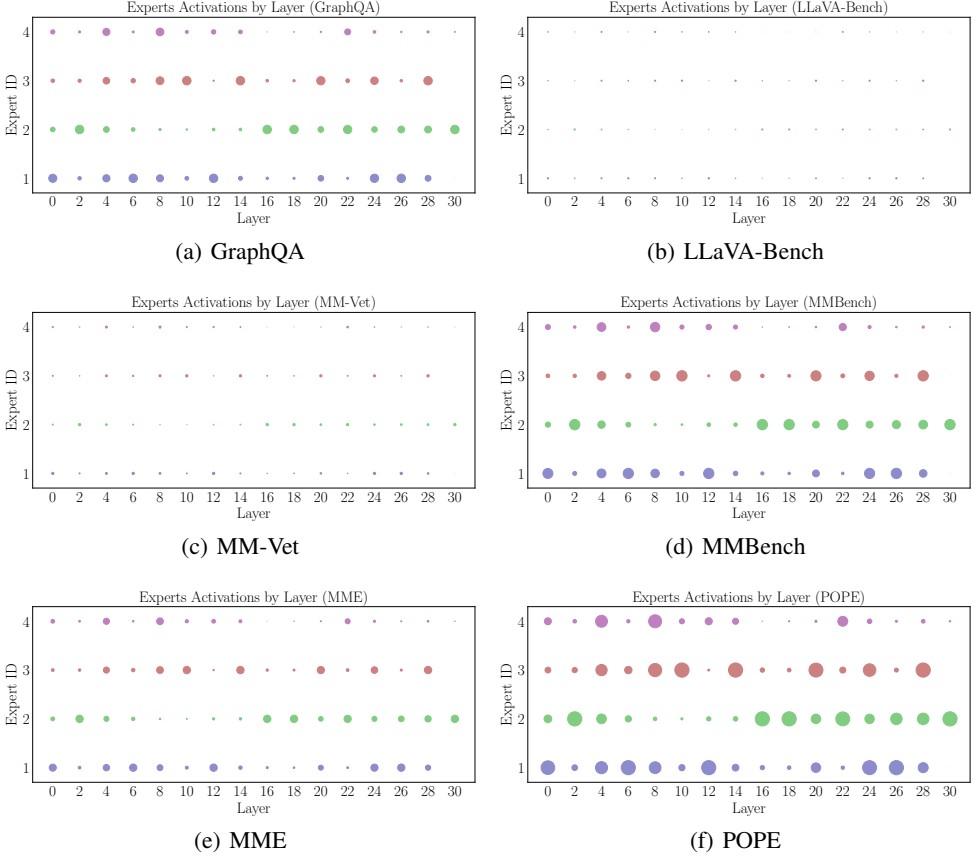

Figure 13: **Activation frequency of experts on various MoE layers and evaluation tasks using Phi-2 as backbone.**

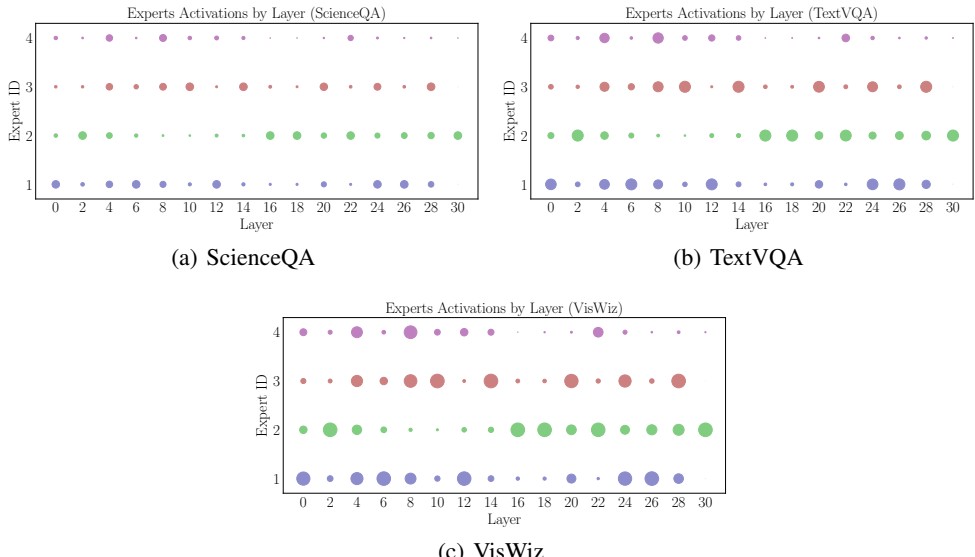

Figure 14: **Activation frequency of experts on various MoE layers and evaluation tasks using Phi-2 as backbone.**

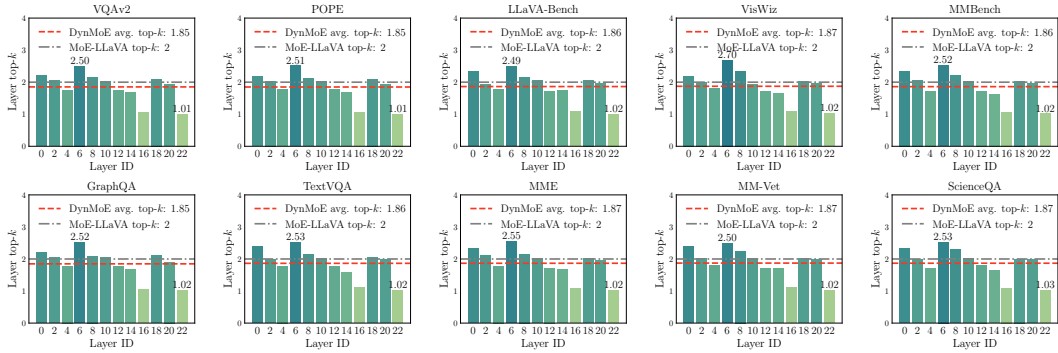

Figure 15: **Average top-$k$ activated experts of DYNMOE on vision-language benchmarks, using Qwen as language backbone.**

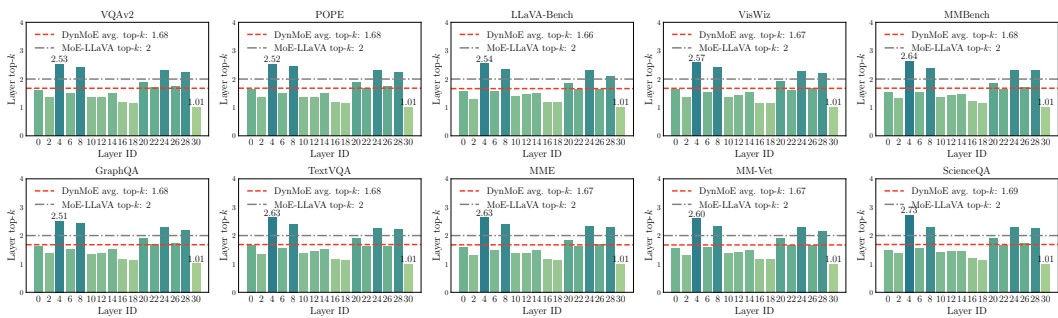

Figure 16: **Average top-$k$ activated experts of DYNMOE on vision-language benchmarks, using Phi-2 as language backbone.**

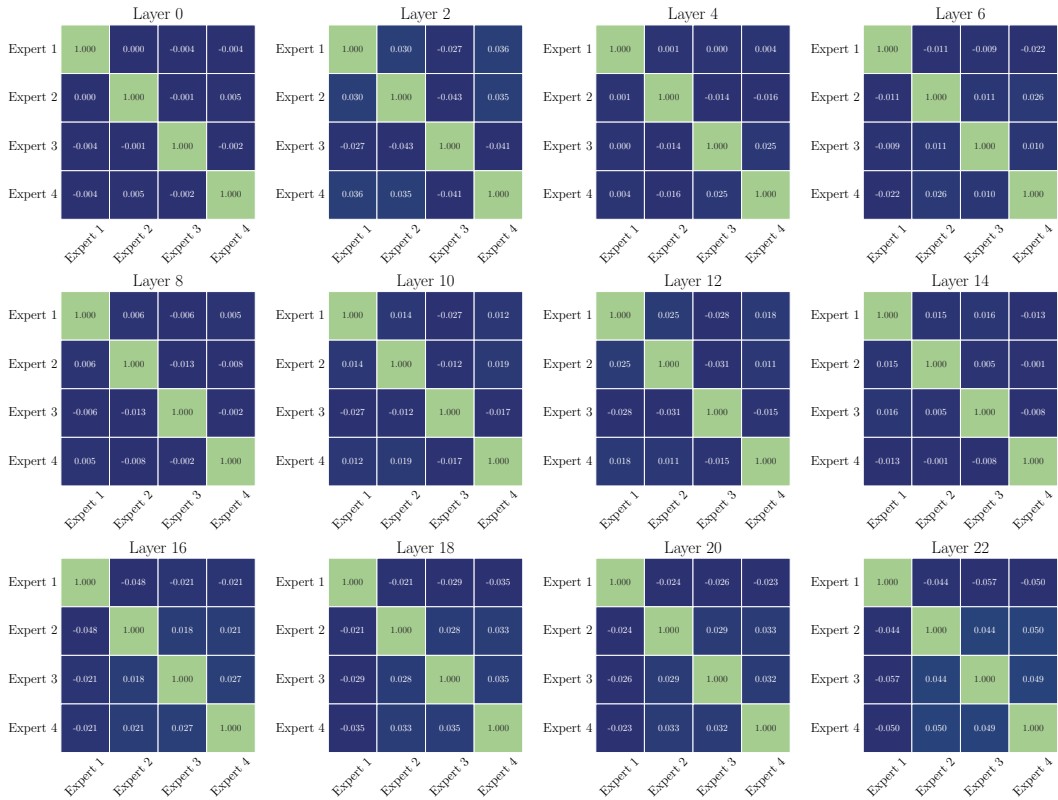

Figure 17: **Layer-wise expert similarity matrix (StableLM).** We record the experts' cosine similarity per layer during test time. It turns out the cosine similarity between experts is close to 0.

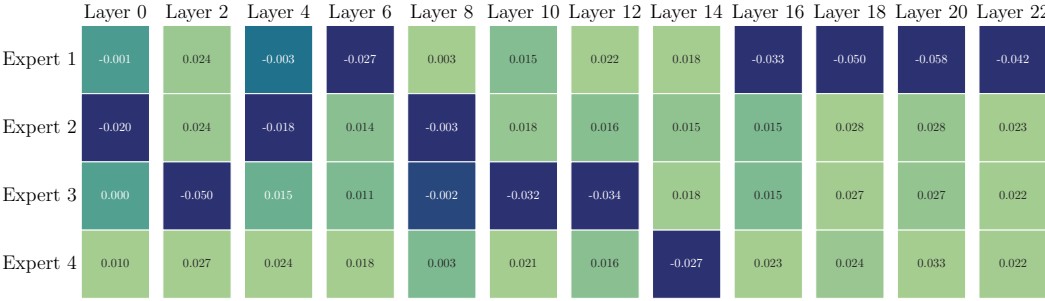

Figure 18: **Layer-wise expert activation threshold (StableLM).** Darker-colored experts are more likely to be activated compared to lighter-colored experts.

1.6B, Qwen-1.8B, and Phi-2-2.7B as the backbone LLMs. The findings demonstrate that these expert representations are nearly orthogonal, suggesting that different experts capture diverse features, which could potentially enhance the model's capacity.

## 9.4 VISUALIZATION OF **G**

In Figures 18, 20, and 22, we present the values of the learned threshold **G**, employing StableLM-1.6B, Qwen-1.8B, and Phi-2-2.7B as the backbone LLMs. The results reveal that for each MoE layer, there is one expert that is more readily activated. This observation is consistent with the design of Deepseek-MoE (Dai et al., 2024).

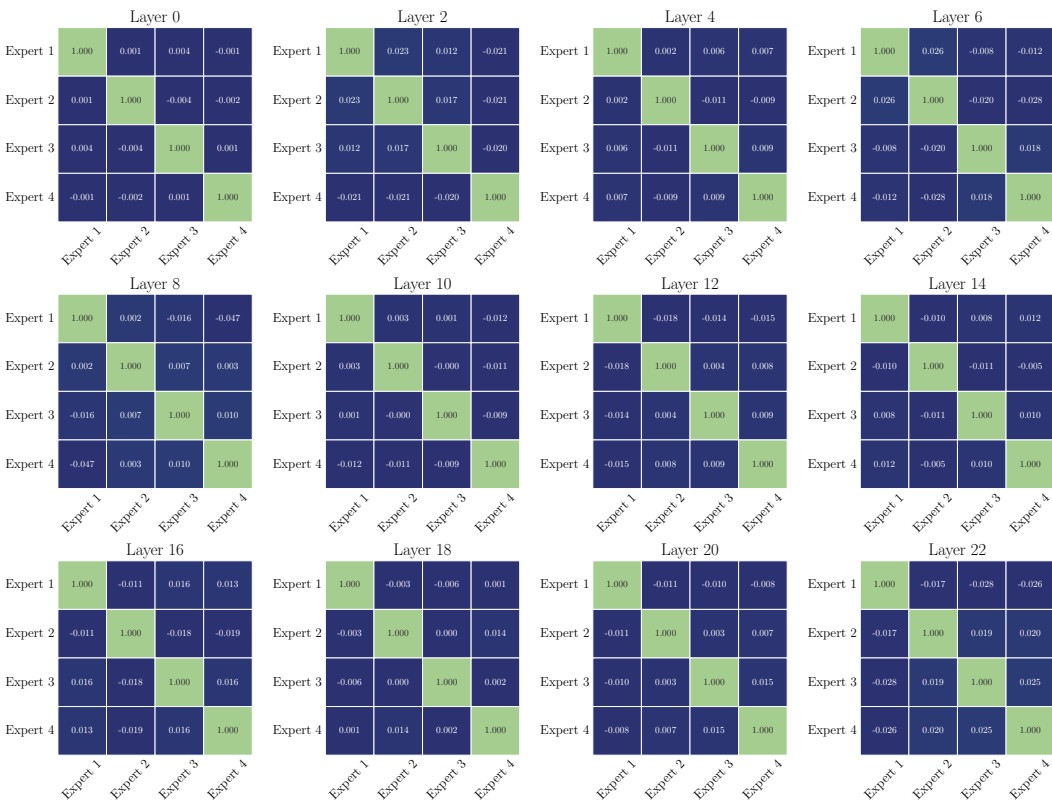

Figure 19: **Layer-wise expert similarity matrix (Qwen).** We record the experts' cosine similarity per layer during test time. It turns out the cosine similarity between experts is close to 0.

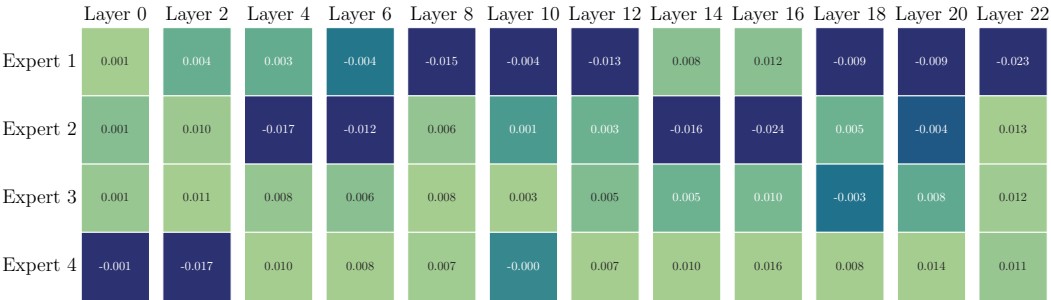

Figure 20: **Layer-wise expert activation threshold (Qwen).** Darker-colored experts are more likely to be activated compared to lighter-colored experts.

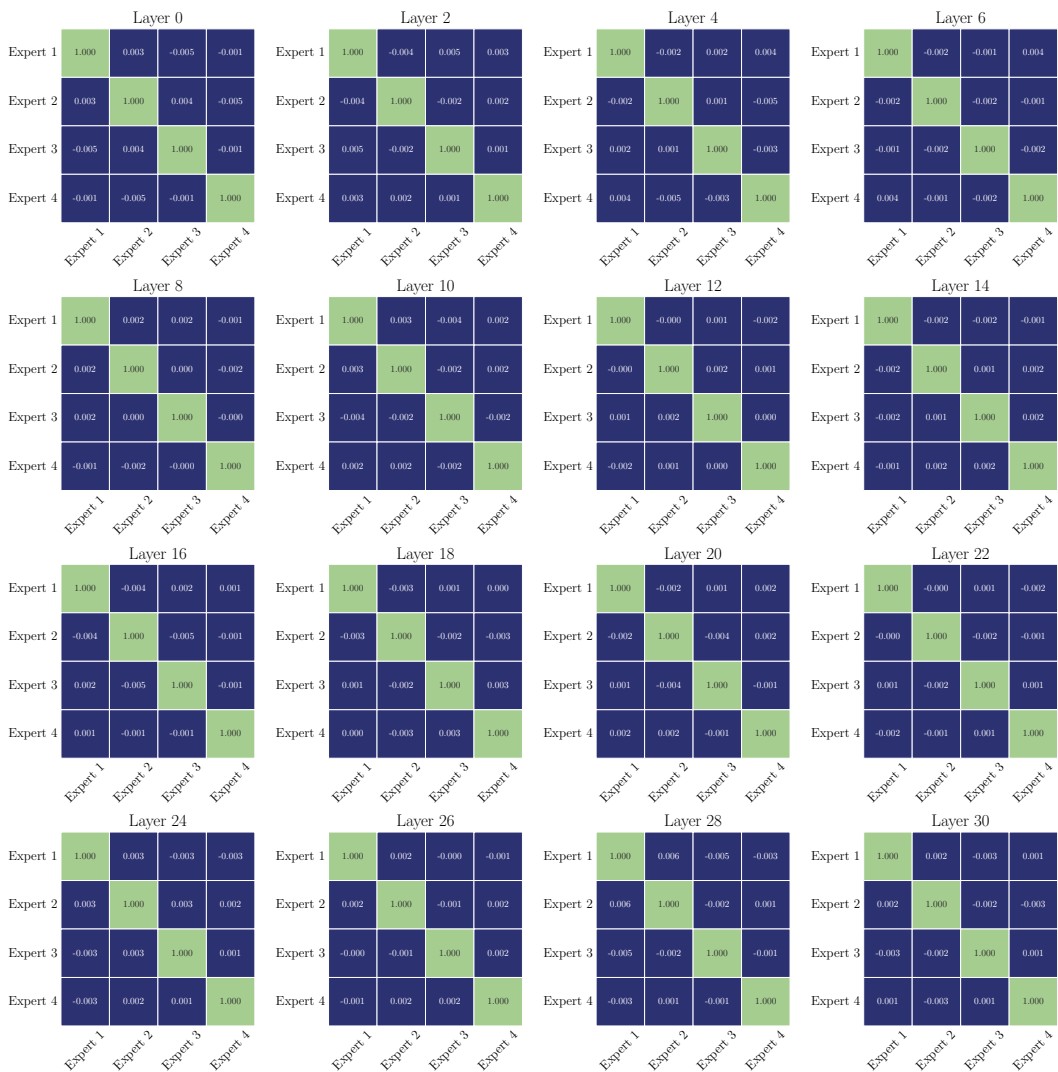

Figure 21: **Layer-wise expert similarity matrix (Phi-2).** We record the experts' cosine similarity per layer during test time. It turns out the cosine similarity between experts is close to 0.

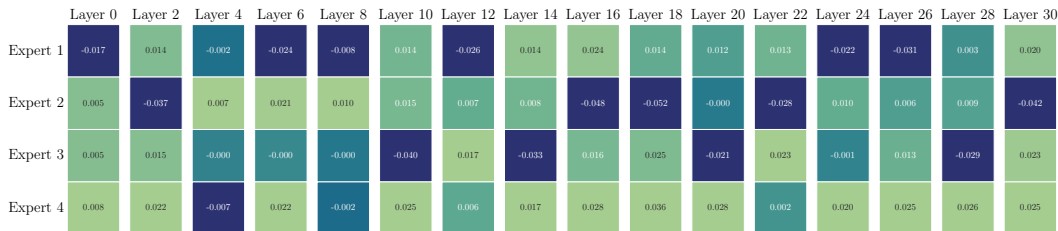

Figure 22: **Layer-wise expert activation threshold (Phi-2).** Darker-colored experts are more likely to be activated compared to lighter-colored experts.

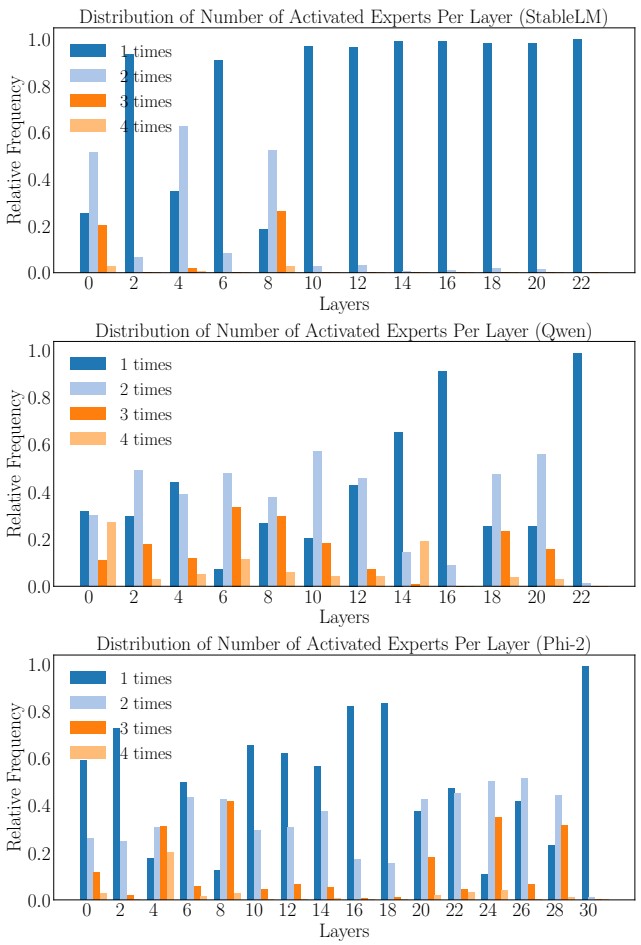

Figure 23: **Distribution of number of activated experts in each layer.** We report the results of StableLM, Qwen, and Phi-2 models, respectively.

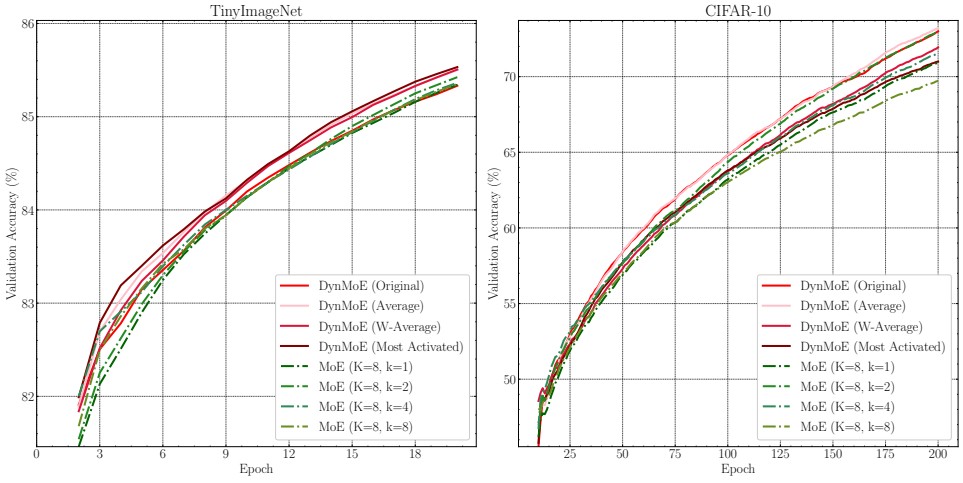

Figure 24: **Convergence curve on CIFAR10 and TinyImageNet datyasets.**

Table 24: **Efficiency evaluation of DYNMOE comparing to MoE-LLaVA.** We conduct experiments on single A100 GPU (80 GB) paired with 16 CPUs using identical environment and identical training/inference configurations. We report the performance of MoE-LLaVA using DeepSpeed's top-2 gating implementation. The symbols ↓ and ↑ indicate that lower and higher values, respectively, denote better performance.

| Model | Training FLOPs ↓ (TFLOPs/step) | Inference FLOPs ↓ (GFLOPs/token) | Inference MACs ↓ (GMACs/token) | Memory Usage ↓ (GB) |
|---|---|---|---|---|
| MoE-LLaVA (StableLM) | 18.23 | 27.62 | 13.34 | 5.98 |
| DynMoE-LLaVA (StableLM) | 17.97 | 25.25 | 12.13 | 5.98 |
| MoE-LLaVA (Qwen) | 34.27 | 23.36 | 11.30 | 6.37 |
| DynMoE-LLaVA (Qwen, Ours) | 34.61 | 22.17 | 10.73 | 6.37 |
| MoE-LLaVA (Phi-2) | 63.43 | 46.87 | 22.73 | 10.46 |
| DynMoE-LLaVA (Phi-2) | 63.36 | 44.92 | 21.72 | 10.46 |

