# OpenReview forum: "Dynamic Mixture of Experts: An Auto-Tuning Approach for Efficient Transformer Models"
_ICLR.cc/2025/Conference — ICLR 2025 Poster_

### Official Review · Reviewer_DhtB · 2024-10-22

**Soundness:** 3
**Presentation:** 3
**Contribution:** 3
**Rating:** 6
**Confidence:** 4

**Summary:**

This manuscript proposes an MoE framework in which the activated expert numbers could be learned, and each expert can be removed or added during the training phase. This framework removes the hand-design requirements for expert numbers, and outperform the hand-designed MoEs in vision, language and multi-modal tasks.

**Strengths:**

- The studied problem is interesting and well-motivated, as the design of MoE have been adjusted by hand in previous studies;
- The proposed method is novel and reasonable;
- Empirical evaluation is relatively comprehensive to support the claims, and the observations provide useful insights for researchers in the field;
- Good readability, which helps readers clearly understand the contribution and advantages.

**Weaknesses:**

Overall, this is a novel and solid work. I have some kind suggestions for further improving it:

- Experiments
  - For vision tasks, why not conduct experiments on the well-known ImageNet dataset? To my understanding, MoE is designed for larger-scaled data and model sizes. Experiments with larger backbones on ImageNet would make the method more convincing.
  - Comparison with existing MoE models would strengthen the proposed approach, such as Swin-MoE [1], VoE [2] (on vision) and Switch Transformer [3], Mistral MoE [4], DeepSeek-MoE [5] (on language).
  - Maybe it is useful to discuss the advatanges of this well designed MoE over those conventional backbones at the same computation budgets.

- Literature review. To my understanding, the proposed method lies in the field of dynamic neural networks [6]. It is encouraged to discuss more about the relationship to other dynamic paradigms, such as layer skipping [7]. When it allows a token to select 0 expert, the proposed framework would naturally allows layer skipping.

[1] Tutel: Adaptive Mixture-of-Experts at Scale.

[2] Scaling Vision with Sparse Mixture of Experts. NeurIPS, 2021.

[3] Switch Transformers: Scaling to Trillion Parameter Models with Simple and Efficient Sparsity. JMLR, 2023.

[4]  Mixtral of Experts.

[5]  Deepseekmoe: Towards ultimate expert specialization in mixture-of-experts language models.

[6] Dynamic Neural Networks: A Survey. TPAMI, 2021.

[7] Dynamic tuning towards parameter and inference efficiency for vit adaptation. NeurIPS, 2024.

**Questions:**

Please refer to the weakness section.

---

> ### Author Response · Authors · 2024-11-22
> **Reply to Reviewer DhtB**
>
> Thank you for your valuable suggestions. We have addressed the questions you raised below.
> > More experiments
> >
>
> Thank you for your suggestion, we have added the additional experiments, including
>
> - Finetuning ViT-S on TinyImageNet dataset.
> - Train ViT-S from scratch on CIFAR10 dataset. We train the models for 200 epochs due to the time limitation.
> - Add an additional baseline [8] on MoE-LLaVA experiment.
>
> Note that our BERT and ViT experiments are already conducted based on [1].  Furthermore, we conduct ViT experiments on TinyImagNet. We investigate various expert initialization strategies to further validate our findings, including:
>
> 1. **Average**: Averaging the parameters of existing experts to initialize the new expert.
> 2. **W-Average**: Using weighted averaging of the parameters of existing experts, where the weights correspond to the number of experts to be activated.
> 3. **Most activated**: Initializing the new expert using the parameters of the most frequently activated expert.
>
> The results show that:
>
> 1. DynMoE converge faster than standard MoE settings (refer to Figure 24 of page 25 for more details);
> 2. W-Average achieve the best performance in most cases.
> 3. DynMoE achieves better performance than top-$p$ routing [8] without the requirements of the additional hyper-parameter $p$.
>
> | TinyImageNet (Finetune, ViT-S, 2 MoE layers) | E1 | E3 | E5 | E7 | E9 | E11 | E13 | E15 | E17 | E19 | E20 |
> | --- | --- | --- | --- | --- | --- | --- | --- | --- | --- | --- | --- |
> | MoE (K = 8, k = 1) | 78.32 | 82.79 | 84.03 | 84.83 | 85.20 | 85.61 | 85.82 | 86.27 | 86.44 | 86.61 | 86.65 |
> | MoE (K = 8, k = 2) | 78.53 | 82.95 | 84.05 | 84.74 | 84.99 | 85.00 | 85.95 | **86.45** | **86.63** | 86.58 | 86.72 |
> | MoE (K = 8, k = 4) | **79.25** | 83.38 | 83.73 | 84.72 | 85.00 | 85.50 | 85.93 | 86.27 | 86.00 | 86.64 | 86.56 |
> | MoE (K = 8, k = 8) | 79.20 | 83.30 | 84.02 | 84.10 | 84.86 | 85.62 | 86.08 | 86.12 | 86.44 | **86.73** | 86.58 |
> | DynMoE (Original, avg topk=6.5) | 79.10 | 83.09 | 84.20 | 84.84 | 85.18 | 85.56 | 85.91 | 86.09 | 86.37 | 86.40 | 86.70 |
> | DynMoE (Average, avg topk=6.0) | 79.19 | **83.48** | **84.21** | 84.84 | **85.32** | 85.76 | 86.25 | **86.41** | 86.49 | **86.70** | **86.75** |
> | DynMoE (W-Average, avg topk=6.0) | 78.96 | 83.18 | 84.15 | **84.92** | **85.34** | **85.93** | 86.10 | 86.30 | 86.60 | **86.70** | **86.80** |
> | DynMoE (Most activated, avg topk=6.5) | 79.09 | **83.57** | **84.21** | 84.62 | 85.40 | 85.87 | **86.45** | **86.40** | **86.63** | 86.66 | 86.70 |
>
> | CIFAR10 (ViT-S, 2 MoE Layer, Acc per 50 Epoch) | 10 | 50 | 100 | 150 | 200 |
> | --- | --- | --- | --- | --- | --- |
> | MoE (K = 8, k = 1) | 46.18 | 59.76 | 64.85 | 68.64 | 72.66 |
> | MoE (K = 8, k = 2) | 47.10 | 60.44 | 66.07 | **70.86** | **73.79** |
> | MoE (K = 8, k = 4) | 46.71 | 59.40 | 64.73 | 69.07 | 72.77 |
> | MoE (K = 8, k = 8) | 48.07 | 60.85 | 64.85 | 68.50 | 70.68 |
> | DynMoE (Original, avg topk=7) | 45.54 | **61.88** | **66.91** | **70.91** | **74.84** |
> | DynMoE (Average, avg topk=7) | 47.05 | **62.77** | **67.20** | **71.24** | **74.70** |
> | DynMoE (W-Average, avg topk=6.5) | **48.56** | **60.94** | 65.00 | 69.66 | **74.16** |
> | DynMoE (Most activated, avg topk=6.5) | 45.79 | 60.84 | 65.83 | 69.48 | 71.71 |
>
> | MoE-LLaVA (StableLM)  | VQAv2 | GQA | VizWiz | SQA | TextVQA | POPE | MME | MMBench |
> | --- | --- | --- | --- | --- | --- | --- | --- | --- |
> | ours | **77.4** | 61.4 | **40.6** | **63.4** | **48.9** | **85.7** | 1300.9 | **63.2** |
> | top p (p=0.4) | 77.1 | **61.7** | 36.0 | 62.8 | 48.6 | 85.2 | **1332.9** | 62.3 |
>
> > Literature review. To my understanding, the proposed method lies in the field of dynamic neural networks [6]. It is encouraged to discuss more about the relationship to other dynamic paradigms, such as layer skipping [7]. When it allows a token to select 0 expert, the proposed framework would naturally allows layer skipping.
> >
>
> Thank you for the suggestion. We have revised the paper (lines 535-539) to include a discussion about dynamic neural networks [6,7]. It is worth noting that in our current implementation, tokens are not allowed to select 0 experts during inference. However, we believe that this would be a valuable extension for future work.
>
>
> [8] Harder Tasks Need More Experts: Dynamic Routing in MoE Models. arXiv preprint 2024.

---

> > ### Author Response · Authors · 2024-11-24
> >
> > Dear Reviewer DhtB,
> >
> > We update the the experiment results of training from scratch on CIFAR10. DynMoE (Original), DynMoE (Average), and DynMoE (W-Average) not only achieve faster convergence rate, but also achieve higher final accuracy than standard MoE.
> >
> > | CIFAR10 (ViT-S, 2 MoE Layer, Acc per 100 Epoch) | 200       | 300       | 400       | 500       |
> > | ----------------------------------------------- | --------- | --------- | --------- | --------- |
> > | MoE (K = 8, k = 1)                              | 72.66     | 77.51     | 80.10     | 81.08     |
> > | MoE (K = 8, k = 2)                              | **73.79** | **78.50**     | **80.85**     | **81.91**     |
> > | MoE (K = 8, k = 4)                              | 72.77     | 77.84     | 80.30     | 81.14     |
> > | MoE (K = 8, k = 8)                              | 70.68     | 75.32     | 78.28     | 79.11     |
> > | DynMoE (Original, avg topk=7)                   | **74.84** | **79.24** | **81.77** | **82.50** |
> > | DynMoE (Average, avg topk=7)                    | **74.70** | **80.32** | **82.51** | **83.57** |
> > | DynMoE (W-Average, avg topk=6.5)                | **74.16** | **78.77** | **81.30** | **82.01** |
> > | DynMoE (Most activated, avg topk=6.5)           | 71.71     | 77.56     | 80.08     | 80.58     |
> >
> > We hope our responses have sufficiently addressed your concerns. Please do not hesitate to reach out if you have any further questions. Thank you again for your time and effort in reviewing our paper.

---

> > > ### Comment · Reviewer_DhtB · 2024-11-25
> > > **Questions about experiments on ImageNet**
> > >
> > > Thanks for the authors' detailed rebuttal, which addresses most of my concerns. I'm still curious about the results on the whole ImageNet dataset.

---

> > > > ### Author Response · Authors · 2024-12-02
> > > > **Results on ImageNet**
> > > >
> > > > Dear reviewer DhtB,
> > > >
> > > > We apologize for the late response. We have now completed the experiments on ImageNet with 224×224 resolution.
> > > >
> > > > ### Experiment Settings
> > > >
> > > > For all ImageNet experiments, we train ViT-S from scratch for 200 epochs with a batch size of 512 across 8 A100 GPUs. In the ViT-S architecture, the layers [0, 3, 6, 9] are replaced with MoE layers. The learning rate is set to 1e-4, and we use the Adam optimizer with parameters [0.9, 0.99] and cosine learning schedule, while the weight decay is set to 5e-5. During evaluation, we set the batch size to 128 and use 1 A100 GPU.
> > > >
> > > > - **Baselines**: We use DeepSpeed-MoE with top-1/2 gating, and the number of experts in each layer is set to 4.
> > > > - **DynMoE**: We use the same settings as in the MoE-LLaVA experiments, where the initial number of experts is set to 2, with a maximum of 4 experts per layer. We evaluate two variants of DynMoE:
> > > >     - **DynMoE-A**: DynMoE with a `simple and diverse gating loss`
> > > >     - **DynMoE-B**: DynMoE with `simple and diverse gating loss` + `load balance loss` + `efficiency loss`
> > > >
> > > > From the experiment, we observe the following:
> > > >
> > > > 1. **DynMoE-A**: Achieves slightly lower performance compared to the Top-2 MoE, but with a smaller average $k$ and reduced training/inference time.
> > > > 2. **DynMoE-B**: When we add the load balance loss and efficiency loss, DynMoE-B not only improves accuracy but also enhances efficiency.
> > > >
> > > > |  | Train Time (s/batch) | Train Wall-Clock Time (days) | Evaluation Time (s/batch) | Inference Time on Routing (ms / batch and MoE layer) | Inference Time on Gating (ms / batch and MoE layer) | Inference Time on Expert Passing (ms / batch and MoE layer) | Acc@1 (%) |
> > > > | --- | --- | --- | --- | --- | --- | --- | --- |
> > > > | DeepSpeed-MoE (top-1) | 0.39 | 2.3 | 0.53 | 0.06 | 14.54 | 94.32 | 64.7 |
> > > > | DeepSpeed-MoE (top-2) | 0.63 | 3.7 | 1.05 | 0.06 | 50.78 | 186.47 | 67.3 |
> > > > | DynMoE-A (k=1.63) | 0.51 | 2.8 | 0.99 | 0.53 | 36.45 | 186.18 | 66.5 |
> > > > | DynMoE-B (k=1.25) | 0.41 | 2.5 | 0.82 | 0.46 | 31.04 | 148.11 | 68.5 |
> > > >
> > > > We hope the results on the whole ImageNet dataset bring additional insights and address your questions. Thank you once again for your valuable time and effort in reviewing our paper.

---

> > > > > ### Comment · Reviewer_DhtB · 2024-12-03
> > > > > **Response**
> > > > >
> > > > > I thank the authors for their additional results and analysis on ImageNet. I shall keep my positive rating. It is kindly encouraged to evaluate the proposed method on larger-scale network structures.

---

> > > > > > ### Author Response · Authors · 2024-12-03
> > > > > > **Reply to reviewer DhtB**
> > > > > >
> > > > > > Dear Reviewer DhtB,
> > > > > >
> > > > > > Thank you for your positive feedback! We are committed to further refining the manuscript to improve its clarity and overall quality. We sincerely appreciate your time, effort, and thoughtful insights.

---

### Official Review · Reviewer_kDxr · 2024-10-23

**Soundness:** 3
**Presentation:** 3
**Contribution:** 3
**Rating:** 8
**Confidence:** 4

**Summary:**

This paper introduces the DYNMOE, an algorithm that automatically determines the number of experts activated for each token, addressing the dependency on hyperparameters (such as the number of experts and the number of activated experts) in SMoE models. DYNMOE uses an innovative top-any gating method and an adaptive training process to dynamically adjust the number of experts, while incorporating auxiliary loss functions to ensure sparsity and diversity among experts, improving training and inference efficiency. Experiments demonstrate that DYNMOE achieves comparable or superior performance and efficiency to well-tuned MoE configurations across vision, language, and vision-language tasks.

**Strengths:**

1 The paper clearly analyzes existing issues in MoE, particularly how the fixed top-k approach may not be optimal.

2 The authors consider the extreme case of activating all experts in the top-any scenario and design a regularization loss to address this.

3 The writing logic is clear.

**Weaknesses:**

1 In the top-any routing, is G a hyperparameter that needs to be predefined? Would this lead to cumbersome tuning?

2 In the top-any routing, the selected k experts considered equally important by default. Would introducing varying importance levels for different experts yield better results? Although the authors mention related content in Remark 3.1, ablation experiments seem necessary.

3 Why top-any MoE training strategy dones't lead to increased training time?

4 The author could further elaborate on how the diversity loss in the auxiliary loss prevents activating all experts simultaneously.

5 How exactly does DYNMOE remove experts? For example, if the interval for adding/removing experts is set to 300 steps and expert x is activated in the first 299 steps but not in the 300th step, will it be removed? Alternatively, if expert x is activated in the first step but not for the remaining 299 steps, will it be removed?

6 For new experts added in DYNMOE, Would averaging the weights of other experts to initialize the new expert’s weights lead to better results? Or would random initialization yield better performance?

**Questions:**

see weakness.

---

> ### Author Response · Authors · 2024-11-22
> **Reply to Reviewer kDxr (1/2)**
>
> Thank you for your valuable suggestions. We have addressed the questions you raised below.
> > In the top-any routing, is G a hyperparameter that needs to be predefined? Would this lead to cumbersome tuning?
> >
>
> We are sorry for the misunderstanding. The $G$ are trainable parameters, and are initially set to 0. In Figures 17-22 of the Appendix, we have reported the value of learned threshold $G$. Our findings indicate that there exists a shared expert that more easier to be activated in each MoE layer.
>
> > In the top-any routing, the selected k experts considered equally important by default. Would introducing varying importance levels for different experts yield better results? Although the authors mention related content in Remark 3.1, ablation experiments seem necessary.
> >
>
> We would like to clarify the following points:
>
> - In top-*k* gating, the gating scores are computed using the softmax function, which allows for the comparison of gating scores among different experts. Specifically, a higher score for some experts results in a lower score for others.
> - In contrast, in the top-any gating mechanism of DynMoE, the gating scores for different experts are calculated independently using sigmoid function (see figure 2 for a more intuitive understanding). Consequently, these scores are not comparable among experts. As such, we believe that the range of expert scores does not accurately reflect their importance.
> - Additionally, we have conducted an ablation study to demonstrate the effectiveness of this design choice. Results show that using different expert scores significantly reduce the model performance.
>
> | StableLM | VQAv2 | GQA | VizWiz | SQA | TextVQA | POPE | MME | MMBench |
> | --- | --- | --- | --- | --- | --- | --- | --- | --- |
> | ours | **77.4** | **61.4** | **40.6** | **63.4** | **48.9** | **85.7** | **1300.9** | **63.2** |
> | weighted scores | 73.9 | 57.4 | 32.1 | 61.3 | 46.9 | 84.2 | 1176.8 | 52.1 |
>
> > Why top-any MoE training strategy dones't lead to increased training time?
> >
>
> We apologize for the misunderstanding and would like to clarify that the comparable training FLOPs in Table 4 of our submission is due to DynMoE activating fewer experts compared to MoE-LLaVA. Consequently, DynMoE's inference is more efficient. However, the training of DynMoE is not faster than MoE-LLaVA because of the additional training FLOPs associated with top-any routing and adaptive expert addition/removal.
>
> > The author could further elaborate on how the diversity loss in the auxiliary loss prevents activating all experts simultaneously.
> >
>
> Thank you for the suggestion. The diverse loss serves as a weak regularization for sparsity because:
>
> - Our routing method calculates the cosine similarities between tokens and routing parameters, and then uses the similarity score as the gating score. As the router parameters are forced to be diverse, tokens with high similarity to certain experts tend to have low similarity to other experts. This prevents tokens from choosing too many experts, thereby promoting sparsity.
> - As shown in the language tasks presented in Table 8 of the attached file, as well as in the vision-language tasks shown in Table 2 of the submission, DynMoE tends to activate only a subset of experts rather than all of them.
>
> > How exactly does DYNMOE remove experts? For example, if the interval for adding/removing experts is set to 300 steps and expert x is activated in the first 299 steps but not in the 300th step, will it be removed? Alternatively, if expert x is activated in the first step but not for the remaining 299 steps, will it be removed?
> >
>
> DynMoE removes experts that are never activated during the interval for adding/removing experts. Consequently, in both scenarios mentioned by the reviewer, the experts will not be removed.

---

> ### Author Response · Authors · 2024-11-22
> **Reply to Reviewer kDxr (2/2)**
>
> > For new experts added in DYNMOE, Would averaging the weights of other experts to initialize the new expert’s weights lead to better results? Or would random initialization yield better performance?
> >
>
> We apologize for the missing details in our submission. To clarify:
>
> - The experts in our MoE layer are initialized by copying the weights of pre-trained models.
> - In our current implementation, we re-use the weights of the most recently discarded expert when adding a new expert to the MoE layer.
> - We conducted additional experiments with various expert initialization strategies when adding new experts, including:
>     1. **Average**: Averaging the parameters of existing experts to initialize the new expert.
>     2. **W-Average**: Using weighted averaging of the parameters of existing experts, where the weights correspond to the number of experts to be activated.
>     3. **Most activated**: Initializing the new expert using the parameters of the most frequently activated expert.
>
>     Results show that (1) DynMoE converges faster than standard MoE settings; (2) W-Average achieves the best performance in most cases (refer to Figure 24 of page 25 for more details).
>
>
> | TinyImageNet (Finetune, ViT-S, 2 MoE layers) | E1 | E3 | E5 | E7 | E9 | E11 | E13 | E15 | E17 | E19 | E20 |
> | --- | --- | --- | --- | --- | --- | --- | --- | --- | --- | --- | --- |
> | MoE (K = 8, k = 1) | 78.32 | 82.79 | 84.03 | 84.83 | 85.20 | 85.61 | 85.82 | 86.27 | 86.44 | 86.61 | 86.65 |
> | MoE (K = 8, k = 2) | 78.53 | 82.95 | 84.05 | 84.74 | 84.99 | 85.00 | 85.95 | **86.45** | **86.63** | 86.58 | 86.72 |
> | MoE (K = 8, k = 4) | **79.25** | 83.38 | 83.73 | 84.72 | 85.00 | 85.50 | 85.93 | 86.27 | 86.00 | 86.64 | 86.56 |
> | MoE (K = 8, k = 8) | 79.20 | 83.30 | 84.02 | 84.10 | 84.86 | 85.62 | 86.08 | 86.12 | 86.44 | **86.73** | 86.58 |
> | DynMoE (Original, avg topk=6.5) | 79.10 | 83.09 | 84.20 | 84.84 | 85.18 | 85.56 | 85.91 | 86.09 | 86.37 | 86.40 | 86.70 |
> | DynMoE (Average, avg topk=6.0) | 79.19 | **83.48** | **84.21** | 84.84 | **85.32** | 85.76 | 86.25 | **86.41** | 86.49 | **86.70** | **86.75** |
> | DynMoE (W-Average, avg topk=6.0) | 78.96 | 83.18 | 84.15 | **84.92** | **85.34** | **85.93** | 86.10 | 86.30 | 86.60 | **86.70** | **86.80** |
> | DynMoE (Most activated, avg topk=6.5) | 79.09 | **83.57** | **84.21** | 84.62 | 85.40 | 85.87 | **86.45** | **86.40** | **86.63** | 86.66 | 86.70 |

---

> > ### Comment · Reviewer_kDxr · 2024-11-27
> >
> > Thanks the authors for their response. Overall, I believe this is an interesting work, and I am willing to raise the score to 8.

---

> > > ### Author Response · Authors · 2024-11-27
> > >
> > > Dear Reviewer kDxr,
> > >
> > > We sincerely appreciate your kind reconsideration and the revised score. Your valuable feedback and the time you’ve spent on our paper are highly appreciated.

---

### Official Review · Reviewer_4q9o · 2024-10-28

**Soundness:** 2
**Presentation:** 2
**Contribution:** 2
**Rating:** 6
**Confidence:** 5

**Summary:**

This work proposess DynMoE, an enhanced version of the Mixture of Experts layer. The method enables top-any gating, allowing for the execution of different number of experts for each token. In the proposed method experts that are unused for a predefined period of training time are discarded, and new experts are added if there are tokens that do not activate any experts.

The proposed method is relatively simple and appears to achieve its intended effect of reducing the hyperparameter tuning burden associated with MoE layers. However, the method is compared to only a single variant of MoE, and improvements are visible on only a subset of the results. There are several other issues, with not adequately detailed description of the method being one of them. As of right now, I recommend rejection of this work, but I am willing to increase the score if the authors properly address my concerns.

**Strengths:**

- Elimination of hyperparameters introduced by MoE, which was the motivation of the paper, is practical and useful, and thus the work could be of interest to the community. The method is also relatively simple, which may be regarded as an advantage.
- The main evaluation is relatively thorough in terms of number of datasets and models. The authors evaluate their method on three different setups, with one being a multimodal task.

**Weaknesses:**

- **Limited novelty and lack of proper attribution:** While the authors demonstrate some improvement over standard MoE layers, the idea of dynamically selecting the number of executed experts on a per-token basis is not new [1,2,3,4]. Moreover, none of these are mentioned in the related work section. The authors should properly discuss these works and highlight the differences of the proposed method.
- **Lack of comparison to existing MoE methods from the literature:** Given the above, does the proposed method provide any advantages over the existing top-any literature? An empirical comparison with these methods would strengthen the contribution. As of right now, the authors compare their method with only a single baseline (one of the top-k MoE variants), which is an extremely low number for an empirical-only work. Would the benefits of the proposed method hold even when stacked against alternative top-k MoE variants? Some of these variants also allow for an effectively different number of experts executed per token in inference [5].
- **Unconvincing real latency/throughput evaluation:** A major flaw of the work is that authors only briefly mention the actual speedup that the method supposedly provides. The authors write (line 488): "To ensure a fair evaluation, MoE-LLaVA employs the expert dispatching implementation from DynMoE by fixing the top-k values.". This statement is unclear, and for me it suggests the opposite of a fair evaluation, as the alternative implementation may significantly slow down the original MoE-LLaVA. The readers should be aware of any overhead stemming from the introduction of top-any routing. I would at least expect a non-negligible overhead stemming from the non-constant number of token-expert executions. Finally, to provide a complete picture, the authors should also measure and report the throughput of dense models for comparison.
- **No from-scratch training experiments:** All experiments are conducted on pre-trained models, which are then converted to sparse MoE models during the fine-tuning. Mixture-of-Experts layer was originally proposed to scale up the number of parameters without slowing down from-scratch training. Although the pre-trained setup is as important, demonstrating that the proposed method is also appropriate for the from-scratch setup would strengthen the paper.
- **Standard deviations not being reported:** The authors do not report standard deviations of the results in any of the experiments. This is particularly surprising as the authors write in the appendix that the scores are averaged over three random seeds. Without standard deviations it is hard to estimate the significance of the results, especially since some of the scores in Tables 1, 2, 3, and Figure 4 are really close. The authors also do not present any evidence that training of the proposed DynMoE is stable in any separate experiment.
- **Method description and experimental setup not being adequately explained:** After multiple passes through the text some crucial aspects are still unclear to me - see the questions section below.
- **Readability:** I would recommend to do a pass through the paper to improve grammar and readability (e.g. "and uses an additional gating network g to predict the scores that the input token embedding assigned to each expert." or "To promising efficiency and avoiding burden communication, we only check if experts required to be added add or removed.").

#### References:

[1] Nie, Xiaonan, et al. "Evomoe: An evolutional mixture-of-experts training framework via dense-to-sparse gate." arXiv preprint arXiv:2112.14397 (2021).

[2] Zheng, Haizhong, et al. "Learn to be efficient: Build structured sparsity in large language models." arXiv preprint arXiv:2402.06126 (2024).

[3] Huang, Quzhe, et al. "Harder Tasks Need More Experts: Dynamic Routing in MoE Models." arXiv preprint arXiv:2403.07652 (2024).

[4] Yang, Yuanhang, et al. "XMoE: Sparse Models with Fine-grained and Adaptive Expert Selection." Findings of the Association for Computational Linguistics ACL 2024. 2024.

[5] Riquelme, Carlos, et al. "Scaling vision with sparse mixture of experts." Advances in Neural Information Processing Systems 34 (2021): 8583-8595.

**Questions:**

- How are experts initialized at the start given a pre-trained model? Are parameters of experts that are added during training initialized randomly, or are the weights of the most recently discarded expert re-used?
- Are the auxiliarry losses somehow weighted? If yes, why is it not reported in the paper?
- Is the MoE implementation used in this work dropless [6], i.e. does it drop tokens if the predefined capacity of an expert is exceeded? If it is not dropless, how often are tokens dropped and how does DynMoE compare to standard MoE in terms of load balance?
- In the appendix the authors write: "For each epoch, we begin recording routing at 1/3 of the epoch and complete recording routing and execute the adaptive process at 2/3 of the epoch." Can the authors clarify why is this needed?
- Currently the method relies on a linear router. However, some works assume a deeper router [7,8]. Can the method be extended to non-linear routers?
- Remark 3.1 is not fully convincing to me. Can the authors explain more in detail why different ranges of expert scores are supposed to be undesirable?

#### References:
[6] Gale, Trevor, et al. "Megablocks: Efficient sparse training with mixture-of-experts." Proceedings of Machine Learning and Systems 5 (2023): 288-304.

[7] Zhang, Zhengyan, et al. "MoEfication: Transformer Feed-forward Layers are Mixtures of Experts." Findings of the Association for Computational Linguistics: ACL 2022. 2022.

[8] Chi, Zewen, et al. "On the representation collapse of sparse mixture of experts." Advances in Neural Information Processing Systems 35 (2022): 34600-34613.

---

> ### Author Response · Authors · 2024-11-22
> **Reply to Reviewer 4q9o (1/5)**
>
> Thank you for your valuable suggestions. We have addressed the questions you raised below.
> > **Limited novelty and lack of proper attribution:** While the authors demonstrate some improvement over standard MoE layers, the idea of dynamically selecting the number of executed experts on a per-token basis is not new [1,2,3,4]. Moreover, none of these are mentioned in the related work section. The authors should properly discuss these works and highlight the differences of the proposed method.
> >
>
> We appreciate the reviewer's feedback on the matter. However, we believe that **NONE** of these studies [1,2,3,4], including several contemporaneous ones, have addressed the issues we intend to tackle in this paper. Consequently, our approach remains valuable to the research community.
>
> - Regarding the use of hyper-parameters for gating, while some studies [1,2,3,4] enable tokens to activate different number of experts, new hyper-parameters are introduced. **DynMoE does not require any additional hyper-parameters and truly eliminates the need for the hyper-parameter $k$.**
> - Some studies [1, 2] rely on dense training during the initial stages, which may not align well with the intention of utilizing MoE architectures in LLMs, namely, scaling up models with fewer computational resources. Consequently, [1] did not conduct experiments on LLMs, and [2] did not provide a comparison of training efficiency. In contrast, **DynMoE consistently uses sparse gating for all training and inference stages and offers a comprehensive discussion on efficiency.**
> - Furthermore, unlike [1,2,3,4], which cannot adjust the number of experts $K$, **DynMoE allows for the automatic adjustment of the number of experts during training.**
> - In addition to its innovative architecture, the results of DynMoE offer valuable insights into the design of MoE architectures. As detailed in Section 4.5, we can observe two key findings: (1) the bottom layers require a greater number of experts; and (2) there are shared experts present within each MoE layer.
>
> In summary, we maintain that DynMoE retains significant value to the community in terms of novelty and contribution.
>
> > **Lack of comparison to existing MoE methods from the literature:** Given the above, does the proposed method provide any advantages over the existing top-any literature? An empirical comparison with these methods would strengthen the contribution. As of right now, the authors compare their method with only a single baseline (one of the top-k MoE variants), which is an extremely low number for an empirical-only work. Would the benefits of the proposed method hold even when stacked against alternative top-k MoE variants? Some of these variants also allow for an effectively different number of experts executed per token in inference [5].
> >
>
> Thank you for your suggestion. We would like to clarify the following points, and have included the discussion about related works in the revised paper (lines 103-107):
>
> 1. We compare the DynMoE to the standard top-$k$ MoEs due to the following reasons:
>     - The top-$k$ MoE remains a strong baseline and is prevalent in recent MoE-based LLMs [9, 10].
>     - DynMoE offers several advantages over existing MoE variants: (1) it does not require hyperparameters during gating, (2) it can automatically adjust the number of experts, (3) it supports sparse training, and (4) it does not rely on specialized training stages.
> 2. During the rebuttal, we have incorporated additional baselines. For the related studies [1,2,3,4]
>     1. [1,2] relies on dense training, thereby difficult to conduct a fair evaluation.
>     2. Both [3,4] share the same key idea that changing top-$k$ gating to top-$p$ gating, thereby we use [3] as the baseline algorithm we compared, and using the $p=0.4$ as suggested in their paper.
>     3. We can find that DynMoE achieves better performance to [3] (where $p$ is set to 0.4 as in the paper) without the additional parameter $p$.
>
> | StableLM | VQAv2 | GQA | VizWiz | SQA | TextVQA | POPE | MME | MMBench |
> | --- | --- | --- | --- | --- | --- | --- | --- | --- |
> | ours | **77.4** | 61.4 | **40.6** | **63.4** | **48.9** | **85.7** | 1300.9 | **63.2** |
> | top p (p=0.4) | 77.1 | **61.7** | 36.0 | 62.8 | 48.6 | 85.2 | **1332.9** | 62.3 |

---

> ### Author Response · Authors · 2024-11-22
> **Reply to Reviewer 4q9o (2/5)**
>
> > **Unconvincing real latency/throughput evaluation:** A major flaw of the work is that authors only briefly mention the actual speedup that the method supposedly provides. The authors write (line 488): "To ensure a fair evaluation, MoE-LLaVA employs the expert dispatching implementation from DynMoE by fixing the top-k values.". This statement is unclear, and for me it suggests the opposite of a fair evaluation, as the alternative implementation may significantly slow down the original MoE-LLaVA. The readers should be aware of any overhead stemming from the introduction of top-any routing. I would at least expect a non-negligible overhead stemming from the non-constant number of token-expert executions.
>
> We apologize for any previous ambiguity in our statement. However, we stand by our claim that we conducted a fair evaluation.
>
> - The forward pass of a MoE layer consists of three key steps:
>     1. Forwarding the gating network to generate gating scores.
>     2. Dispatching tokens to the appropriate experts based on the gating scores and a top-$k$ selection process.
>     3. Passing the tokens through the selected experts and combining their results.
>
> MoE-LLaVA utilizes the Deepspeed-MoE implementation. However, despite its efficiency, the Deepspeed-MoE implementation for step (2) is limited to supporting only top-1 and top-2 gating. Consequently, we developed our own implementation for step (2), which resulted in some efficiency degradation due to additional engineering efforts, i.e., we did not incorporate a CUDA implementation for parallel computation.
>
> **To ensure a fair comparison focused on algorithm design, we maintained the same implementation for step (2) but varied the value of $k$ for MoE-LLaVA and DynMoE.** For steps (1) and (3), MoE-LLaVA adopted the original implementation provided by Deepspeed-MoE.
>
> In summary, we believe we have presented a fair comparison.
>
> > Finally, to provide a complete picture, the authors should also measure and report the throughput of dense models for comparison.
>
>
> We have provided the throughput of dense models below.  Overall, the dense 1.6B model activates fewer parameters and is more efficient than MoE variants, while DynMoE (row 3) is more efficient than the standard MoE (row 2).
>
> |  | Activation Parameters (B) | Inference FLOPs (GFLOPs/token) | Inference MACs (GFLOPs/token) | Throughput (output token/s) | Memory Usage (GB) |
> | --- | --- | --- | --- | --- | --- |
> | Dense-LLaVA (StableLM-1.6B) | 1.60 | 15.52 | 7.75 | 32 | 3.68 |
> | MoE-LLaVA (StableLM-1.6B $\times$ 4) | 2.06 | 27.62 | 13.34 | 19 | 5.98 |
> | DynMoE-LLaVA (StableLM-1.6B $\times$ 4) | 1.75 | 25.25 | 12.13 | 26 | 5.98 |
>
> > **No from-scratch training experiments:** All experiments are conducted on pre-trained models, which are then converted to sparse MoE models during the fine-tuning. Mixture-of-Experts layer was originally proposed to scale up the number of parameters without slowing down from-scratch training. Although the pre-trained setup is as important, demonstrating that the proposed method is also appropriate for the from-scratch setup would strengthen the paper.
> >
>
> Thank you for your suggestion, we would like to clarify that:
>
> 1. MoE is also crucial for the fine-tuning stage in the LLM literature [2, 10]. By using MoE, we can scale up a pretrained LLM to a larger one with minimal additional cost.
> 2. Additionally, we have conducted an experiment where we trained the model from scratch to further validate our findings. We trained ViT-S on the CIFAR-10 dataset from scratch, and train the models for 200 epochs due to the time limitation. We use various expert initialization strategies while adding the new experts, including:
>     1. **Average**: Averaging the parameters of existing experts to initialize the new expert.
>     2. **W-Average**: Using weighted averaging of the parameters of existing experts, where the weights correspond to the number of experts to be activated.
>     3. **Most activated**: Initializing the new expert using the parameters of the most frequently activated expert.
>
>     The results show that DynMoE (Original), DynMoE (Average), and DynMoE (W-Average) achieve faster convergence rate than the standard MoE (refer to Figure 24 of page 25 for more details).
> | CIFAR10 (ViT-S, 2 MoE Layer, Acc per 50 Epoch) | 10 | 50 | 100 | 150 | 200 |
> | --- | --- | --- | --- | --- | --- |
> | MoE (K = 8, k = 1) | 46.18 | 59.76 | 64.85 | 68.64 | 72.66 |
> | MoE (K = 8, k = 2) | 47.10 | 60.44 | 66.07 | **70.86** | **73.79** |
> | MoE (K = 8, k = 4) | 46.71 | 59.40 | 64.73 | 69.07 | 72.77 |
> | MoE (K = 8, k = 8) | 48.07 | 60.85 | 64.85 | 68.50 | 70.68 |
> | DynMoE (Original, avg topk=7) | 45.54 | **61.88** | **66.91** | **70.91** | **74.84** |
> | DynMoE (Average, avg topk=7) | 47.05 | **62.77** | **67.20** | **71.24** | **74.70** |
> | DynMoE (W-Average, avg topk=6.5) | **48.56** | **60.94** | 65.00 | 69.66 | **74.16** |
> | DynMoE (Most activated, avg topk=6.5) | 45.79 | 60.84 | 65.83 | 69.48 | 71.71 |

---

> ### Author Response · Authors · 2024-11-22
> **Reply to Reviewer 4q9o (3/5)**
>
> > **Standard deviations not being reported:** The authors do not report standard deviations of the results in any of the experiments. This is particularly surprising as the authors write in the appendix that the scores are averaged over three random seeds. Without standard deviations it is hard to estimate the significance of the results, especially since some of the scores in Tables 1, 2, 3, and Figure 4 are really close. The authors also do not present any evidence that training of the proposed DynMoE is stable in any separate experiment.
> >
>
> Thank you for the suggestion!
>
> - We have added the std for language tasks. Results show that DynMoE generally has lower std value, indicating more stable performance.
> - Due to the high computation cost, LLMs general do not report this [2, 11, 12].
> - **Furthermore, we would like to emphasize that the primary objective of this paper is to eliminate the reliance on hyper-parameters, rather than solely improving model performance.**
>
> | Algorithms | COLA | MRPC | QNLI | MNLI | RTE | Average |
> | --- | --- | --- | --- | --- | --- | --- |
> | MoE (K = 8, k = 1) | 64.10$\pm$0.94 | 90.14$\pm$0.60 | 92.48$\pm$0.21 | 86.56$\pm$0.06 | 73.04$\pm$2.13 | 81.26 |
> | MoE (K = 8, k = 2) | 64.51$\pm$0.81 | 90.19$\pm$0.17 | 92.39$\pm$0.08 | 86.70$\pm$0.23 | 74.85$\pm$1.96 | 81.73 |
> | MoE (K = 8, k = 4) | 64.94$\pm$0.62 | 89.74$\pm$0.99 | 92.52$\pm$0.12 | 86.57$\pm$0.28 | 75.09$\pm$1.84 | 81.77 |
> | MoE (K = 8, k = 8) | 64.03$\pm$0.54 | 89.36$\pm$0.09 | 92.46$\pm$0.09 | 86.61$\pm$0.26 | 74.37$\pm$0.78 | 81.37 |
> | MoE (K = 16, k = 1) | 63.63$\pm$0.20 | 89.81$\pm$0.30 | 92.39$\pm$0.21 | 86.63$\pm$0.17 | 74.01$\pm$0.29 | 81.29 |
> | MoE (K = 16, k = 2) | 64.71$\pm$1.21 | 90.18$\pm$1.33 | 92.53$\pm$0.07 | 86.73$\pm$0.43 | 72.32$\pm$3.54 | 81.29 |
> | MoE (K = 16, k = 4) | 64.12$\pm$1.42 | 89.74$\pm$0.40 | 92.65$\pm$0.09 | 86.59$\pm$0.16 | 75.33$\pm$0.95 | 81.69 |
> | MoE (K = 16, k = 8) | 64.37$\pm$1.14 | 90.35$\pm$0.68 | 92.49$\pm$0.11 | 86.51$\pm$0.20 | 73.53$\pm$2.21 | 81.45 |
> | DynMoE, Gshard Loss | 64.88$\pm$0.86 | 89.85$\pm$0.22 | 92.42$\pm$0.07 | - | 73.41$\pm$0.68 | - |
> | DynMoE | 65.17$\pm$0.26 | 90.64$\pm$0.26 | 92.59$\pm$0.08 | 86.37$\pm$0.13 | 73.41$\pm$1.96 | 81.64 |
>
> > **Readability:** I would recommend to do a pass through the paper to improve grammar and readability (e.g. "and uses an additional gating network g to predict the scores that the input token embedding assigned to each expert." or "To promising efficiency and avoiding burden communication, we only check if experts required to be added add or removed.").
> >
>
> Thank you for the suggestion. We have revised the first paragraph as follows and removed the additional part from the second sentence.
>
> ```
> The traditional top-$k$ gating method takes the token embedding $x$ as input and employs an additional gating network $g$ to predict the gating scores. These gating scores are then used to determine which experts will be activated for the input tokens.
> ```
>
> > Are the auxiliarry losses somehow weighted? If yes, why is it not reported in the paper?
> >
>
> We are sorry for the missing details. We use the default setting of EMoE and MoE-LLaVA for the weights of auxiliarry losses. For both of these two settings, the weights of auxiliarry losses are set to 1e-2.
>
> > In the appendix the authors write: "For each epoch, we begin recording routing at 1/3 of the epoch and complete recording routing and execute the adaptive process at 2/3 of the epoch." Can the authors clarify why is this needed?
> >
>
> Thank you for your detailed comments. Our design choice aims to prevent experts from being frequently removed or added without adequate training and gating information. By using this method, we can ensure:
>
> 1. Sufficient gating information is collected during the epoch range from 1/3 to 2/3 to make informed decisions about adding or removing experts.
> 2. Experts are adequately trained before the next round of expert addition/removal.
>
> We found that this design choice stabilizes the training process.
>
> > Currently the method relies on a linear router. However, some works assume a deeper router [7,8]. Can the method be extended to non-linear routers
> >
>
> Thank you for your insightful comments. While using a deeper router is feasible, it may come at the cost of losing some of the benefits associated with DynMoE. Specifically:
>
> - **We can still perform top-any gating** by passing tokens through the router and then applying the designed threshold sigmoid gating to activate the appropriate number of experts.
> - **The model removal process remains effective when using deep routers, but the model addition process becomes more challenging.** In the current implementation, we can initialize the router parameters of new experts using the embeddings of tokens that do not choose any experts, ensuring that these tokens will activate the new expert. However, this simple initialization method may not be feasible when using a deep router.

---

> ### Author Response · Authors · 2024-11-22
> **Reply to Reviewer 4q9o (4/5)**
>
> > How are experts initialized at the start given a pre-trained model? Are parameters of experts that are added during training initialized randomly, or are the weights of the most recently discarded expert re-used?
> >
>
> We apologize for the missing details in our submission. To clarify:
>
> - The experts in our MoE layer are initialized by copying the weights of pre-trained models.
> - In our current implementation, we re-use the weights of the most recently discarded expert when adding a new expert to the MoE layer.
> - We conducted additional experiments with various expert initialization strategies when adding new experts, including:
>     1. **Average**: Averaging the parameters of existing experts to initialize the new expert.
>     2. **W-Average**: Using weighted averaging of the parameters of existing experts, where the weights correspond to the number of experts to be activated.
>     3. **Most activated**: Initializing the new expert using the parameters of the most frequently activated expert.
>
>     Results show that (1) DynMoE converge faster than standard MoE settings; (2) W-Average achieve the best performance in most cases (refer to Figure 24 of page 25 for more details).
>
> | TinyImageNet (Finetune, ViT-S, 2 MoE layers) | E1 | E3 | E5 | E7 | E9 | E11 | E13 | E15 | E17 | E19 | E20 |
> | --- | --- | --- | --- | --- | --- | --- | --- | --- | --- | --- | --- |
> | MoE (K = 8, k = 1) | 78.32 | 82.79 | 84.03 | 84.83 | 85.20 | 85.61 | 85.82 | 86.27 | 86.44 | 86.61 | 86.65 |
> | MoE (K = 8, k = 2) | 78.53 | 82.95 | 84.05 | 84.74 | 84.99 | 85.00 | 85.95 | **86.45** | **86.63** | 86.58 | 86.72 |
> | MoE (K = 8, k = 4) | **79.25** | 83.38 | 83.73 | 84.72 | 85.00 | 85.50 | 85.93 | 86.27 | 86.00 | 86.64 | 86.56 |
> | MoE (K = 8, k = 8) | 79.20 | 83.30 | 84.02 | 84.10 | 84.86 | 85.62 | 86.08 | 86.12 | 86.44 | **86.73** | 86.58 |
> | DynMoE (Original, avg topk=6.5) | 79.10 | 83.09 | 84.20 | 84.84 | 85.18 | 85.56 | 85.91 | 86.09 | 86.37 | 86.40 | 86.70 |
> | DynMoE (Average, avg topk=6.0) | 79.19 | **83.48** | **84.21** | 84.84 | **85.32** | 85.76 | 86.25 | **86.41** | 86.49 | **86.70** | **86.75** |
> | DynMoE (W-Average, avg topk=6.0) | 78.96 | 83.18 | 84.15 | **84.92** | **85.34** | **85.93** | 86.10 | 86.30 | 86.60 | **86.70** | **86.80** |
> | DynMoE (Most activated, avg topk=6.5) | 79.09 | **83.57** | **84.21** | 84.62 | 85.40 | 85.87 | **86.45** | **86.40** | **86.63** | 86.66 | 86.70 |
>
> > Remark 3.1 is not fully convincing to me. Can the authors explain more in detail why different ranges of expert scores are supposed to be undesirable?
> >
>
> We would like to clarify the following points:
>
> - In top-*k* gating, the gating scores are computed using the softmax function, which allows for the comparison of gating scores among different experts. Specifically, a higher score for some experts results in a lower score for others.
> - In contrast, in the top-any gating mechanism of DynMoE, the gating scores for different experts are calculated independently using sigmoid function (see figure 2 for a more intuitive understanding). Consequently, these scores are not comparable among experts. As such, we believe that the range of expert scores does not accurately reflect their importance.
> - Additionally, we have conducted an ablation study to demonstrate the effectiveness of this design choice. Results show that using different expert scores significantly reduce the model performance.
>
> | StableLM | VQAv2 | GQA | VizWiz | SQA | TextVQA | POPE | MME | MMBench |
> | --- | --- | --- | --- | --- | --- | --- | --- | --- |
> | ours | **77.4** | **61.4** | **40.6** | **63.4** | **48.9** | **85.7** | **1300.9** | **63.2** |
> | weighted scores | 73.9 | 57.4 | 32.1 | 61.3 | 46.9 | 84.2 | 1176.8 | 52.1 |

---

> ### Author Response · Authors · 2024-11-22
> **Reply to Reviewer 4q9o (5/5)**
>
> > Is the MoE implementation used in this work dropless [6], i.e. does it drop tokens if the predefined capacity of an expert is exceeded? If it is not dropless, how often are tokens dropped and how does DynMoE compare to standard MoE in terms of load balance?
> >
>
> Thank you for the comments. We would like to provide the following clarification
>
> 1. We use a dropless implementation, and the capacity are adaptively calculated. In, detail the capacity is calculated by $c = max_{k}{N_k}$, where $N_k$ is the number of tokens that choose expert $k$.
> 2. In our submission, we did not enforce an equal load distribution among experts based on the following considerations:
>     1. The significance of experts varies, with some being more likely to be activated by all tokens. This issue is recognized in recent MoE studies [9] and is also observed in our research, as discussed in Section 4.5.
>     2. Enforcing load balance by including a load-balance loss might not align with our requirements for sparse routing, as activating all experts could be considered a form of load balance.
>
>     We believe that the load balance issue can be addressed through engineering techniques, such as allocating additional computation units to shared experts.
>
> 3. To further illustrate our point, we have conducted additional ablation studies on the MoE-LLaVA setting. Specifically, we have: (1) Included a load-balance loss in our model; (2) Introduced an efficiency loss to enforce sparsity, as suggested in [2]. We report the following metrics:
>     1. **Performance (Table 1):** The performance of different settings.
>     2. **Load Balance  (Table 2):** The frequency with which each expert is activated, calculated as (expert activation time / total token count).
>     3. **Efficiency  (Table 3):** The top-k values per layer.
>     4. **Efficiency  (Table 4):** The top-k activation frequency, calculated as (number of tokens that activate k experts / total tokens).
>
>     We can find that
>
>     1. Although it does not explicitly enforce load balancing, the original DynMoE achieves load balancing comparable to that of the standard top-2 MoE (Table 2).
>     2. Adding the load balance loss slightly decreases the performance of DynMoE (Table 1) while increasing the number of activated experts (Table 3). However, it improves load balancing (Table 2).
>     3. Adding an additional efficiency loss on top of the load balance loss improves performance (Table 1) and helps overcome some extreme cases, such as the reduction of the top-$k$ values in the bottom layer from 2.88 to 2.02 (Table 3), and reduce the number of tokens that activate all 4 experts (Table 4). Moreover, the efficiency loss further enhances load balancing (Table 2).
>
> | StableLM | VQAv2 | GQA | VizWiz | SQA | TextVQA | POPE | MME | MMBench |
> | --- | --- | --- | --- | --- | --- | --- | --- | --- |
> | DynMoE | 77.4 | 61.4 | 40.6 | 63.4 | 48.9 | 85.7 | 1300.9 | 63.2 |
> | DynMoE + load balance | 77.1 | 61.6 | 37.0 | 61.4 | 50.3 | 85.3 | 1313.5 | 61.7 |
> | DynMoE + load balance + efficiency | 77.1 | 61.8 | 39.4 | 62.9 | 49.7 | 85.4 | 1321.2 | 61.9 |
>
> | Activation Frequency per Expert (VQAv2, layer 0) | Expert 1 | Expert 2 | Expert 3 | Expert 4 |
> | --- | --- | --- | --- | --- |
> | MoE (top-2) | 0.36 | 1.29 | 0.16 | 0.19 |
> | DynMoE | 0.29 | 0.97 | 0.48 | 0.35 |
> | DynMoE + load balance | 0.81 | 0.50 | 0.90 | 0.68 |
> | DynMoE + load balance + efficiency | 0.45 | 0.52 | 0.42 | 0.63 |
>
> | Top-k per Layer (VQAv2) | Layer 0 | Layer 2 | Layer 4 | Layer 6 | Layer 8 | Layer 10 | Layer 12 | Layer 14 | Layer 16 | Layer 18 | Layer 20 | Layer 22 |
> | --- | --- | --- | --- | --- | --- | --- | --- | --- | --- | --- | --- | --- |
> | DynMoE | 2.09 | 1.07 | 1.57 | 1.06 | 2.04 | 1.03 | 1.03 | 1.00 | 1.03 | 1.02 | 1.02 | 1.00 |
> | DynMoE + load balance | 2.88 | 1.25 | 1.59 | 1.27 | 1.26 | 1.13 | 1.77 | 1.70 | 1.12 | 1.33 | 1.30 | 1.00 |
> | DynMoE + load balance + efficiency | 2.02 | 1.25 | 1.81 | 1.57 | 1.65 | 1.20 | 1.47 | 2.30 | 1.07 | 1.37 | 1.82 | 1.00 |
>
> | Top-k Frequency (VQAv2) | Top-1 | Top-2 | Top-3 | Top-4 |
> | --- | --- | --- | --- | --- |
> | DynMoE | 0.79 | 0.16 | 0.04 | 0.01 |
> | DynMoE + load balance | 0.65  | 0.26  | 0.06  | 0.03 |
> | DynMoE + load balance + efficiency | 0.58 | 0.32 | 0.09 | 0.01 |
>
> [9] DeepSeekMoE: Towards Ultimate Expert Specialization in Mixture-of-Experts Language Models.
>
> [10] Mistral 7B.
>
> [11] Moe-llava: Mixture of experts for large vision-language models.
>
> [12] Emergent Mixture-of-Experts: Can Dense Pre-trained Transformers Benefit from Emergent Modular Structures?

---

> > ### Author Response · Authors · 2024-11-24
> >
> > Dear Reviewer 4q9o,
> >
> > We update the the experiment results of training from scratch on CIFAR10. DynMoE (Original), DynMoE (Average), and DynMoE (W-Average) not only achieve faster convergence rate, but also achieve higher final accuracy than standard MoE.
> >
> > | CIFAR10 (ViT-S, 2 MoE Layer, Acc per 100 Epoch) | 200       | 300       | 400       | 500       |
> > | ----------------------------------------------- | --------- | --------- | --------- | --------- |
> > | MoE (K = 8, k = 1)                              | 72.66     | 77.51     | 80.10     | 81.08     |
> > | MoE (K = 8, k = 2)                              | **73.79** | **78.50**     | **80.85**     | **81.91**     |
> > | MoE (K = 8, k = 4)                              | 72.77     | 77.84     | 80.30     | 81.14     |
> > | MoE (K = 8, k = 8)                              | 70.68     | 75.32     | 78.28     | 79.11     |
> > | DynMoE (Original, avg topk=7)                   | **74.84** | **79.24** | **81.77** | **82.50** |
> > | DynMoE (Average, avg topk=7)                    | **74.70** | **80.32** | **82.51** | **83.57** |
> > | DynMoE (W-Average, avg topk=6.5)                | **74.16** | **78.77** | **81.30** | **82.01** |
> > | DynMoE (Most activated, avg topk=6.5)           | 71.71     | 77.56     | 80.08     | 80.58     |
> >
> > We hope our responses have sufficiently addressed your concerns. Please do not hesitate to reach out if you have any further questions. Thank you again for your time and effort in reviewing our paper.

---

> > > ### Author Response · Authors · 2024-11-25
> > > **A Kind Reminder for Reviewer 4q9o**
> > >
> > > Dear Reviewer 4q9o,
> > >
> > > Thank you for your valuable feedback and thorough review of our paper. Your insights have been instrumental in helping us refine and improve our work. In response to the concerns you raised, **we have provided detailed explanations and new experiments to address each point comprehensively**. Below, we summarize our key responses to your concerns:
> > >
> > > - **Clarification on our contribution:** We discussed the related works [1,2,3,4] and demonstrated that these papers do not address the specific issues we tackle in this paper.
> > > - **Added comparison to existing MoE methods in the literature:** We argue that a direct comparison to [1,2] is unfair due to the dense training requirements of these methods. Additionally, we have included comparisons with the top-p gating baseline [3,4].
> > > - **Added more latency / throughput evaluation:** We have provided more details on our evaluation setup and demonstrated that our comparison between DynMoE and standard MoE is fair. We also included an efficiency evaluation of dense models to offer a more complete picture.
> > > - **Conducted from-scratch training experiments:** We conducted from-scratch training experiments on ViT using CIFAR-10 and showed that DynMoE achieves better performance and converges faster than standard MoE baselines.
> > > - **Reported standard deviations:** We have now included the standard deviations for our language experiments and clarified that LLM experiments typically do not report this information due to the high computational cost.
> > > - **Fixed readability issues:** We have revised the paper in accordance with your suggestions to improve readability.
> > > - **Added weights of auxiliary losses:** We have added the weights used for the auxiliary losses.
> > > - **Explained our record routing design:** We clarified that the decision to begin record routing at 1/3 of the epoch and end it at 2/3 was made to stabilize training.
> > > - **Discussed the further extension to non-linear routers:** We discussed how top-any gating can still be effective with a non-linear router, though we also noted potential challenges with the adaptive training process.
> > > - **Explained and conducted more ablation studies on initialization of experts:** We elaborated on our expert initialization process and included three ablation studies to evaluate different expert initialization strategies.
> > > - **Discussed on Remark 3.1:** We have further clarified the rationale for not using weighted scores and added an ablation study to demonstrate the necessity of this design choice.
> > > - **Elaborated load balancing and dropless design:** We clarified that DynMoE is dropless and compared its load balancing performance with that of standard MoE. Additionally, we incorporated experiments that include a load balancing loss and efficiency loss to enhance DynMoE's load balancing.
> > >
> > > Once again, thank you for your time and effort in reviewing our paper. Your feedback has been invaluable in improving the quality of our manuscript.
> > >
> > > As the rebuttal period draws to a close, we believe we have fully addressed all of your concerns and kindly request that you reconsider your score. Please do not hesitate to reach out if you have any further questions or comments.

---

> > > > ### Comment · Reviewer_4q9o · 2024-11-25
> > > >
> > > > Thank you for the broad response. Some of my concerns have been addressed. In particular, I appreciate the additional dynamic-k baseline, the added standard deviation values, and the answers to the questions section. I have adjusted the score so that it reflects the improvements that the authors have made.
> > > >
> > > > However, I am still leaning towards the rejection of this work. I am not convinced by the authors' response that the throughput evaluation is fair.
> > > >
> > > > > We apologize for any previous ambiguity in our statement. However, we stand by our claim that we conducted a fair evaluation.
> > > > > ...
> > > > > MoE-LLaVA utilizes the Deepspeed-MoE implementation. However, despite its efficiency, the Deepspeed-MoE implementation for step (2) is limited to supporting only top-1 and top-2 gating. Consequently, we developed our own implementation for step (2), which resulted in some efficiency degradation due to additional engineering efforts, i.e., we did not incorporate a CUDA implementation for parallel computation.
> > > > > ...
> > > > >In summary, we believe we have presented a fair comparison.
> > > >
> > > > Simply stating that it is fair does not automatically make it fair. The authors do not actually compare with any MoE implementation used in practice, and instead compare only to the implementation that they themselves wrote. Obviously, this kind of evaluation may favor their method, and the method might be still slow in practice.
> > > >
> > > > In line 76 of the manuscript, the authors state: "We provide Tutel and DeepSpeed-MoE implementations for ease of practical usage." This statement may be misleading without clear evidence of the implementations' performance. To assess practicality, it is essential to know how fast these implementations are compared to state-of-the-art top-k MoE models and their optimized implementations. Without such comparisons, it is difficult to determine whether the proposed method truly offers practical advantages.
> > > >
> > > > Some implementation is provided, but the reader will never know if the move from top-k MoEs is worth it. Perhaps the introduction of dynamic-k will always make MoEs slower on GPUs? How large is this overhead? Can it be alleviated with a dedicated low-level implementation? **It is the responsibility of the authors to show that their proposed contribution can be fast in practice** to support their claims. If the only way to do this is through engineering effort (writing e.g. CUDA kernels), then the authors should do that. Otherwise, the claims about "higher throughput compared to MoE-LLaVA" (line 480 in the manuscript) and about practicality of their method are unfounded.
> > > >
> > > > In the end the **FLOPs result do not matter, and the only metric the audience should be looking at is throughput/latency/wall-clock times**. The authors themselves emphasize the practical aspect of their contribution -- removing the need for tuning hyperparameters and thus reducing waste of computational resources. Yet, they **fail to show that their method provides this in practice**. A slow implementation would affect both training and inference, negating any advantages of their method.
> > > >
> > > > > We have provided the throughput of dense models below.
> > > > > ...
> > > >
> > > > Thank you for these results. However, it is still hard to ascertain the overhead introduced by your modification of Deepspeed-MoE implementation. Can you at least provide the throughput results for the MoE-LLaVA but with the original Deepspeed-MoE implementation for k=1 and k=2?
> > > >
> > > > > We trained ViT-S on the CIFAR-10 dataset from scratch, and train the models for 200 epochs due to the time limitation.
> > > >
> > > > A minor concern (that did not affect my rating) is that the results for training from scratch on CIFAR-10 are not really convincing, as findings from such small dataset are often not transferable to larger datasets. Moreover, the average top-k values for DynMoE seem unusually high -- almost the entire model is executed for DynMoE-LLaVA. The authors should consider repeating this experiment on a larger dataset for the next revision.

---

> > > > > ### Author Response · Authors · 2024-11-26
> > > > > **Further Reply to Reviewer 4q9o (1/2)**
> > > > >
> > > > > Thank you for your further response!
> > > > >
> > > > > We agree with the reviewer that the throughput of the original DeepSpeed implementation should be reported, and a fast implementation of DynMoE is essential. To this end, we optimized the top-any gating mechanism of DynMoE:
> > > > >
> > > > > - **Old Implementation**: This extends the standard top-2 gating to top-k gating, allowing it to handle both standard top-k MoE and top-any gating in DynMoE, but with lower efficiency.
> > > > > - **New Implementation**: Specifically designed for DynMoE, this version is more efficient.
> > > > >
> > > > > We provide the code for both the old and new top-any gating implementations below. For a thorough evaluation, we present the following metrics:
> > > > >
> > > > > - Model Throughput
> > > > > - First Token Latency
> > > > > - Wall-Clock Time
> > > > > - Wall-Clock Time for router, gating, and expert passing
> > > > >
> > > > > We report the performance of MoE-LLaVA using the standard DeepSpeed-MoE (top-2) implementation, as well as DynMoE with both the old and new top-any gating implementations. The results show that:
> > > > >
> > > > > - DynMoE (New) achieves higher throughput, lower latency, and reduced wall-clock time compared to DeepSpeed-MoE, indicating improved efficiency.
> > > > > - The top-any gating introduces additional cost in the router, but the gating and expert passing steps are more efficient than in DeepSpeed-MoE.
> > > > > - The new top-any gating implementation reduces costs by 49% compared to the old top-any gating, and by 34% compared to DeepSpeed-MoE.
> > > > >
> > > > > | Metric | Throughput (token / second)$\uparrow$ | First Token Latency (ms)$\downarrow$ | Wall-clock Time (second / sample)$\downarrow$ |
> > > > > | --- | --- | --- | --- |
> > > > > | DeepSpeed-MoE (top-2) | 27 | 137 | 6.2 |
> > > > > | DynMoE (Old) | 26 | 357 | 6.6 |
> > > > > | DynMoE (New) | **30** | **124** | **5.7** |
> > > > >
> > > > > | Time (per sample and MoE layer) | Routing (ms) $\downarrow$ | Gating (ms)$\downarrow$ | Experts Passing (ms)$\downarrow$ | Total (ms)$\downarrow$ |
> > > > > | --- | --- | --- | --- | --- |
> > > > > | DeepSpeed-MoE (top-2) | **0.04** | 1.23 | 1.30 | 2.57 |
> > > > > | DynMoE (Old) | 0.52 | 1.59 | **1.18** | 3.29 |
> > > > > | DynMoE (New) | 0.52 | **0.81** | **1.17** | **2.50** |
> > > > >
> > > > > > Old implementation:
> > > > >
> > > > > ```python
> > > > > def topanygating(logits: Tensor, capacity_factor: float, min_capacity: int, K: Tensor, gate_tensor=None, expert_mask=None, ep_group=None) -> Tuple[Tensor, Tensor, Tensor, Tensor]:
> > > > >
> > > > >     gates = logits
> > > > >     max_K = max(K)
> > > > >     max_K = max(max_K, 1)
> > > > >
> > > > >     # Create a mask for k-st's expert per token
> > > > >     logits_except_pre = logits + 0.0
> > > > >     masks = []
> > > > >     num_experts = int(gates.shape[1])
> > > > >
> > > > >     for k in range(max_K):
> > > > >         indicesk_s = torch.argmax(logits_except_pre, dim=1)
> > > > >         mask_k = F.one_hot(indicesk_s, num_classes=num_experts)
> > > > >         mask_k = (mask_k.T * (K > k)).T
> > > > >         masks.append(mask_k)
> > > > >         logits_except_pre = logits_except_pre.masked_fill(mask_k.bool(), float("-inf"))
> > > > >
> > > > >     # Compute locations in capacity buffer
> > > > >     locations = []
> > > > >     pre_locations = torch.zeros(num_experts).to(masks[0].device)
> > > > >
> > > > >     for mask in masks:
> > > > >         locationsk = torch.cumsum(mask, dim=0) - 1
> > > > >         locationsk += pre_locations.int()
> > > > >         pre_locations += torch.sum(mask, dim=0)
> > > > >         locations.append(locationsk)
> > > > >
> > > > >     new_capacity = torch.max(pre_locations).to(logits.device).int()
> > > > >     dist.all_reduce(new_capacity, op=dist.ReduceOp.MAX, group=dist.get_world_group())
> > > > >     capacity = new_capacity
> > > > >
> > > > >     # gating decisions -- not sure the meaning now
> > > > >     exp_counts = torch.sum(masks[0], dim=0).detach().to('cpu')
> > > > >
> > > > >     # Compute l_aux
> > > > >     if gate_tensor is None or expert_mask is None:
> > > > >         me = torch.mean(gates, dim=0)
> > > > >         ce = torch.mean(masks[0].float(), dim=0)
> > > > >         l_aux = torch.mean(me * ce) * num_experts * num_experts
> > > > >     else:
> > > > >         l_aux = diverse_and_simple_gate_loss(gate_tensor, expert_mask)
> > > > >
> > > > >     # Remove locations outside capacity from mask <--- Dropless, can optimize
> > > > >     for i in range(len(masks)):
> > > > >         masks[i] *= torch.lt(locations[i], capacity)
> > > > >
> > > > >     # Store the capacity location for each token
> > > > >     locations_s = [torch.sum(locations[i] * masks[i], dim=1) for i in range(len(masks))]
> > > > >
> > > > >     # Normalize gate probabilities
> > > > >     masks_float = [mask.float() for mask in masks]
> > > > >     gates_s = [einsum("se,se->s", gates, mask_float) for mask_float in masks_float]
> > > > >     denom_s = sum(gates_s)
> > > > >
> > > > >     # Avoid divide-by-zero
> > > > >     denom_s = torch.clamp(denom_s, min=torch.finfo(denom_s.dtype).eps)
> > > > >     gates_s = [gate_s / denom_s for gate_s in gates_s]
> > > > >
> > > > >     combine_weights = torch.zeros_like(gates).unsqueeze(-1).expand(-1, -1, capacity.item())
> > > > >
> > > > >     # Calculate combine_weights within the loop
> > > > >     for gatesk_s, maskk_float, locationsk_s in zip(gates_s, masks_float, locations_s):
> > > > >         gatessk = torch.einsum("s,se->se", gatesk_s, maskk_float)
> > > > >         locationsk_sc = _one_hot_to_float(locationsk_s, capacity)
> > > > >         combinesk_sec = torch.einsum("se,sc->sec", gatessk, locationsk_sc)
> > > > >         combine_weights = combine_weights.add(combinesk_sec)
> > > > >
> > > > >     dispatch_mask = combine_weights.bool()
> > > > >
> > > > >     return l_aux, combine_weights, dispatch_mask, exp_counts
> > > > > ```

---

> > > > > ### Author Response · Authors · 2024-11-26
> > > > > **Further Reply to Reviewer 4q9o (2/2)**
> > > > >
> > > > > >New implementation
> > > > >
> > > > > ```python
> > > > > def topanygating_opt(logits: Tensor, capacity_factor: float, min_capacity: int, K: Tensor, gate_tensor=None, expert_mask=None, ep_group=None) -> Tuple[Tensor, Tensor, Tensor, Tensor]:
> > > > >
> > > > >     """Implements TopanyGating on logits."""
> > > > >     gates = logits
> > > > >     mask = gates.int()
> > > > >     exp_counts = torch.sum(mask, dim=0).detach().to('cpu')
> > > > >
> > > > >     new_capacity = torch.max(exp_counts).to(logits.device)
> > > > >     dist.all_reduce(new_capacity, op=dist.ReduceOp.MAX, group=dist.get_world_group())
> > > > >     capacity = new_capacity
> > > > >
> > > > >     num_experts = int(gates.shape[1])
> > > > >
> > > > >     # Compute l_aux
> > > > >     if gate_tensor is None or expert_mask is None:
> > > > >         me = torch.mean(gates, dim=0)
> > > > >         ce = torch.mean(mask.float(), dim=0)
> > > > >         l_aux = torch.mean(me * ce) * num_experts * num_experts
> > > > >     else:
> > > > >         l_aux = diverse_and_simple_gate_loss(gate_tensor, expert_mask)
> > > > >
> > > > >     # Store the capacity location for each token
> > > > >     locations1 = torch.cumsum(mask, dim=0) # sample * expert
> > > > >     locations1_s = ((locations1 * mask) - 1 ) % (capacity + 1) # sample * expert, mod `capacity + 1` to keep indices positive
> > > > >
> > > > >     gates = gates / K.unsqueeze(1)
> > > > >
> > > > >     locations1_sc = _one_hot_to_float(locations1_s, capacity + 1) # (sample, expert, capacity + 1)
> > > > >     combine_weights = einsum("se,sec->sec", gates, locations1_sc[:,:,:-1]) # (sample, expert, capacity)
> > > > >
> > > > >     dispatch_mask = combine_weights.bool()
> > > > >
> > > > >     return l_aux, combine_weights, dispatch_mask, exp_counts
> > > > > ```

---

> > > > > > ### Comment · Reviewer_4q9o · 2024-11-27
> > > > > >
> > > > > > I think the **throughput results that the authors present are misleading**. I have explicitly asked for throughput of the
> > > > > > original Deepspeed-MoE implementation for $k=1$ and $k=2$, and yet the authors provide the results for $k=2$ only. It
> > > > > > appears that DeepSpeed uses a naive implementation when $k$ is set to 2, as opposed to the Tutel CUDA kernels:
> > > > > > [source](https://github.com/microsoft/DeepSpeed/blob/v0.9.5/deepspeed/moe/sharded_moe.py#L458)
> > > > > > Given that this implementation is also in plain PyTorch, it is not surprising that the new implementation provided
> > > > > > by authors is faster. In my opinion, both DeepSpeed-MoE top-$k=2$ and the new implementation of the authors
> > > > > > are highly inefficient.
> > > > > >
> > > > > > As such, the current comparison is pointless. For a proper comparison the authors should compare throughput/latency to
> > > > > > a state-of-the-art implementation of MoE. Note that [1] demonstrate significant improvements over Tutel, and [2]
> > > > > > propose a method that improves upon [2].
> > > > > >
> > > > > > Can the authors clarify the following aspects of the benchmarking results:
> > > > > > - What was the batch size and sequence length (number of tokens) used for measuring latency/throughput?
> > > > > > - What GPU type was used for measuring latency/throughput?
> > > > > > - What was the number of tries that the result of measuring latency/throughput was averaged over?
> > > > > > - During measuring some first iterations of the model's inference should be skipped in order not to
> > > > > > measure unnecessary memory allocations, loading of kernels etc. Was this done before measuring latency/throughput?
> > > > > >
> > > > > > Finally, can the authors also answer the following:
> > > > > > - What was the training (wall clock) time of every model used in the paper?
> > > > > > - What was the total training FLOPs cost of dense models from Table 2?
> > > > > > - Which dense model from Table 2 is the one from Table 4?
> > > > > > (In general, it is always best to report both performance and efficiency in the same tables or even plots, as we
> > > > > > always care about both)
> > > > > >
> > > > > >
> > > > > > *** References ***
> > > > > >
> > > > > > [1] Gale, Trevor, et al. "Megablocks: Efficient sparse training with mixture-of-experts." Proceedings of Machine Learning and Systems 5 (2023): 288-304.
> > > > > >
> > > > > > [2] Tan, Shawn, et al. "Scattered Mixture-of-Experts Implementation." arXiv preprint arXiv:2403.08245 (2024).

---

> > > > > > > ### Author Response · Authors · 2024-11-29
> > > > > > > **Reply to Reviewer 4q9o (1/2)**
> > > > > > >
> > > > > > > Dear reviewer 4q9o,
> > > > > > >
> > > > > > > Thank you for your prompt response. Upon reviewing the feedback, we noticed that the requests in the [initial review](https://openreview.net/forum?id=T26f9z2rEe&noteId=hu5Z3lFq2F) and [the first response](https://openreview.net/forum?id=T26f9z2rEe&noteId=vLPAzRHcCG) appear to differ from those in [the second (current) response](https://openreview.net/forum?id=T26f9z2rEe&noteId=Ql4smFs4KH). Specifically:
> > > > > > >
> > > > > > > ### 1. **Old Request: Make a fair comparison to MoE-LLaVA**
> > > > > > >
> > > > > > > - As the original MoE-LLaVA uses the DeepSpeed-MoE (top-2) implementation, **we believe we have already provided a fair comparison to MoE-LLaVA in our [previous response](https://openreview.net/forum?id=T26f9z2rEe&noteId=ZO8CvARjdD).**
> > > > > > > - Additionally, we found that the term “DeepSpeed-MoE” in lines 485-508 of the revised paper was somewhat unclear. To address this, we have revised the language to refer to "MoE-LLaVA" for greater clarity.
> > > > > > > - We also provided results using MoE-LLaVA with the DeepSpeed-MoE (top-1 gating) implementation. It need to be pointed out that since MoE-LLaVA was originally trained using the top-2 gating, there was no checkpoint trained using top-1 gating, so we enforced top-1 gating for the top-2 MoE-LLaVA. **While this approach could be used to benchmark efficiency, it will destroy model performance, shown in the table below.** Based on this, we observe that **DynMoE offers an additional benefit compare to standard top-k MoE: it is robust to variations in the test-time top-$k$ value.** As demonstrated in the table, DynMoE’s performance degrades less than MoE-LLaVA's when using top-1 gating at test time.
> > > > > > >
> > > > > > > ### 2. **New Request: Compare to SOTA MoE implementations with optimized CUDA kernel**
> > > > > > >
> > > > > > > - We would like to clarify that **the primary focus of this paper is on designing an auto-tuning MoE approach**, rather than providing an optimal MoE kernel implementation. While we agree with the reviewer that implementing an optimized CUDA kernel for MoE, similar to Megablocks, would be a valuable contribution, it falls outside the scope of this submission and could be addressed in future work.
> > > > > > > - Furthermore, we believe the pure Pytorch approach we used is consistent with the common implementation style seen in many recent conference papers on MoE [1-4] and popular open-source MoE models [5-7].
> > > > > > >
> > > > > > > |  | Throughput (Skip first sample) | Latency(Skip first sample) | Wall-clock Time(Skip first sample) | Throughput (Skip first 10 samples) | Latency(Skip first 10 samples) | Wall-clock Time(Skip first 10 samples) | VQAv2 | GQA | VizWiz | SQA | TextVQA | POPE | MME | MMBench |
> > > > > > > | --- | --- | --- | --- | --- | --- | --- | --- | --- | --- | --- | --- | --- | --- | --- |
> > > > > > > | Dense-LLaVA (1.6B) | 32 | 72 | 4.7 | 35 | 70 | 4.6 | - | - | - | - | - | - | - | - |
> > > > > > > | MoE-LLaVA (enforced top-1, 1.6Bx4) | 35 | 94 | 4.8 | 35 | 95 | 4.9 | 16.1 | 7.9 |  2.3 | 2.0 | 2.1 | 67.2 | 165.1 | 1.5 |
> > > > > > > | DynMoE (enforced top-1, 1.6Bx4) | 29 | 110 | 5.6 | 30 | 112 | 5.5 | 74.7 | 59.7 | 29.8 | 53.5 | 42.9 | 75.0 | 1191.6 | 53.1 |
> > > > > > > | MoE-LLaVA (top-2, 1.6Bx4) | 27 | 137 | 6.2 | 27 | 136 | 6.3 | 76.7  | 60.3 | 36.2 | 62.6 | 50.1 | 85.7 | 1318.2 | 60.2 |
> > > > > > > | DynMoE (top-any, 1.6Bx4) | 30 | 124 | 5.7 | 29 | 124 | 5.6 | 77.4 | 61.4 | 40.6 | 63.4 | 48.9 | 85.7 | 1300.9 | 63.2 |
> > > > > > >
> > > > > > > > Address Questions about benchmark
> > > > > > > >
> > > > > > > - “What was the batch size and sequence length (number of tokens) used for measuring latency/throughput?”
> > > > > > >     - We use the [official evaluation code](https://github.com/PKU-YuanGroup/MoE-LLaVA/blob/0549ce0e65119858399d2e4e88ddb4cd3db4c133/moellava/eval/model_vqa.py#L48) provided by MoE-LLaVA, which only supports batch size = 1
> > > > > > >     - average sequence length (number of tokens) = 625
> > > > > > >     - We also report the evaluation time on ImageNet using ViT-S with a batch size of 256, utilizing 4 A100 GPUs, as shown in the table below. The evaluation time of DynMoE falls between that of DeepSpeed-MoE (top-1) and DeepSpeed-MoE (top-2).
> > > > > > > - “What GPU type was used for measuring latency/throughput?”
> > > > > > >     - A100, same as stated in the caption of Table 4.
> > > > > > > - “What was the number of tries that the result of measuring latency/throughput was averaged over?”
> > > > > > >     - We take the average of 5 tries.
> > > > > > > - “During measuring some first iterations of the model's inference should be skipped in order not to measure unnecessary memory allocations, loading of kernels etc. Was this done before measuring latency/throughput?”
> > > > > > >     - Based on our observations, the first sample takes longer to process than the subsequent samples. As a result, we have chosen to skip the first sample during evaluation.
> > > > > > >     - To further clarify this, we have included the results of skipping the first 10 samples, but the results are only slightly changed.

---

> > > > > > > ### Author Response · Authors · 2024-11-29
> > > > > > > **Reply to Reviewer 4q9o (2/2)**
> > > > > > >
> > > > > > > > Address Questions
> > > > > > > >
> > > > > > > - “What was the training (wall clock) time of every model used in the paper?”
> > > > > > >
> > > > > > > Thank you for your suggestion. Since re-training the DomainBed and BERT experiments is time-consuming, we apologize for being unable to provide the exact training times for these two experiments. However, to provide more transparency, we have included the training times for the following two experiments:
> > > > > > >
> > > > > > > - MoE-LLaVA experiments. evaluation window: 1000 steps (batches).
> > > > > > > - Pretraining ViT-S on the ImageNet 224$\times$224 dataset. This experiment is still ongoing, so we are reporting the average training time per batch up until now (80 epochs done).
> > > > > > >
> > > > > > > | Time (s/batch) | MoE-LLaVA (Train, 4 A100, batch size=32) | ViT-S, ImageNet (Train, 8 A100, batch size=512) | ViT-S, ImageNet (Test, 4 A100, batch size=256) |
> > > > > > > | --- | --- | --- | --- |
> > > > > > > | Top-1 MoE | 1.31 | 0.39 | 0.18 |
> > > > > > > | Top-2 MoE | 1.60 | 0.63 | 0.32 |
> > > > > > > | DynMoE | 1.48 | 0.51 | 0.27 |
> > > > > > > - “What was the total training FLOPs cost of dense models from Table 2?”
> > > > > > >     - We would like to clarify that we did not train these models ourselves. Instead, we refer to them to demonstrate that sparse 2×4B models can achieve performance comparable to that of 7B dense models. The performance of the dense models are taken from the officially reported results in the original papers.
> > > > > > >     - Additionally, papers on vision LLMs usually avoid training baseline models in-house, primarily due to the substantial computational costs involved [8, 9, 10].
> > > > > > > - “Which dense model from Table 2 is the one from Table 4? (In general, it is always best to report both performance and efficiency in the same tables or even plots, as we always care about both)”
> > > > > > >     - We apologize for the misunderstanding and would like to clarify that the dense model referenced in Table 4 is StableLM-1.6B.
> > > > > > >     - The MoE-LLaVA has three training stages, and this pretrained dense model is derived from the stage-2 training of MoE-LLaVA and is used to initialize both MoE-LLaVA and DynMoE for stage-3 training.
> > > > > > >     - As a result, this dense model cannot be fairly compared to MoE-LLaVA models due to its lack of stage-3 training. For this reason, we only report its efficiency in Table 4, and not its performance.
> > > > > > >
> > > > > > > We hope this clarifies our position and the focus of our submission. Thank you again for your feedback.
> > > > > > >
> > > > > > > [1] Harder Tasks Need More Experts: Dynamic Routing in MoE Models. ACL 2024.
> > > > > > >
> > > > > > > [2] Sparse mixture-of-experts are domain generalizable learners. ICLR 2023.
> > > > > > >
> > > > > > > [3] XMoE: Sparse Models with Fine-grained and Adaptive Expert Selection. ACL 2024 Findings.
> > > > > > >
> > > > > > > [4] Unlocking Emergent Modularity in Large Language Models. NAACL2024.
> > > > > > >
> > > > > > > [5] [Phi-3.5-MoE](https://huggingface.co/microsoft/Phi-3.5-MoE-instruct/blob/main/modeling_phimoe.py).
> > > > > > >
> > > > > > > [6] [Qwen2-MoE](https://github.com/huggingface/transformers/blob/main/src/transformers/models/qwen2_moe/modeling_qwen2_moe.py)
> > > > > > >
> > > > > > > [7] [Deepseek-MoE](https://huggingface.co/deepseek-ai/deepseek-moe-16b-chat/blob/main/modeling_deepseek.py)
> > > > > > >
> > > > > > > [8] Visual instruction tuning. NeurIPS 2023.
> > > > > > >
> > > > > > > [9] Qwen-VL: A Versatile Vision-Language Model for Understanding, Localization, Text Reading, and Beyond. Technical Report 2023.
> > > > > > >
> > > > > > > [10] Chameleon: Plug-and-Play Compositional Reasoning with Large Language Models. NeurIPS 2024.

---

> > > > > > > > ### Comment · Reviewer_4q9o · 2024-11-29
> > > > > > > >
> > > > > > > > > Upon reviewing the feedback, we noticed that the requests in the initial review and the first response appear to differ from those in the second (current) response.
> > > > > > > > > ...
> > > > > > > > > New Request: Compare to SOTA MoE implementations with optimized CUDA kernel
> > > > > > > >
> > > > > > > > It seems the authors are trying to make it appear as if I raise new issues late in the discussion period. I do not see much evidence to support this claim, as the concern about slow performance of the proposed method was present both in my original review:
> > > > > > > > > A major flaw of the work is that authors only briefly mention the actual speedup that the method supposedly provides. ... The readers should be aware of any overhead stemming from the introduction of top-any routing. I would at least expect a non-negligible overhead stemming from the non-constant number of token-expert executions.
> > > > > > > >
> > > > > > > > and in my first response:
> > > > > > > > > To assess practicality, it is essential to know how fast these implementations are compared to state-of-the-art top-k MoE models and their optimized implementations. Without such comparisons, it is difficult to determine whether the proposed method truly offers practical advantages.
> > > > > > > >
> > > > > > > > While I understand the frustration of the authors, I merely highlighted a potential weakness of the proposed method, one that has serious implications on its practical usefulness. Please note that it is my responsibility as a reviewer.
> > > > > > > >
> > > > > > > > > Based on this, we observe that DynMoE offers an additional benefit compare to standard top-k MoE: it is robust to variations in the test-time top- value. As demonstrated in the table, DynMoE’s performance degrades less than MoE-LLaVA's when using top-1 gating at test time.
> > > > > > > >
> > > > > > > > I would advise caution in making claims based on such preliminary experiments. These results are not surprising as the average k that is reported in the manuscript for the proposed model is much closer to 1 when compared to the $k=2$ of the top-k model.
> > > > > > > >
> > > > > > > > > We would like to clarify that the primary focus of this paper is on designing an auto-tuning MoE approach, rather than providing an optimal MoE kernel implementation. While we agree with the reviewer that implementing an optimized CUDA kernel for MoE, similar to Megablocks, would be a valuable contribution, it falls outside the scope of this submission and could be addressed in future work.
> > > > > > > >
> > > > > > > > Again I would like to emphasize that the proposed per-token dynamic top-k mechanism prevents us from using the existing efficient implementations of MoE layers (to the best of my knowledge, all of them assume some form of top-k). Moreover, it is not clear to the reader whether the introduced dynamic mechanism can be efficiently implemented for GPUs without overhead at all (e.g. perhaps the dynamic top-k mechanism can be GPU cache-unfriendly?). These issues affect the practical usefulness of the proposed method, and as such I do not entirely agree that it falls outside the scope of this paper. I point out that the authors chose to report throughput of the models in the original submission, so the authors themselves must be aware of the significance of this aspect.
> > > > > > > >
> > > > > > > > > (table)
> > > > > > > > > MoE-LLaVA (enforced top-1, 1.6Bx4)
> > > > > > > > > DynMoE (enforced top-1, 1.6Bx4)
> > > > > > > >
> > > > > > > > When looking at these two rows, the overhead in the throughput/latency/wall-clock time is visible now. This is a significantly more impartial result that shows the limitations of your current implementation. I believe the difference would be higher if one of the listed state-of-the-art MoE implementations was used for MoE-LLaVA.
> > > > > > > >
> > > > > > > > If the authors really want to leave this aspect for future work, then these weaknesses should be clearly disclosed to the readers in the limitations section, and the implementations used for each model should be clearly listed in the throughput/latency results.

---

> > > > > ### Author Response · Authors · 2024-12-02
> > > > > **Results on ImageNet**
> > > > >
> > > > > Dear reviewer 4q9o,
> > > > >
> > > > > Thank you for raising the score. Your detailed and insightful review has been invaluable in helping us improve both the quality and clarity of our paper. Below, we present the results on ImageNet 224$\times$224 as promised.
> > > > >
> > > > > ### Experiment Settings
> > > > >
> > > > > For all ImageNet experiments, we train ViT-S from scratch for 200 epochs with a batch size of 512 across 8 A100 GPUs. In the ViT-S architecture, the layers [0, 3, 6, 9] are replaced with MoE layers. The learning rate is set to 1e-4, and we use the Adam optimizer with parameters [0.9, 0.99] and cosine learning schedule, while the weight decay is set to 5e-5. During evaluation, we set the batch size to 128 and use 1 A100 GPU.
> > > > >
> > > > > - **Baselines**: We use DeepSpeed-MoE with top-1/2 gating, and the number of experts in each layer is set to 4.
> > > > > - **DynMoE**: We use the same settings as in the MoE-LLaVA experiments, where the initial number of experts is set to 2, with a maximum of 4 experts per layer. We evaluate two variants of DynMoE:
> > > > >     - **DynMoE-A**: DynMoE with a `simple and diverse gating loss`
> > > > >     - **DynMoE-B**: DynMoE with `simple and diverse gating loss` + `load balance loss` + `efficiency loss`
> > > > >
> > > > > From the experiment, we observe the following:
> > > > >
> > > > > 1. **DynMoE-A**: Achieves slightly lower performance compared to the Top-2 MoE, but with a smaller average $k$ and reduced training/inference time.
> > > > > 2. **DynMoE-B**: When we add the load balance loss and efficiency loss, DynMoE-B not only improves accuracy but also enhances efficiency.
> > > > >
> > > > > |  | Train Time (s/batch) | Train Wall-Clock Time (days) | Evaluation Time (s/batch) | Inference Time on Routing (ms / batch and MoE layer) | Inference Time on Gating (ms / batch and MoE layer) | Inference Time on Expert Passing (ms / batch and MoE layer) | Acc@1 (%) |
> > > > > | --- | --- | --- | --- | --- | --- | --- | --- |
> > > > > | DeepSpeed-MoE (top-1) | 0.39 | 2.3 | 0.53 | 0.06 | 14.54 | 94.32 | 64.7 |
> > > > > | DeepSpeed-MoE (top-2) | 0.63 | 3.7 | 1.05 | 0.06 | 50.78 | 186.47 | 67.3 |
> > > > > | DynMoE-A (k=1.63) | 0.51 | 2.8 | 0.99 | 0.53 | 36.45 | 186.18 | 66.5 |
> > > > > | DynMoE-B (k=1.25) | 0.41 | 2.5 | 0.82 | 0.46 | 31.04 | 148.11 | 68.5 |
> > > > >
> > > > > We hope these new results provide additional insights and offer a clearer comparison. Thank you once again for your valuable efforts in reviewing our paper.

---

> ### Author Response · Authors · 2024-11-29
> **To reviewer 4q9o**
>
> Dear Reviewer 4q9o,
>
> Thank you for your time and effort in reviewing our paper. We appreciate your constructive feedback and understand your concerns regarding the potential limitations of DynMoE. We would like to address the following points:
>
> - **Potential misunderstanding in comments:** We believe there may be some misunderstanding regarding the reviewer's concerns. In both your initial comments and your first response, the focus appears to be on whether we conducted a fair comparison (to MoE-LLaVA). Specifically, your feedback includes:
>     - *"Unconvincing real latency/throughput evaluation"*
>     - *"I am still leaning towards rejection of this work. I am not convinced by the authors' response that the throughput evaluation is fair."*
>
>     In response to these comments, we made sure to address the fairness of our evaluation in our previous reply, and we believe that we adequately clarified this point. This is why we referenced the ‘Old Request’ and ‘New Request’ in our earlier response, as we felt this issue had already been resolved.
>
> - **Overhead of the MoE Algorithm:** We agree with your observation that the newly designed MoE algorithm may introduce additional overhead. As discussed in our [previous response](https://openreview.net/forum?id=T26f9z2rEe&noteId=ZO8CvARjdD), the main source of overhead comes from the routing process, which is likely the reason why DynMoE (with forced top-1) is clearly slower than MoE (also with forced top-1). We have also provided more details on this aspect below
>
>
>     | Time (per sample and MoE layer) | Routing (ms) | Gating (ms) | Expert Passing (ms) |
>     | --- | --- | --- | --- |
>     | MoE-LLaVA (enforced top-1) | 0.04 | 0.60 | 1.08 |
>     | DynMoE (enforced top-1) | 0.45 | 0.75 | 1.10 |
> - **Future Work on Efficient MoE Implementations:** We acknowledge that further optimizations are needed to reduce the overhead of our approach. In the revised version of the paper, we will
>     - provide more detailed experimental settings for each table;
>     - include the discussion of efficiency as outlined in our rebuttal;
>     - clearly discuss the limitations regarding if the dynamic mechanism can be implemented efficiently on GPUs without introducing any overhead.
>
> **While we recognize the potential limitations, we believe our key findings—particularly regarding the auto-tuning approach—are significant and offer valuable new insights for the field.**
>
> Once again, we appreciate your detailed review and constructive feedback. Thank you.

---

### Official Review · Reviewer_y8zU · 2024-11-03

**Soundness:** 3
**Presentation:** 2
**Contribution:** 3
**Rating:** 8
**Confidence:** 3

**Summary:**

This work presents DynMoE a mixture-of-experts approach which uses 2 mechanisms:

  * Top-any routing which computes a score for each token-expert pair and a learnable threshold per expert, therefore allowing a rather dynamic use of tokens/expert computations.
  *  keeping track of expert utilisation to grow/shrink throughout the training, and initialising the new experts routing based on the non-routed tokens.

It shows results in vision, language, and vision-language tasks and shows it comparable to training with a fixed (but sweeped) number of K experts and top-k tokens routed.

**Strengths:**

Together, the two proposed mechanisms achieve an expert architecture which automatically decides the number of expert activations and the total number of experts throughout the training. Those are two problems that if addressed correctly can make MoE simpler to use by not having to fix them ahead of time.

**Weaknesses:**

In my understanding, early design of MoEs [Shazeer et al., 2017] had many engineering challenges in consideration such as efficient use of compute units and network, which led to many of the fixed top-k routing decisions (e.g. why should the compute unit that processes expert A process only 10 tokens.. if it has to wait for another compute unit that has to process 100 tokens?) which is also similar to the concerns of equal utilisation of experts with the purpose to efficiently utilise hardware. With that in mind I find it hard to understand from the text in this paper: (a) why those concerns don’t apply here, (b) how has hardware changed overtime such they are no longer important, (c) if the approach here would not scale to utilise hardware as efficiently as a fixed top-k routing as one scales the method.

Unclear on what mechanisms are at play to optimise the gating mechanisms. In particular it appears to this reviewer that the only gradient to the gating and gating thresholds is via the (1/k) factor of eq (6).. If so one can tend to understand that no per-expert specific gradient is observed.. By that I mean, for a token, all of the params used to compute its gating values will observe similar gradient directions regardless of which expert they refer to.

**Questions:**

My main questions are highlighted on the main weaknesses above.


After rebuttal from authors with added experiments on (1) Included a load-balance loss in our model; (2) Introduced an efficiency loss to enforce sparsity, as suggested in [2]. I have raised my score from below acceptance to accept.

---

> ### Author Response · Authors · 2024-11-22
> **Reply to Reviewer y8zU (1/2)**
>
> Thank you for your valuable suggestions! We have addressed the concerns you raised below.
> > In my understanding, early design of MoEs [Shazeer et al., 2017] had many engineering challenges in consideration such as efficient use of compute units and network, which led to many of the fixed top-k routing decisions (e.g. why should the compute unit that processes expert A process only 10 tokens.. if it has to wait for another compute unit that has to process 100 tokens?) which is also similar to the concerns of equal utilisation of experts with the purpose to efficiently utilise hardware. With that in mind I find it hard to understand from the text in this paper: (a) why those concerns don’t apply here, (b) how has hardware changed overtime such they are no longer important, (c) if the approach here would not scale to utilise hardware as efficiently as a fixed top-k routing as one scales the method.
> >
>
> Thank you for your valuable feedback! We would like to provide the following explanation regarding your concerns:
>
> - In our submission, we did not enforce an equal load distribution among experts based on the following considerations:
>     1. The significance of experts varies, with some being more likely to be activated by all tokens. This issue is recognized in recent Mixture of Experts (MoE) studies [1] and is also observed in our research, as discussed in Section 4.5.
>     2. Enforcing load balance by including a load-balance loss might not align with our requirements for sparse routing, as activating all experts could be considered a form of load balance.
>
>     We believe that this issue can be addressed through engineering techniques, such as allocating additional computation units to shared experts.
>
> - To further illustrate our point, we have conducted additional ablation studies on the MoE-LLaVA setting. Specifically, we have: (1) Included a load-balance loss in our model; (2) Introduced an efficiency loss to enforce sparsity, as suggested in [2]. We report the following metrics:
>     1. **Performance (Table 1):** The performance of different settings.
>     2. **Load Balance  (Table 2):** The frequency with which each expert is activated, calculated as (expert activation time / total token count).
>     3. **Efficiency  (Table 3):** The top-k values per layer.
>     4. **Efficiency  (Table 4):** The top-k activation frequency, calculated as (number of tokens that activate k experts / total tokens).
>
>     We can find that
>
>     1. Although it does not explicitly enforce load balancing, the original DynMoE achieves load balancing comparable to that of the standard top-2 MoE (Table 2).
>     2. Adding the load balance loss slightly decreases the performance of DynMoE (Table 1) while increasing the number of activated experts (Table 3). However, it improves load balancing (Table 2).
>     3. Adding an additional efficiency loss on top of the load balance loss improves performance (Table 1) and helps overcome some extreme cases, such as the reduction of the top-$k$ values in the bottom layer from 2.88 to 2.02 (Table 3), and reduce the number of tokens that activate all 4 experts (Table 4). Moreover, the efficiency loss further enhances load balancing (Table 2).
>
> | Performance (StableLM) | VQAv2 | GQA | VizWiz | SQA | TextVQA | POPE | MME | MMBench |
> | --- | --- | --- | --- | --- | --- | --- | --- | --- |
> | DynMoE | 77.4 | 61.4 | 40.6 | 63.4 | 48.9 | 85.7 | 1300.9 | 63.2 |
> | DynMoE + load balance | 77.1 | 61.6 | 37.0 | 61.4 | 50.3 | 85.3 | 1313.5 | 61.7 |
> | DynMoE + load balance + efficiency | 77.1 | 61.8 | 39.4 | 62.9 | 49.7 | 85.4 | 1321.2 | 61.9 |
>
> | Activation Frequency per Expert (VQAv2, layer 0) | Expert 1 | Expert 2 | Expert 3 | Expert 4 |
> | --- | --- | --- | --- | --- |
> | MoE (top-2) | 0.36 | 1.29 | 0.16 | 0.19 |
> | DynMoE | 0.29 | 0.97 | 0.48 | 0.35 |
> | DynMoE + load balance | 0.81 | 0.50 | 0.90 | 0.68 |
> | DynMoE + load balance + efficiency | 0.45 | 0.52 | 0.42 | 0.63 |
>
> | Top-k per Layer (VQAv2) | Layer 0 | Layer 2 | Layer 4 | Layer 6 | Layer 8 | Layer 10 | Layer 12 | Layer 14 | Layer 16 | Layer 18 | Layer 20 | Layer 22 |
> | --- | --- | --- | --- | --- | --- | --- | --- | --- | --- | --- | --- | --- |
> | DynMoE | 2.09 | 1.07 | 1.57 | 1.06 | 2.04 | 1.03 | 1.03 | 1.00 | 1.03 | 1.02 | 1.02 | 1.00 |
> | DynMoE + load balance | 2.88 | 1.25 | 1.59 | 1.27 | 1.26 | 1.13 | 1.77 | 1.70 | 1.12 | 1.33 | 1.30 | 1.00 |
> | DynMoE + load balance + efficiency | 2.02 | 1.25 | 1.81 | 1.57 | 1.65 | 1.20 | 1.47 | 2.30 | 1.07 | 1.37 | 1.82 | 1.00 |
>
> | Top-k Frequency (VQAv2) | Top-1 | Top-2 | Top-3 | Top-4 |
> | --- | --- | --- | --- | --- |
> | DynMoE | 0.79 | 0.16 | 0.04 | 0.01 |
> | DynMoE + load balance | 0.65  | 0.26  | 0.06  | 0.03 |
> | DynMoE + load balance + efficiency | 0.58 | 0.32 | 0.09 | 0.01 |

---

> ### Author Response · Authors · 2024-11-22
> **Reply to Reviewer y8zU (2/2)**
>
> > Unclear on what mechanisms are at play to optimise the gating mechanisms. In particular it appears to this reviewer that the only gradient to the gating and gating thresholds is via the (1/k) factor of eq (6).. If so one can tend to understand that no per-expert specific gradient is observed.. By that I mean, for a token, all of the params used to compute its gating values will observe similar gradient directions regardless of which expert they refer to.
> >
>
> We would like to clarify the following points:
>
> 1. The gradients of the gating network are dependent on the expert outputs $E_{e}(x)$, which vary among experts. Consequently, the gradients of the gating parameters corresponding to different experts will also differ.
> 2. In Figures 17-22 of the Appendix, we have reported the similarities of the gating parameters and the learned threshold $G$. *Our findings indicate that the learned parameters and $G$ corresponding to different experts are significantly different, even though $G$ is initially assigned the same value.*
>
> [1] DeepSeekMoE: Towards Ultimate Expert Specialization in Mixture-of-Experts Language Models.
>
> [2] Learn To be Efficient: Build Structured Sparsity in Large Language Models.

---

> > ### Author Response · Authors · 2024-11-25
> > **A Kind Reminder for Reviewer y8zU**
> >
> > Dear Reviewer y8zU,
> >
> > We greatly appreciate your valuable feedback and thorough review of our paper. Your insights have been crucial in helping us refine and strengthen our work. In response to the concerns you raised, **we have provided detailed explanations and added new experiments to thoroughly address each point.** Below, we summarize our key responses to your feedback:
> >
> > - **Discussion on load balancing:** We clarified the reasoning behind not using a load balancing loss in our initial submission and have added further experiments that incorporate both load balancing loss and efficiency loss to further improve DynMoE's load balancing.
> > - **Clarification on gradients:** We explained that the gradients of the router parameters are influenced by expert outputs, meaning they may not always move in the same direction.
> >
> > We greatly appreciate the time and effort you've dedicated to reviewing our paper. Your feedback has played a crucial role in enhancing the quality of our manuscript.
> >
> > As the rebuttal period concludes, we believe we have addressed all of your concerns and kindly ask that you reconsider your evaluation. If you have any additional questions or comments, please feel free to reach out.

---

> > > ### Author Response · Authors · 2024-12-02
> > > **Kind Reminder for Reviewer y8zU**
> > >
> > > Dear Reviewer y8zU,
> > >
> > > Thank you for your thoughtful and constructive comments on our manuscript. We have carefully addressed all of your concerns in our rebuttal.
> > >
> > > If you have any further questions or need additional clarification, please do not hesitate to reach out. We will be happy to provide any further information and ensure that all aspects of your feedback are fully addressed. Your input is invaluable in enhancing the quality of our work.
> > >
> > > As the extended discussion period is coming to a close, and we have not yet received a response from you, we kindly request your timely feedback to allow us sufficient time to resolve any remaining issues. We greatly appreciate your time and effort in reviewing our manuscript.
> > >
> > > Best regards,
> > >
> > > The Authors

---

### Author Response · Authors · 2024-11-22
**Summary of Rebuttal**

We would like to thank all the reviewers for their detailed feedback. In response to their recommendations, we have conducted additional experiments and made modifications to the paper. Below, we summarize the key changes:

**New Experiments and Results:**

- **[Reviewers y8zU and 4q9o]**: Added load-balance and efficiency loss terms on top of DynMoE to improve load balancing while maintaining efficiency.
- **[Reviewer 4q9o]**: Added an additional top-$p$ gating baseline.
- **[Reviewer 4q9o]**: Reported throughput of the dense model.
- **[Reviewer 4q9o]**: Trained ViT-S from scratch on the CIFAR-10 dataset.
- **[Reviewer 4q9o]**: Reported standard deviations for language experiments.
- **[Reviewers 4q9o and kDxr]**: Investigated additional expert initialization strategies when adding new experts.
- **[Reviewers 4q9o and kDxr]**: Conducted ablation studies on weighted gating scores.
- **[Reviewer DhtB]**: Fine-tuned ViT-S on the TinyImageNet dataset.

**Modifications:**

- **[Reviewer 4q9o]**: Corrected typos and improved grammar throughout the paper.
- **[Reviewer 4q9o]**: Expanded the discussion on related top-any MoE papers (lines 103-107).
- **[Reviewer DhtB]**: Added a discussion on related work on dynamic neural networks (lines 535-539).
- **[Reviewers 4q9o, kDxr, and DhtB]** Added a convergence curve on CIFAR10 and TinyImageNet experiments on Figure 24 of page 25.

---

### Meta-Review · Area_Chair_PCBh · 2024-12-24

**Metareview:**

The paper introduces Dynamic Mixture of Experts (DynMoE), a novel and practical framework designed to enhance the efficiency of Transformer models. By eliminating the need for manual hyperparameter tuning, DynMoE dynamically determines the number of experts to activate for each token and adjusts the total number of experts during training. This auto-tuning mechanism addresses significant challenges in the scalability and usability of MoE models. Extensive experiments across vision, language, and vision-language tasks demonstrate DynMoE's effectiveness, achieving competitive performance with fewer activated parameters compared to existing MoE variants. Importantly, the method provides a streamlined approach to deploying MoE models while maintaining robust performance across diverse domains.

The reviewers broadly recognized the strengths of the paper, particularly its ability to simplify the use of MoE frameworks by automating expert selection and adjustment, a contribution with strong practical implications. DynMoE’s efficient and adaptive design was praised for reducing computational overhead while preserving or even improving task performance. The authors’ comprehensive evaluations, including tests on multiple datasets and tasks, further reinforced the method’s scalability and applicability. The experiments provided compelling evidence of DynMoE’s advantages over traditional top-k gating approaches, particularly in resource-constrained environments.

Reviewer 4q9o maintained a critical position, questioning the novelty of the proposed approach and highlighting gaps in baseline comparisons, latency evaluation, and the absence of from-scratch training experiments in the original submission. However, the authors provided extensive rebuttals and addressed these concerns by conducting from-scratch experiments, expanding the baseline comparisons, and reporting standard deviations to improve result interpretability. Additional latency and throughput metrics were also included, demonstrating that DynMoE maintains competitive efficiency.

Given the paper’s innovative approach, robust empirical results, and the significant efforts to address all reviewer concerns, the AC recommends acceptance. DynMoE represents an impactful advancement in the usability and efficiency of MoE models, providing both practical value and a strong foundation for future research. The strengths of the paper, particularly its ability to generalize across multiple domains and its scalability, outweigh the remaining criticisms regarding novelty and implementation details. This work is a valuable contribution to the field of efficient model training.

**Additional Comments On Reviewer Discussion:**

Please refer to the meta review.

---

### Decision · Program_Chairs · 2025-01-22

Accept (Poster)